# Enhancing Optimizer Stability: Momentum Adaptation of The NGN Step-size

Rustem Islamov[1]   Niccoló Ajroldi[2]   Antonio Orvieto[2,3,4]   Aurelien Lucchi[1]
[1]University of Basel   [2]Max Planck Institute for Intelligent Systems
[3] ELLIS Institute Tübingen   [4]Tübingen AI Center

## Abstract

Modern optimization algorithms that incorporate momentum and adaptive step-size offer improved performance in numerous challenging deep learning tasks. However, their effectiveness is often highly sensitive to the choice of hyperparameters, especially the learning rate (LR). Tuning these parameters is often difficult, resource-intensive, and time-consuming. Therefore, recent efforts have been directed toward enhancing the stability of optimizers across a wide range of hyper-parameter choices [79]. In this paper, we introduce an algorithm that matches the performance of state-of-the-art optimizers while improving stability through a novel adaptation of the NGN step-size method [66]. Specifically, we propose a momentum-based version (NGN-M) that attains the standard convergence rate of $\mathcal{O}(1/\sqrt{K})$ under common assumptions, without the need for interpolation condition or assumptions of bounded stochastic gradients or iterates, in contrast to previous approaches. Additionally, we empirically demonstrate that the combination of the NGN step-size with momentum results in high robustness while delivering performance that is comparable to or surpasses other state-of-the-art optimizers.

## 1 Introduction

Adaptive methods such as Adam [44] and RMSprop [30] are widely used in machine learning due to their established advantages over (momentum) SGD, particularly in tasks such as training Transformers [8, 88, 89]. These methods adaptively scale the step-size across different dimensions (parameters) based on their respective statistics, effectively acting as a diagonal preconditioning.

Although these methods perform well in practice, existing theoretical analyses typically require stringent assumptions on the noise structure of the stochastic gradients, such as sub-Gaussian noise [49] or affine noise models [90, 106]: Relaxing these assumptions remains an open challenge. Another well-known issue of Adam is its performance sensitivity to the LR hyperparameter [96, 10], particularly when training Transformers, where loss spikes are commonly observed [60, 97]. This often necessitates careful adjustments of the hyperparameters throughout the training process [107, 11], which can be costly in terms of computational resources [64]. Consequently, there has been growing interest in developing optimization methods that are more robust to hyperparameter selection [79]. In addition to adapting LR, Adam and other state-of-the-art optimizers also rely on momentum [71], a broadly used technique that has been shown to enhance performance both theoretically [14, 19, 35, 36] and practically [10, 21, 38]. Besides speeding up convergence, momentum is known as a technique to reduce the variance of stochastic algorithms [57, 15], improving stability as well as generalization in some settings [38].

In this work, we address the aforementioned drawbacks of Adam by developing a new algorithm based on the recently proposed NGN step-size [66], an improved variant of the Stochastic Polyak Step-size [54] that has demonstrated strong resilience to LR hyperparameter tuning. In particular,

NGN was shown never to diverge for any choice of the LR hyperparameter in the convex setting, and to exhibit strong curvature adaptation properties strengthened by theoretical guarantees. However, the step-size of Orvieto and Xiao [66] simply adapts the LR through a scalar multiplier, leaving to future work the incorporation of momentum and coordinate-wise variants – needed in complex problems such as optimizing transformers, as motivated above. Here, we develop a momentum and step-size adaptive version of NGN designed to enhance robustness[1] in terms of hyperparameter selection. We also present a theoretical analysis alongside a practical evaluation of this approach, showcasing its improvements over current state-of-the-art methods.

In summary, our contributions are as follows:

1. We introduce a new algorithm named NGN-M that combines the NGN step-size with momentum. We theoretically show that NGN-M achieves a convergence rate $\mathcal{O}(1/\sqrt{K})$ in the convex regime without the typical requirements of interpolation or bounded gradient assumptions found in earlier works on Polyak step-size;

2. We focus on the problem of adapting the step-size rule towards a coordinate-wise diagonal preconditioning. By integrating this diagonal step-size strategy with momentum, we develop a new variant of NGN, called NGN-MD;

3. The theoretical results are supported by extensive empirical validation in various deep learning settings where we demonstrate that NGN-M and NGN-MD not only preserve the robustness property of the NGN step-size, but improve it further in many cases. LR hyperparameter resilience comes together with better or comparable performance to state-of-the-art algorithms.

## 2  Related Works

**Polyak Step-size.**   When training a deep network with standard optimizers, a tuned LR is crucial but time-consuming and resource-intensive [26]. This issue is at the root of recent research focusing on transferring hyperparameters across architectures at different scales, therefore avoiding expensive tuning pipelines [99, 100, 7]. Yet, in the convex setting, choosing LR can already be difficult – an issue that was studied already in Polyak [72] and gave rise to the first adaptive method: the Polyak Stepsize (PS). Recently, there has been a renewed interest in adapting PS to modern settings [54, 67, 39], delivering a theoretically principled way to scale the gradient magnitude during training adaptively. PS-inspired methods have gained increasing interest for their simplicity and adaptability, as they utilize local curvature and smoothness information to accelerate algorithms and facilitate faster convergence. Orvieto and Xiao [66] recently introduced a variant of the Stochastic Polyak step-size, called NGN, which further enhances the robustness to LR hyperparameter and solidifies the link to Gauss-Newton preconditioning. The theoretical analysis in Orvieto and Xiao [66] demonstrated that NGN does not diverge regardless of the choice of LR hyperparameter, and converges fast when the LR is appropriately tuned. In contrast, the current theory of the SPS step-size with fixed LR hyperparameters [54] proves convergence to the exact solution only if the interpolation condition holds[2].

**Polyak Step-size and Heavy-ball Momentum.**   Heavy-ball momentum methods, stemming from the work of Polyak [71], have gained significant attention over the years due to their benefits, including acceleration on convex quadratics [37, 48, 6], convex-like [92], and non-convex problems [14], as well as their variance reduction abilities [57, 15]. This has led to growing interest in the combination of Polyak step-size and heavy-ball momentum, which is an active area of research [3, 77, 3, 93, 63, 28]. Recently, Schaipp et al. [79] demonstrated that a geometrically principled combination of SPS and momentum leads to lower sensitivity to LR hyperparameter, although they did not provide strong theoretical convergence guarantees.

**Diagonal Polyak Step-size.**   Coordinate-wise adaptive step-sizes are essential in training Transformer architectures due to the varying parameter-wise scaling and conditioning of the problem

---

[1]It is worth emphasizing that the terms robustness and stability have been used in a different sense in the literature [12, 103]. In this work, we focus on the stability of the choice of the LR hyperparameter.

[2]In our notation, this means that $\sigma_{\text{int}}^2 = 0$.

Table 1: Summary of existing methods exploiting Polyak-type adaptive step-sizes and their convergence guarantees. **Mom.**=Supports momentum; **Diag.**=Supports diagonal step-sizes. $\sigma_{\text{int}}^2$ is defined in Section 4. $\mathcal{O}$ notation hides absolute, problem-dependent constants and logarithmic factors.

| Method | Rate [a] | Mom. | Diag. | Comments |
|:---:|:---:|:---:|:---:|:---:|
| SPS$_{\text{max}}$ [54] | $\mathcal{O}(1/K + \sigma_{\text{int}}^2)$ | ✗ | ✗ | Conv. to non-vanishing neighbourhood |
| ALR-SMAG [93] | $\mathcal{O}((1-\rho)^K + \sigma_{\text{int}}^2)$ | ✓ | ✗ | Strong convexity Conv. to non-vanishing neighbourhood |
| Momo [79] | $\mathcal{O}(1/\sqrt{K})$ | ✓ | ✗ | Bounded stoch. gradients Interpolation |
| Momo-Adam [79] | ✗ | ✓ | ✓ | Momo framework for Adam |
| MomSPS$_{\text{max}}$ [63] | $\mathcal{O}(1/K + \sigma_{\text{int}}^2)$ | ✓ | ✗ | Conv. to non-vanishing neighbourhood |
| NGN [66] | $\mathcal{O}(1/\sqrt{K})$ | ✗ | ✗ | – |
| IAM [28] | $\mathcal{O}(1/\sqrt{K})$ | ✓ | ✗ | Knowledge of $f_i(x^*)$ |
| NGN-M (Alg. 1) [This work] | $\mathcal{O}(1/\sqrt{K})$ | ✓ | ✗ | – |
| NGN-MDv1 and NGN-MDv2 (Alg. 2) [This work] | ✗ | ✓ | ✓ | Combination of NGN-M and NGN-D |
| NGN-D (Alg. 3) [This work] | $\mathcal{O}(1/\sqrt{K})$ | ✗ | ✓ | – |

[62, 108]. Algorithms employing parameter-wise LR, such as Adam and SignSGD [4], typically outperform non-diagonal methods in language modeling tasks by addressing issues such as class imbalance (where certain words appear more frequently than others) [46, 47] and heavy-tailed noise [104, 105, 13]. It is, therefore, paramount in current setups to deliver adaptive LR improvements targeted to the coordinate-wise (diagonal) regime. However, most Polyak-step-size-based algorithms only focus on a single LR for all parameters [54, 93, 27, 63, 66]. Only a few works propose a diagonal-wise modification of Polyak-step-size by either using Adam preconditioner [79] as a weight matrix or incorporating second-order information from the objective function [51, 76].

**Quantitative Measure of Robustness.** To quantify robustness, we adopt the learning-rate (LR) sensitivity metric of Wortsman et al. [97]. Let $\ell_\gamma$ denote the final performance metric (e.g., negative test accuracy) when training with LR $\gamma$, and $\ell_0$ the value at initialization. We define $\ell^* := \min_{\gamma \in [a,b]} \ell_\gamma$ as the best achievable metric within the LR range $[a, b]$. The LR sensitivity is then $\mathbb{E}_{\gamma \in [a,b]}[\min\{\ell_0, \ell_\gamma\} - \ell^*]$. We estimate this expectation by averaging over the LR values in our sweep grid for each algorithm and task.

**Comparison to prior work.** Table 1 provides a theoretical comparison of various Polyak step-size-based algorithms that incorporate momentum and/or diagonal step-size, highlighting the differences between the theoretical results presented in this work and those from prior works.

## 3 Algorithm design of NGN-M and NGN-D

In Orvieto and Xiao [66], the NGN step-size is derived by applying a Gauss–Newton update on a regularized first-order expansion of $r(x) := \sqrt{f(x)}$. At the current point $x^k$, they linearized $r(x^k + p) \approx r(x^k) + \nabla r(x^k)^\top p$. Thus the next iterate is given as $x^{k+1} = x^k + p^k$ where

$$p^k := \text{argmin}_p \left[ (r(x^k) + \nabla r(x^k)^\top p)^2 + \frac{1}{2c}\|p\|^2 \right]. \qquad (1)$$

It turns out that the problem above has a closed-form solution

$$p^k = -\gamma_k \nabla f(x^k) \quad \text{where} \quad \gamma_k := \frac{c}{1 + \frac{c}{2f(x^k)}\|\nabla f(x^k)\|^2},$$

with $\gamma_k$ representing the NGN step-size. In Orvieto and Xiao [66], convergence guarantees were established for both convex and general non-convex settings. Importantly, the convex analysis shows

that NGN exhibits a non-divergence property, regardless of the step-size hyperparameter $c$ (see Theorem 4.5 in [66]). Due to this property, the NGN step-size is a strong candidate to achieve better robustness w.r.t. the choice of LR hyperparameter.

## 3.1 How to Add Momentum and What to Expect?

There are several approaches to combining the adaptive Polyak-type step-size with heavy-ball momentum. Broadly, existing algorithms can be divided into two categories: the first category involves computing the Polyak step-size in the usual manner and incorporating it into the standard heavy-ball update [63]. In contrast, algorithms from the second category first determine an update direction using exponential weighted averaging of the stochastic gradient and momentum variable, and then compute the Polyak-type step-size based on the computed direction [93, 79]. Following this principled approach, we test two possible versions for combining the NGN step-size and momentum:

$$\text{Ver.1}: \begin{cases} \gamma_k = \frac{c}{1 + \frac{c}{2f_{S_k}(x^k)}\|\nabla f_{S_k}(x^k)\|^2} \\ m^k = \beta m^{k-1} + (1-\beta)\gamma_k \nabla f_{S_k}(x^k) \\ x^{k+1} = x^k - m^k \end{cases} \quad \text{Ver.2}: \begin{cases} m^k = \beta m^{k-1} + (1-\beta)\nabla f_{S_k}(x^k) \\ \gamma_k = \frac{c}{1 + \frac{c}{2f_{S_k}(x^k)}\|m^k\|^2} \\ x^{k+1} = x^k - \gamma_k m^k \end{cases} .$$

Before we proceed, we should answer the question: *"What do we expect from the combination of NGN step-size and momentum?"*. First, we aim to preserve, and ideally enhance, NGN's robustness to the LR hyperparameter. To this end, we propose incorporating (heavy-ball) momentum, which is known to increase the range of LR that leads to convergence [71]. Additionally, we seek improved performance, achieving accelerated convergence akin to the advantage of SGD with momentum (SGDM) over standard SGD in convex settings. With these goals in mind, we now show that version 1 meets all of these criteria, while version 2 is less suitable. To gain some intuition regarding the performance of these two variants, we start by conducting a simple experiment on a quadratic function $f(x) = \frac{1}{2}\|Ax - b\|^2$ where $A$ is a data matrix from the normalized Diabetes dataset [85] and $b$ is a vector of labels. Based on the results from Figure 1 (first), we observe that variant 1 achieves accelerated convergence as SGDM for middle-range step-size hyperparameters ($c \in \{10^1, 10^2\}$) and does not diverge for large LR hyperparameter ($c \in \{10^3\}$). Conversely, version 2 has a worse convergence rate than version 1 for middle-range LR hyperparameters and diverges for large ones ($c \in \{10^3\}$): see Figure 1 (second). Therefore, we theoretically analyze and practically test version 1, which we call NGN-M.

## 3.2 Evidence of Robustness of NGN-M

To illustrate the advantages of the design choice of NGN-M, we first consider the Rosenbrock function $f(x, y) = (x - 1)^2 + 100(y - x^2)^2$, whose minimizer is at $(1, 1)$. Starting from $(-1.2, 1)$, we run both NGN-M and SGDM over a wide range of LR hyperparameters $\{10^{-3}, \ldots, 10^2\}$. As shown in Figure 1, we observe that $(i)$ for small LR hyperparameter both methods successfully converge to $(1, 1)$; $(ii)$ SGDM already diverges for LR hyperparameter $10^{-2}$; By contrast, NGN-M remains stable even up to $c = 10^2$, thanks to its adaptive LR that automatically adjusts with the local curvature. Figure H.3 further traces the optimization trajectories: NGN-M converges reliably for every tested value of $c$, whereas SGDM fails outside its narrow stability window. Finally, in Appendix H.1 we repeat these experiments on a synthetic multi-modal function and find that NGN-M consistently finds the global minimum, while SGDM typically becomes trapped in a nearby suboptimal local minimum.

## 3.3 Diagonal Step-size for NGN

We propose two alternatives to make NGN step-size parameter-wise adaptive. In the first approach, we modify an approach of (1): The next iterate $x^{k+1}$ is obtained by minimizing an approximation of the regularized first-order Taylor expansion of $r(x) := \sqrt{f(x)}$ around $x^k$, namely, $x^{k+1} = x^k + p^k$ where for a preconditioning matrix $\Sigma_k$

$$p^k = \operatorname{argmin}_p \left[ (r(x^k) + \nabla r(x^k)^\top p)^2 + \frac{1}{2c}\|p\|_{\Sigma_k}^2 \right]. \tag{2}$$

The intuition is that $\Sigma_k \in \mathbb{R}^{d \times d}$ can penalize each parameter with its own weight while in vanilla NGN the penalization is the same for all parameters, and $f$ is an objective function we aim to minimize.

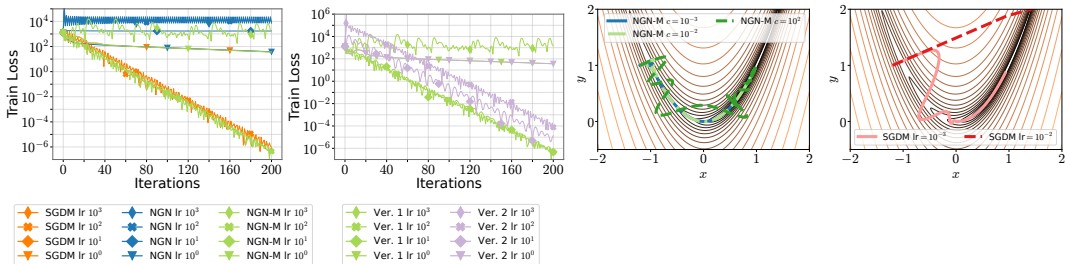

Figure 1: **First:** Comparison of SGDM, NGN, NGN-M for linear regression on normalized Diabetes dataset varying a step-size hyperparameter. **Second:** Comparison of two options on how momentum can be used in combination with NGN step-size. **Third and fourth:** Comparison of SGDM and NGN-M on the Rosenbrock function.

Performing simple derivations (see Appendix F), we obtain the following update rule

$$x^{k+1} = x^k - \frac{c}{1 + \frac{c}{2f(x^k)}\|\nabla f(x^k)\|^2_{\boldsymbol{\Sigma}_k^{-1}}}\boldsymbol{\Sigma}_k^{-1}\nabla f(x^k). \tag{3}$$

Note that by choosing $\boldsymbol{\Sigma}_k$ to be an identity matrix, the step-size $\gamma_k$ in (3) reduces to the vanilla NGN step-size.

Alternatively, we can adopt a simpler, parameter-wise rule: For each parameter $j$, we replace the full gradient norm in the NGN step-size with its own partial derivative $\nabla_j f_{S_k}(x^k)$. Both of the described per-coordinate variants can be further adjusted by an RMSprop-style preconditioner $\mathbf{D}_k = \text{diag}((\mathbf{D}_k)_{(1)}, \ldots, (\mathbf{D}_k)_{(d)})$ and lead to the following update rule (see Alg. 2 for a full description)

NGN-MDv1 : $\begin{cases} \gamma_k = \frac{c}{1 + \frac{c}{2f(x^k)}\|\nabla f_{S_k}(x^k)\|^2_{\mathbf{D}_k^{-1}}} \\ \boldsymbol{\Sigma}_k^{-1} = \gamma_k \mathbf{D}_k^{-1} \end{cases}$ NGN-MDv2 : $\begin{cases} \gamma_k^{(j)} = \frac{c/(\mathbf{D}_k)_{(j)}}{1 + \frac{c/(\mathbf{D}_k)_j}{2f(x^k)}(\nabla_j f_{S_k}(x^k))^2} \\ \boldsymbol{\Sigma}_k^{-1} = \text{diag}(\gamma_k^{(1)}, \ldots, \gamma_k^{(d)}) \end{cases}$

$$x^{k+1} = x^k - (1 - \beta_1)\boldsymbol{\Sigma}_k^{-1}\nabla f_{S_k}(x^k) + \beta_1(x^k - x^{k-1})$$

We highlight that both versions have the same number of hyperparameters as Adam. From an empirical evaluation of the two versions of NGN-MD in Figure 2, we observe that the first choice improves the performance of NGN-M while maintaining robustness to the LR hyperparameter. A more detailed discussion on the two versions of NGN-MD algorithms is deferred to Appendix F.1. However, the robustness of NGN-MD also depends on the choice of preconditioner. When the preconditioner is sensitive to variations in the loss landscape or hyperparameters, NGN-MDv may be less robust than NGN-M. In our experiments, we find that the RMSprop preconditioner performs well in practice. Other preconditioners, such as AdaFisher [25], could be integrated into NGN-MD, potentially providing more accurate curvature approximations and improved stability with respect to the LR. We leave a systematic study of these alternatives to future work.

In the special case $\beta_1 = 0$ and $\boldsymbol{\Sigma}_k = \mathbf{I}$, NGN-MDv2 reduces to NGN-D (Algorithm 3). To the best of our knowledge, NGN-D is the first algorithm that uses a per-parameter Polyak-type step-size while achieving the standard $\mathcal{O}(1/\sqrt{K})$ rate under smoothness and bounded noise variance assumptions; see detailed discussion in Appendix C.

## 4 Theoretical Analysis of NGN-M

### 4.1 Problem Formulation and Notation

We consider the classic Empirical Risk Minimization (ERM) problem that typically appears when training machine learning models, namely,

$$\min_{x \in \mathbb{R}^d} \left[ f(x) \coloneqq \frac{1}{n}\sum_{i=1}^n f_i(x) \right], \tag{4}$$

---
**Algorithm 1** NGN-M
---

1: **Input:** $x^{-1} = x^0 \in \mathbb{R}^d$, step-size hyperparameter $c > 0$, momentum parameter $\beta \in [0, 1)$
2: **for** $k = 0, 1, \ldots, K - 1$ **do**
3:    Sample $S_k \subseteq [n]$
4:    $\gamma_k = \frac{c}{1 + \frac{c}{2 f_{S_k}(x^k)} \|\nabla f_{S_k}(x^k)\|^2}$
5:    $x^{k+1} = x^k - (1 - \beta)\gamma_k \nabla f_{S_k}(x^k) + \beta(x^k - x^{k-1})$
6: **end for**

---
**Algorithm 2** NGN-MD
---

1: **Input:** $x^0 \in \mathbb{R}^d$, step-size hyperparameter $c > 0$, momentum parameters $\beta_1, \beta_2 \in [0, 1)$, stabilization parameter $\varepsilon > 0$, second-order momentum $v^0 = 0$
2: **for** $k = 0, 1, \ldots, K - 1$ **do**
3:    Sample $S_k \subseteq [n]$
4:    $v^k = \beta_2 v^{k-1} + (1 - \beta_2)(\nabla f_{S_k}(x^k) \odot \nabla f_{S_k}(x^k))$
5:    $\mathbf{D}_k = \mathrm{diag}(\varepsilon \mathbf{I} + \sqrt{v^k / (1 - \beta_2^k)})$
6:    For NGN-MDv1: $\gamma_k = \frac{c}{1 + \frac{c}{2 f_{S_k}(x^k)} \|\nabla f_{S_k}(x^k)\|^2_{\mathbf{D}_k^{-1}}}$
7:    For NGN-MDv1: $\mathbf{\Sigma}_k^{-1} = \gamma_k \mathbf{D}_k^{-1}$
8:    For NGN-MDv2: $\mathbf{\Sigma}_k^{-1} = \mathrm{diag}(\gamma_k^{(1)}, \ldots, \gamma_k^{(d)})$ where $\gamma_k^{(j)} = \frac{c/(\mathbf{D}_k)_{(j)}}{1 + \frac{c}{2 f_{S_k}(x^k) \cdot (\mathbf{D}_k)_{(j)}} (\nabla_j f_{S_k}(x^k))^2}$
9:    $x^{k+1} = x^k - (1 - \beta_1)\mathbf{\Sigma}_k^{-1} \nabla f_{S_k}(x^k) + \beta_1(x^k - x^{k-1})$
10: **end for**

---

where $x$ are the parameters of a model we aim to train, $n$ is the number of data points in the dataset, $d$ is the number of parameters, $x^*$ denotes the solution to (4), and $f_i$ represents the loss associated with the $i$-th data point/batch. We assume that each $f_i$ is differentiable and non-negative[3] and that the global optimal value is bounded, i.e. $f^* = \mathrm{argmin}_x f(x) \in \mathbb{R}$. Moreover, we assume that we have access to mini-batch stochastic losses $f_S$ during training such that $f_S^* := \mathrm{argmin}_x f_S(x) < \infty$ for any $S \subseteq [n]$ picked uniformly at random.

We analyze the convergence of NGN-M under assumptions that are often used in the analysis of the Polyak step-size [54, 67, 66, 63, 79].

**Assumption 4.1.** Each $f_i$ is convex and $L$-smooth, i.e., for all $x, y \in \mathbb{R}^d$ and $i \in [n]$ we have $\langle \nabla f_i(x), y - x \rangle \geq f_i(x) - f_i(y)$ and $\|\nabla f_i(x) - \nabla f_i(y)\| \leq L\|x - y\|$.

**Assumption 4.2.** The interpolation $\sigma_{\mathrm{int}}^2 := \mathbb{E}_S[f^* - f_S^*]$ and positive $\sigma_{\mathrm{pos}}^2 := \mathbb{E}_S[f_S^*]$ errors are bounded. We say that the interpolation holds if $\sigma_{\mathrm{int}}^2 = 0$, where $S$ is a sampled mini-batch.

### 4.2 Convergence Guarantees

**Theorem 4.3.** *Let Assumptions 4.1, 4.2 hold. Let the step-size hyperparameter $c > 0$ and the momentum parameter $\beta = \frac{\lambda}{1 + \lambda}$ be constants where $\lambda \leq \min\{cL, 0.5(1 + cL)^{-1}(1 + 2cL)^{-1}\}$. Then the iterates of* NGN-M *(Algorithm 1) satisfy*

$$\mathbb{E}\left[f(\overline{x}^{K-1}) - f(x^*)\right] \leq \frac{\|x^0 - x^*\|^2 (1 + 2cL)^2}{cK} + 8cL(1 + 2cL)^2 \sigma_{\mathrm{int}}^2 + 2cL \max\{2cL - 1, 0\} \sigma_{\mathrm{pos}}^2,$$

*where $\overline{x}^{K-1}$ is chosen uniformly at random from $\{x^0, \ldots, x^{K-1}\}$. Moreover, if we set $c = \mathcal{O}(1/\sqrt{K})$ then we obtain $\mathbb{E}\left[f(\overline{x}^{K-1}) - f(x^*)\right] \leq \mathcal{O}(1/\sqrt{K})$.*

The convergence of NGN-M is provided in the convex setting, which is motivated by recent works that observe convex-like structures in the landscape of networks [34, 31] and agreement between convex theory and practice [80]. Importantly, we show that $(i)$ when the constant $c$ is sufficiently small, NGN-M attains the same convergence rate as SGDM [22]. Moreover, for any choice of $c$, we demonstrate that the NGN-M iterates provably converge to a neighborhood of the optimum and

---

[3]Common losses, e.g. cross-entropy, satisfy this condition.

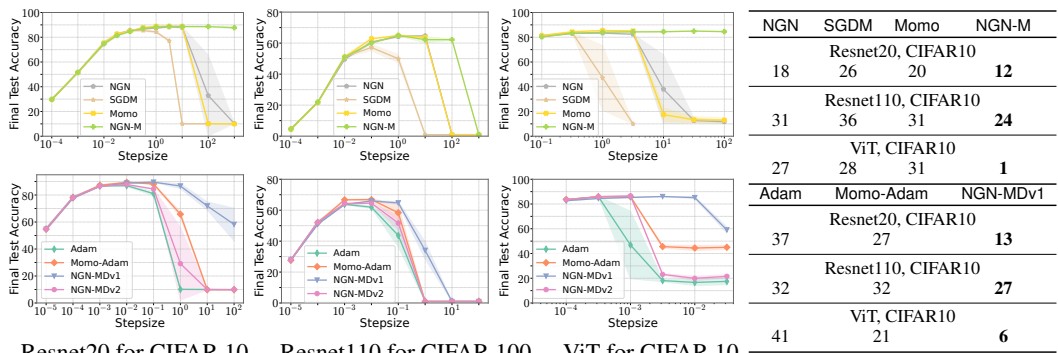

| NGN | SGDM | Momo | NGN-M |
|---|---|---|---|
| Resnet20, CIFAR10 | | | |
| 18 | 26 | 20 | **12** |
| Resnet110, CIFAR10 | | | |
| 31 | 36 | 31 | **24** |
| ViT, CIFAR10 | | | |
| 27 | 28 | 31 | **1** |
| Adam | Momo-Adam | NGN-MDv1 | |
| Resnet20, CIFAR10 | | | |
| 37 | 27 | **13** | |
| Resnet110, CIFAR10 | | | |
| 32 | 32 | **27** | |
| ViT, CIFAR10 | | | |
| 41 | 21 | **6** | |

Resnet20 for CIFAR 10    Resnet110 for CIFAR 100    ViT for CIFAR 10

Figure 2: Stability performance of algorithms varying LR hyperparameter ($c$ for NGN-M, NGN-MDv1 and NGN-MDv2, $\alpha_0$ for Momo and Momo-Adam, and LR for SGDM and Adam). We refer to Figures I.1 to I.3, I.5, and I.8 for the results on additional workloads.

Table 2: Test accuracy LR sensitivity of the different optimizers shown in Figure 2.

thereafter remain within it; $(ii)$ Unlike prior works on Polyak step-size, our analysis does not rely on strong assumptions such as bounded gradients, interpolation, or a bounded domain; $(iii)$ For small values of $c$, NGN-M converges to the exact solution while algorithms such as MomSPS and ALR-SMAG were shown to converge up to a non-vanishing neighborhood of the solution only, due to an inherent limitation of the stochastic Polyak step-size [65]. Regarding the momentum parameter $\beta$, the typical (large) value $\beta = 0.9$ performs well in our own experiments. Theoretically, however, $\beta$ is recommended to be chosen sufficiently small to ensure convergence with the NGN step-size [63]. This discrepancy between theoretical guidance and practical implementation has also been observed in prior works on momentum [23, 53, 93, 92, 63]. Interestingly, under the additional interpolation condition $\sigma_{\text{int}}^2 = 0$, we can establish convergence even for large momentum values, including the commonly used choice $\beta = 0.9$ (see Appendix D.2). This suggests that the small-$\beta$ requirement may reflect limitations of current proof techniques rather than an intrinsic restriction of NGN-M. Extending the analysis to arbitrary $\beta$ in the stochastic regime without interpolation remains an open problem. Our intuition, however, is that simultaneously choosing both $c$ and $\beta$ without restriction is not feasible: fixing one hyperparameter arbitrarily necessitates imposing constraints on the other to prevent divergence; $(v)$ Theorem 4.3 requires the total iteration count $K$ to be specified in advance; this assumption is standard in the complexity analysis of optimization algorithms [24, 61]. Since this can be impractical, we also establish convergence under a diminishing step-size of order $1/\sqrt{k}$ in Appendix D.3, which removes the need to preset $K$; $(vi)$ Finally, we corroborate our analysis as we run NGN-M with the theory-derived values of $c$ to a quadratic problem that satisfies all our assumptions: We observe NGN-M's rapid convergence with theoretical step-size hyperparameters in practice—see Appendix H.3 and Figure H.4 therein.

**Key Ingredients of the Proof.** We discuss the key steps of the proof to highlight the main challenges in the analysis. First, we make use of the Iterative Moving Average (IMA) formulation of momentum [81]. Specifically, we define a sequence of virtual iterates $\{z^k\}$ whose update rule is of the form

$$z^{k+1} = x^k - \gamma_k \nabla f_{S_k}(x^k), \quad x^{k+1} = \frac{\lambda}{1+\lambda} x^k + \frac{1}{1+\lambda} z^{k+1}, \quad \text{where } z^0 := x^0 \text{ and } \beta = \frac{\lambda}{1+\lambda}.$$

Next, one of the key technical strategies we follow is splitting the step-size $\gamma_k$ into two parts: a fixed term $\rho = \frac{c}{(1+cL)(1+2cL)} = \mathcal{O}(c)$ and a changing term $\widetilde{\gamma}_k \leq \frac{3c^2 L}{1+2cL} = \mathcal{O}(c^2)$. This decomposition of the step-size $\gamma_k$ enables us to regulate the balance between the descent term, which drives improvement in the objective, and the error term, which reflects possible inaccuracies. More precisely, the descent term is weighted by $c$ while the error term proportional to $\sigma_{\text{int}}^2$ is weighted by $c^2$, which suggests that $c$ has to be chosen to tradeoff the two terms to lead to the exact convergence similarly to the standard analysis of SGD [22]. In contrast, MomSPS and Momo algorithms achieve the exact convergence only under the interpolation regime.

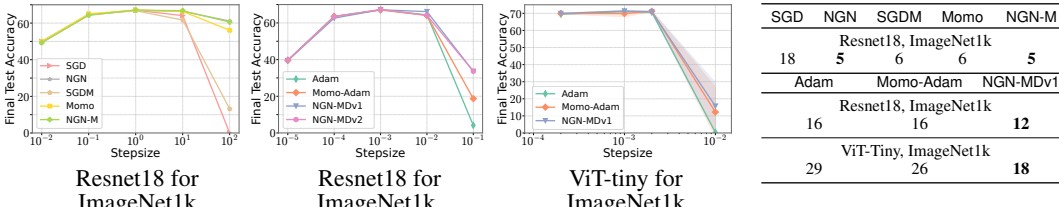

| SGD | NGN | SGDM | Momo | NGN-M |
|---|---|---|---|---|
| | | Resnet18, ImageNet1k | | |
| 18 | **5** | 6 | 6 | **5** |
| Adam | | Momo-Adam | | NGN-MDv1 |
| | | Resnet18, ImageNet1k | | |
| 16 | | 16 | | **12** |
| | | ViT-Tiny, ImageNet1k | | |
| 29 | | 26 | | **18** |

Figure 3: Stability performance on ImageNet1k varying the LR hyperparameter. NGN-M and NGN-MDv1 achieve higher accuracy for a wider range of the LR hyperparameters. We refer to Figure I.4 for results on train loss stability and additional results on ImageNet32.

Table 3: Test accuracy LR sensitivity of the different optimizers shown in Figure 3.

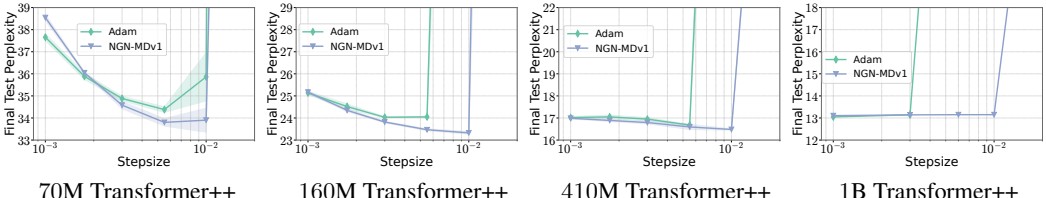

Figure 4: Comparison of stability to the LR hyperparameter across model sizes and optimizers in language modeling. We refer to Figures I.11 to I.14 for the results that report update magnitude when training 160M model and training dynamics across all model sizes.

## 5 Experiments

We now turn to the empirical evaluation of the proposed algorithms against several benchmarks. The detailed experiment setup, including the choice of hyperparameters as well as additional experimental results and details, can be found in Appendix I. The best performance of algorithms is reported in Tables 7 (momentum-based algorithms), 8 (algorithms with momentum and component-wise step-size), and 9 (algorithms with component-wise step-size). For clarity and quick reference, all links to the paper's empirical results are summarized in Table 6, while Appendix I provides additional details about the training and tokenization pipeline.

**Comparison on Standard Benchmarks.** First, we test the performance of NGN-M against other methods that use momentum, such as SGDM, Momo, MomSPS, ALR-SMAG, and NGN. The tests include the training of Resnet20 [29] and ViT [17] on the CIFAR10 dataset [45], and Resnet110 on CIFAR100. Second, we test the performance of NGN-MD against Adam and Momo-Adam that – contrary to NGN-M – both use component-wise preconditioning. All experiments in this section do not use LR schedulers or weight decay.

From Tables 7 and 8 we observe that NGN-M and NGN-MDv1 exhibit competitive performance across all settings we tested, matching the best performance of other algorithms. Importantly, NGN-M and NGN-MDv1 demonstrate significantly greater robustness to the choice of the LR hyperparameter. Indeed, Figure 2 shows that the range of LR hyperparameter that allows NGN-M and NGN-MDv1 to perform optimally is much wider: We can, for instance, use step-sizes that are 1-2 orders of magnitude larger than the optimal one without a significant drop in performance. This is particularly evident when training ResNet20 and ViT models. Besides, we clearly observe that momentum consistently improves the stability of NGN across all settings. A similar trend can be observed when considering the LR sensitivity metric: see Table 2. We refer to Appendix I for additional ablation studies against other optimizers and results when training small-scale NLP models.

**Vision Experiments on ImageNet.** Now we switch to larger tasks and datasets. We train a ResNet18 on ImageNet1k [16]. This represents the first task in which we pair our proposed algorithms with LR schedule. As illustrated in Figure 3, NGN-M and NGN-MDv1 achieve the highest test accuracy, while exhibiting higher robustness across larger LR, improving over both NGN and Momo. Among adaptive methods, NGN-MDv1 compares favorably against Adam and Momo-Adam, while once again

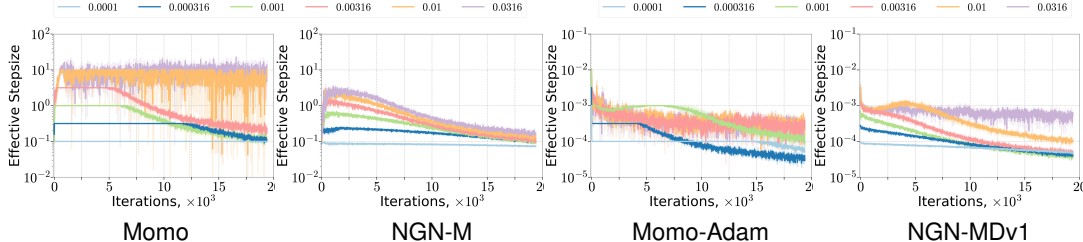

Figure 5: The step-size of Momo, NGN-M (**two left**), Momo-Adam and NGN-MDv1 (**two right**) during the training of ViT on CIFAR10. We demonstrate the step-sizes $\tau_k$ for Momo and Momo-Adam and $\gamma_k$ for NGN-M and NGN-MDv1 varying step-size hyperparameters $\alpha_0$ and $c$ of the algorithms (indicated in the legend). We refer to Figures I.9 and I.10 for the results in training Resnet20.

achieving higher performance on a wider range of LR (Table 8). Appendix I.4 reports additional ablations on ImageNet32 and train loss stability results.

Next, we test the effectiveness of the algorithms when training ViT-Tiny on ImageNet1k. This model is trained for a longer horizon, making it notoriously sensitive to the initial LR and requiring an adaptive step-size. We follow the protocol of Schaipp et al. [79]. As highlighted in Figure 3 and Table 8, NGN-MDv1 achieves the highest test accuracy across adaptive methods. Moreover, at a larger LR, Adam diverges, whereas both Momo-Adam and NGN-MDv1 maintain more stable training dynamics. The LR sensitivity metric reported in Table 3 supports our observations.

**Language Modeling.** Pre-training Large Language Models represents a challenging optimization task. To achieve competitive performance, optimizers with adaptive step-size are needed, and preventing instabilities in low-precision training often requires careful hyperparameter tuning. To evaluate the capability of NGN-MDv1 in this setting, we train decoder-only transformers [74] with 70M, 160M, 410M, and 1B parameters around Chinchilla optimum [33] on SlimPajama-627B [86]. For each model, we retune the LR, using 3 seeds for the first three models and 1 seed for the 1B.

As reported in Figure 4 and Table 8, we note that NGN-MDv1 matches the performance of Adam across all model sizes. However, NGN-MDv1 achieves competitive performance even for LR hyperparameter $c = 10^{-2}$ while Adam's performance drops significantly. This phenomenon is consistent across all scales we tested, suggesting that the optimal LR of NGN-MDv1 is shifted towards larger values, but also that the algorithm is less sensitive to such a hyperparameter. We additionally discuss how to introduce weight decay in NGN-MDv1 and report additional ablations on its role in this training task in Appendix G. Moreover, we report the ablation studies when varying the momentum parameter in Appendix I.12, demonstrating the improved stability to momentum parameters.

**Effective Step-size of NGN-M and NGN-MDv1.** As shown in Figure 5, the effective LR of NGN-M and NGN-MDv1 is inherently adaptive: it rises sharply at the start, and then gradually decreases, resembling annealing schedules commonly used in practice. By contrast, the effective step-size of Momo and Momo-Adam is largely fixed for large $\alpha_0$, effectively reducing them to SGDM and Adam and limiting their resilience. Evidence across ResNet20 training (Figures I.6, I.7, I.9 and I.10) and large-scale language modeling (Figures I.11 to I.13) shows that the NGN step-size is more conservative, automatically decreasing when needed to stabilize training—even for large $c$. This adaptivity underlies the LR robustness of NGN-M and NGN-MDv1.

Table 4: Train time of Adam and NGN-MDv1 when training language models.

| Model | Method | Time per Iteration (sec) | Time per Optimizer Update (sec) |
|---|---|---|---|
| 70M | AdamW | $1.63_{\pm 0.01}$ | $0.0048_{\pm 0.0002}$ |
|  | NGN-MDv1 | $1.65_{\pm 0.01}$ | $0.0130_{\pm 0.0002}$ |
| 160M | AdamW | $3.33_{\pm 0.03}$ | $0.0088_{\pm 0.0003}$ |
|  | NGN-MDv1 | $3.37_{\pm 0.02}$ | $0.0239_{\pm 0.0003}$ |
| 410M | AdamW | $8.41_{\pm 0.06}$ | $0.0838_{\pm 0.0009}$ |
|  | NGN-MDv1 | $8.68_{\pm 0.06}$ | $0.2154_{\pm 0.0007}$ |

**Computation Cost of NGN-MD.** Implementing NGN-MDv1 can be slightly more expensive, but the overall cost is modest as we show next. First, we emphasize that the implementation does not even require additional matrix–vector products since the preconditioner is diagonal; the only extra work is computing $\|\nabla f(x_k)\|^2_{\mathbf{D}_k^{-1}}$, which amounts to an additional pass over the gradient. This can also be incorporated into the update of $\mathbf{D}_k$, avoiding extra matrix operations. In practice, our naive implementation is about $2.5\times$ slower per update than PyTorch's AdamW (see Table 4) due to the need for two passes over the gradient. However, since forward/backward computations dominate runtime, the overall training speed remains largely comparable. Since our focus is on demonstrating the stability benefits of the NGN step-size, we leave efficiency improvements to future work. The extended discussion is reported in Appendix F.2.

## 6 Conclusion and Future Work

This work introduced several novel adaptations of the NGN step-size method, incorporating support for momentum and/or diagonal step-size. We provided a theoretical analysis of the convergence rates for these algorithms and conducted an extensive empirical evaluation of their performance. The experimental results show that combining momentum with the NGN step-size yields high robustness to step-size hyperparameter choices and performs competitively with state-of-the-art algorithms across various settings.

Given the significant complexity of the task, we defer the theoretical explanation of the step-size resilience properties of NGN-M for large values of $\beta$ and analysis in the non-convex setting to future work, including classes of structured non-convex functions such as PL [73], $\alpha$-$\beta$-condition [34], or Aiming [52]. It would also be worthwhile to study NGN-M under weaker smoothness assumptions [104, 1]. Furthermore, while the two proposed methods for incorporating weight decay into NGN-MDv1 outperform AdamW in training language models, they still exhibit some sensitivity to the step-size hyperparameter. This may, in part, be due to the limited understanding of the expected effects of the weight decay technique, a topic that requires further investigation. We acknowledge that computing NGN step-size at a large scale may cause runtime overhead, and discuss this limitation in Appendix F.2. We also recognize that integrating NGN-MDv1 with advanced parallelism schemes, such as Tensor Parallelism [83] or ZeRO-2 [75], introduces additional compute and communication overhead, and will require further adaptation of the algorithm. Nevertheless, our results provide valuable guidance for developing inherently more stable optimizers. As a next step, it would be fascinating to investigate whether the resilience of emerging methods like Muon [40] can be further improved by incorporating the NGN step-size.

## Acknowledgement

Rustem Islamov and Aurelien Lucchi acknowledge the financial support of the Swiss National Foundation, SNF grant No 207392. Antonio Orvieto acknowledges the financial support of the Hector Foundation.

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

# Appendix

## Contents

## A    Equivalent Formulations of NGN-M

We remind that the iterates of NGN-M are the following

$$x^{k+1} = x^k - (1-\beta)\gamma_k \nabla f_{S_k}(x^k) + \beta(x^k - x^{k-1})$$
$$= x^k - (1-\beta)\frac{c}{1 + \frac{c}{2f_{S_k}(x^k)}\|\nabla f_{S_k}(x^k)\|^2}\nabla f_{S_k}(x^k) + \beta(x^k - x^{k-1}).$$

We can rewrite the update rule using Iterative-Moving Average (IMA) approach presented in Proposition 1.6, Sebbouh et al. [81].

**Lemma A.1** (Proposition C.8 [63], Lemma 7.3 in [22])**.** *The iterates* $\{x^k\}$ *generated by* NGN-M *are equivalent to the sequence* $\{x^k\}$ *generated by IMA update*

$$z^{k+1} = z^k - \gamma_k \nabla f_{S_k}(x^k), \quad x^{k+1} = \frac{\lambda}{1+\lambda}x^k + \frac{1}{1+\lambda}z^{k+1}, \tag{5}$$

*where*

$$\beta = \frac{\lambda}{1+\lambda}, \qquad z^{k+1} = x^{k+1} + \lambda(x^{k+1} - x^k), \quad \text{and} \quad x^{-1} = z^0 = x^0. \tag{6}$$

*Proof.* Let the sequences $\{x^k\}$ and $\{z^k\}$ be defined according to Equation (5). Let $\beta$ be defined as $\frac{\lambda}{1+\lambda}$. Then we have

$$x^{k+1} = \frac{\lambda}{1+\lambda}x^k + \frac{1}{1+\lambda}z^{k+1}$$
$$= \frac{\lambda}{1+\lambda}x^k + \frac{1}{1+\lambda}(z^k - \gamma_k \nabla f_{S_k}(x^k))$$
$$= \frac{\lambda}{1+\lambda}x^k + \frac{1}{1+\lambda}((1+\lambda)x^k - \lambda x^{k-1} - \gamma_k \nabla f_{S_k}(x^k))$$
$$= x^k - \frac{1}{1+\lambda}\gamma_k \nabla f_{S_k}(x^k) + \frac{\lambda}{1+\lambda}(x^k - x^{k-1}).$$

It remains to use (6) as we have $\beta = \frac{\lambda}{1+\lambda}$ and $1 - \beta = 1 - \frac{\lambda}{1+\lambda} = \frac{1}{1+\lambda}$. $\qquad \square$

## B    Technical Lemmas and Definitions

**Definition B.1.** We say that the function $\phi$ admits **L**-smooth with parameters $\mathbf{L} := (L_1, \ldots, L_d), L_j \geq 0 \,\forall j \in [d]$, if the following inequality holds for all $x, h \in \mathbb{R}^d$

$$\phi(x+h) \leq \phi(x) + \langle \nabla\phi(x), h \rangle + \tfrac{1}{2}h^\top \mathbf{L}h. \tag{7}$$

*Remark* B.2. If we set for all $j \in [d]$ $L_j := L$ then Definition B.1 reduces to standard $L$-smoothness.

This assumption is typically used in the context of coordinate adaptive algorithms such as SignSGD [4, 78].

**Definition B.3.** The function $\phi: \mathbb{R}^d \to \mathbb{R}$ satisfies *PŁ-condition* with constant $\mu > 0$ if for all $x, y \in \mathbb{R}^d$ we have

$$\|\nabla f(x)\|^2 \geq 2\mu(f(x) - f^*). \tag{8}$$

**Assumption B.4.** We assume that the coordinate-wise variance of the stochastic estimator is bounded, i.e. for all $x \in \mathbb{R}^d$ and $j \in [d]$ we have

$$\mathbb{E}_S\left[|(\nabla_j f_S(x) - \nabla_j f(x)|^2\right] \leq \sigma_j^2. \tag{9}$$

**Lemma B.5** (Lemma 4.9 from [66]). *Let each $f_i$ be $L$-smooth for all $i$, then the step-size of NGN satisfies*

$$\gamma_k \in \left[\frac{c}{1+cL}, c\right]. \tag{10}$$

**Lemma B.6** (Lemma 4.2 from [66]). *Let each $f_i$ be $L$-smooth for all $i$, then the iterates of NGN satisfy*

$$\gamma_k^2\|\nabla f_{S_k}(x^k)\|^2 \leq \frac{4cL}{1+2cL}\gamma_k(f_{S_k}(x^k) - f_{S_k}^*) + \frac{2c^2L}{1+cL}\max\left\{\frac{2cL-1}{2cL+1}, 0\right\}f_{S_k}^*. \tag{11}$$

**Lemma B.7** (Gradient Upper Bound). *Let $\phi\colon \mathbb{R}^d \to \mathbb{R}$ satisfy Definition B.1. Then, for all $x \in \mathbb{R}^d$ and all $j \in [d]$ we have*

$$2L_j(f(x) - f^*) \geq (\nabla_j f(x))^2. \tag{12}$$

*Proof.* From Definition B.1 we have

$$f^* = \min_{x\in\mathbb{R}^d} f(x) \leq \min_{h_j\in\mathbb{R}} f(x + h_j e_j) \leq f(x) + \min_{h_j\in\mathbb{R}}\left[\nabla_j f(x)h_j + \frac{L_j}{2}h_j^2\right].$$

Now we can explicitly compute the minimum in the right-hand side. The optimal value is achieved at

$$h_j^* := -\frac{1}{L_j}\nabla_j f(x),$$

therefore,

$$
\begin{aligned}
f^* &\leq& f(x) + \nabla_j f(x)h_j^* + \frac{L_j}{2}(h_j^*)^2 \\
&=& f(x) - \frac{1}{L_j}(\nabla_j f(x))^2 + \frac{1}{2L_j}(\nabla_j f(x))^2 \\
&=& f(x) - \frac{1}{2L_j}(\nabla_j f(x))^2,
\end{aligned}
$$

which is equivalent to the statement of the lemma. $\square$

## C   Convergence of NGN-D

First, we provide NGN-D pseudocode and the main convergence results.

---
**Algorithm 3** NGN-D
---
1: **Input:** $x^0 \in \mathbb{R}^d$, step-size parameter $c > 0$
2: **for** $k = 0, 1, \ldots, K-1$ **do**
3:     Sample a batch $S_k \subseteq [n]$ and compute $f_{S_k}$ and $\nabla f_{S_k}(x^k)$
4:     Compute $\gamma_k^{(j)} = \frac{c}{1 + \frac{c}{2f_{S_k}(x^k)}(\nabla_j f_{S_k}(x^k))^2}$
5:     Update
$$x_{(j)}^{k+1} = x_{(j)}^k - \gamma_k^{(j)}\nabla_j f_{S_k}(x^k)$$
6: **end for**

---

**Theorem C.1.** *Let each $f_i$ satisfies Definition B.1. Assume that Assumption B.4 holds. Then the iterates of NGN-D (Algorithm 3) with step-size parameters $\{c_j\}_{j=1}^d$ such that $c_j \leq 1/2L_j$ satisfy*

$$\min_{0\leq k<K}\mathbb{E}\left[\|\nabla f(x^k)\|^2\right] \leq \frac{12(f(x^0) - f^*)}{c_{\min}K} + \frac{1}{c_{\min}}\sum_{j=1}^d 18L_jc_j^2\sigma_j^2, \tag{13}$$

*where $c_{\min} := \min_{j\in[d]} c_j$. Moreover, if $c_j = \mathcal{O}(\varepsilon^2)$ for all $j \in [d]$ then after $K = \mathcal{O}(\varepsilon^{-4})$ we obtain $\min_{0\leq k<K}\mathbb{E}\left[\|\nabla f(x^k)\|^2\right] \leq \mathcal{O}(\varepsilon^2)$.*

NGN-D converges with classic rate $\mathcal{O}(1/\sqrt{K})$ similar to Adagrad [94]. We highlight that, to the best of our knowledge, NGN-D is the first algorithm that uses diagonal Polyak-type stepsize and converges under standard smoothness and bounded variance assumptions without requirements of bounded gradients and interpolation.

**Theorem C.2.** *Let $f$ satisfies PŁ-condition and each $f_i$ satisfies Definition B.1. Assume that Assumption B.4 holds. Then the iterates of NGN-D (Algorithm 3) with step-size parameters $\{c_j\}_{j=1}^d$ such that $c_j \leq \min\{1/2L_j, 6/\mu\}$ satisfy*

$$\mathbb{E}\left[f(x^K) - f^*\right] \leq (1 - \mu c_{\min}/6)^K (f(x^0) - f^*) + \frac{9}{\mu c_{\min}} \sum_{j=1}^d L_j c_j^2 \sigma_j^2, \tag{14}$$

*where $c_{\min} := \min_{j \in [d]} c_j$. Moreover, if $c_j = \mathcal{O}(\varepsilon)$ for all $j \in [d]$ then after $K = \max\{\mathcal{O}(\varepsilon^{-1}), \mathcal{O}(1)\} \log \varepsilon^{-1}$ iterations we obtain $\mathbb{E}\left[f(x^K) - f^*\right] \leq \mathcal{O}(\varepsilon)$.*

To the best of our knowledge, this is the first result of the convergence of the Polyak-like step-size algorithm under the PŁ-condition. The convergence guarantees are similar to that of SGD [22].

Now we are ready to derive the step-size bounds.

**Lemma C.3** (Step-size Bounds). *Let $f_{S_k}(x): \mathbb{R}^d \to \mathbb{R}$ be a stochastic loss of batch $S_k$ at iteration $k$. Let $f_{S_k}(x)$ satisfy Definition (B.1). Consider $\gamma_j^k$ as in NGN-D (Algorithm 3), then we have*

$$\gamma_j^k \in \left[\frac{c_j}{1 + c_j L_j}, c_j\right]. \tag{15}$$

*Proof.* From Lemma B.7 we have $2L_j(f_{S_k}(x^k) - f_{S_k}^*) \geq (\nabla_j f_{S_k}(x^k))^2$. Since we assume that each $f_{S_k}^* \geq 0$, then $2L_j f_{S_k}(x^k) \geq (\nabla_j f_{S_k}(x^k))^2$, or equivalently,

$$0 \leq \frac{(\nabla_j f_{S_k}(x))^2}{2f_{S_k}(x)} \leq L_j.$$

Therefore, for all $j \in [d]$ we have

$$\gamma_j^k = \frac{c_j}{1 + \frac{c_j}{2f_{S_k}(x^k)}(\nabla_j f_{S_k}(x^k))^2} \leq \frac{c_j}{1} = c_j,$$

and

$$\gamma_j^k = \frac{c_j}{1 + \frac{c_j}{2f_{S_k}(x^k)}(\nabla_j f_{S_k}(x^k))^2} \geq \frac{c_j}{1 + c_j L_j},$$

which concludes the proof. $\square$

**Lemma C.4** (Fundamental Equality). *Consider $\gamma_j^k$ as in NGN-D (Algorithm 3). Then the following equality holds*

$$\gamma_j^k (\nabla_j f_{S_k}(x^k))^2 = 2\left(\frac{c_j - \gamma_j^k}{c_j}\right) f_{S_k}(x^k). \tag{16}$$

*Proof.* From NGN-D (Algorithm 3) we have

$$\left(1 + \frac{c_j}{2f_{S_k}(x^k)}(\nabla_j f_{S_k}(x^k))^2\right)\gamma_j^k = c_j,$$

which one can rewrite as

$$\frac{c_j}{2f_{S_k}(x^k)}(\nabla_j f_{S_k}(x^k))^2 \gamma_j^k = c_j - \gamma_j^k.$$

It is left to divide both sides by $\frac{2f_{S_k}(x^k)}{c_j}$. $\square$

## C.1 Convergence in General Non-convex Setting

**Theorem C.1.** *Let each $f_i$ satisfies Definition B.1. Assume that Assumption B.4 holds. Then the iterates of* NGN-D *(Algorithm 3) with step-size parameters $\{c_j\}_{j=1}^d$ such that $c_j \leq 1/2L_j$ satisfy*

$$\min_{0 \leq k < K} \mathbb{E}\left[\|\nabla f(x^k)\|^2\right] \leq \frac{12(f(x^0) - f^*)}{c_{\min}K} + \frac{1}{c_{\min}} \sum_{j=1}^d 18L_j c_j^2 \sigma_j^2, \tag{13}$$

*where $c_{\min} := \min_{j \in [d]} c_j$. Moreover, if $c_j = \mathcal{O}(\varepsilon^2)$ for all $j \in [d]$ then after $K = \mathcal{O}(\varepsilon^{-4})$ we obtain $\min_{0 \leq k < K} \mathbb{E}\left[\|\nabla f(x^k)\|^2\right] \leq \mathcal{O}(\varepsilon^2)$.*

*Proof.* First, we write separable Definition B.1

$$
\begin{aligned}
f(x^{k+1}) - f(x^k) &= f\left(x^k - \sum_{j=1}^d \gamma_j^k \nabla_j f_{S_k}(x^k) e_j\right) - f(x^k) \\
&\leq -\sum_{j=1}^d \nabla_j f(x^k) \cdot \gamma_j^k \nabla_j f_{S_k}(x^k) + \frac{1}{2} \sum_{j=1}^d L_j (\gamma_j^k \nabla_j f_{S_k}(x^k))^2 \\
&\leq -\sum_{j=1}^d \nabla_j f(x^k) \cdot \gamma_j^k \nabla_j f_{S_k}(x^k) + \frac{1}{2} \sum_{j=1}^d L_j \sigma_j^2 (\nabla_j f_{S_k}(x^k))^2. \quad (17)
\end{aligned}
$$

Note that both $\gamma_j^k$ and $\nabla_j f_{S_k}(x^k)$ depend on the realization $S_k$, thus we can not directly apply conditional expectation with respect to $x^k$, as in this case we would have to analyze the product $\gamma_j^k \nabla_j f_{S_k}(x^k)$. Given bounds of the step-size $\gamma_j^k$ from Lemma C.3, we can write the step-size as follows

$$\gamma_j^k = \frac{c_j}{1 + c_j L_j} + \nu_j^k \frac{c_j^2 L_j}{1 + c_j L_j},$$

where $\nu_j^k \in [0, 1]$ is a random variable. Varying the value of $\nu_j^k$ from 0 to 1 we cover the whole range of $\gamma_j^k$. Thus, we continue as follows

$$
\begin{aligned}
&-\gamma_j^k \nabla_j f(x^k) \nabla_j f_{S_k}(x^k) \\
&= -\frac{c_j}{1 + c_j L_j} \nabla_j f(x^k) \nabla_j f_{S_k}(x^k) - \frac{c_j^2 L_j}{1 + c_j L_j} \nu_j^k \nabla_j f(x^k) \nabla_j f_{S_k}(x^k) \\
&\leq -\frac{c_j}{1 + c_j L_j} \nabla_j f(x^k) \nabla_j f_{S_k}(x^k) + \frac{c_j^2 L_j}{1 + c_j L_j} |\nu_j^k| \cdot |\nabla_j f(x^k) \nabla_j f_{S_k}(x^k)| \\
&\leq -\frac{c_j}{1 + c_j L_j} \nabla_j f(x^k) \nabla_j f_{S_k}(x^k) + \frac{c_j^2 L_j}{1 + c_j L_j} \cdot |\nabla_j f(x^k) \nabla_j f_{S_k}(x^k)|.
\end{aligned}
$$

Now we use the inequality $|ab| \leq \frac{1}{2}a^2 + \frac{1}{2}b^2 + \frac{1}{2}|a - b|^2$, and derive

$$
\begin{aligned}
2\mathbb{E}_k\left[|\nabla_j f(x^k) \nabla_j f_{S_k}(x^k)|\right] &\leq |\nabla_j f(x^k)|^2 + \mathbb{E}_k\left[|\nabla_j f_{S_k}(x^k)|^2\right] + \mathbb{E}_k\left[|\nabla_j f(x^k) - \nabla_j f_{S_k}(x^k)|^2\right] \\
&\leq 2|\nabla_j f(x^k)|^2 + 2\mathbb{E}_k\left[|\nabla_j f(x^k) - \nabla_j f_{S_k}(x^k)|^2\right] \\
&\leq 2|\nabla_j f(x^k)|^2 + 2\sigma_j^2.
\end{aligned}
$$

Therefore, we get

$$
\begin{aligned}
-\mathbb{E}_k\left[\gamma_j^k \nabla_j f(x^k) \nabla_j f_{S_k}(x^k)\right] &\leq -\frac{c_j}{1 + c_j L_j} |\nabla_j f(x^k)|^2 + \frac{c_j^2 L_j}{1 + c_j L_j}\left(|\nabla_j f(x^k)|^2 + \sigma_j^2\right) \\
&= -c_j\left(\frac{1 - c_j L_j}{1 + c_j L_j}\right)|\nabla_j f(x^k)|^2 + \frac{c_j^2 L_j}{1 + c_j L_j} \sigma_j^2. \quad (18)
\end{aligned}
$$

We plug in (18) into (17) and get

$$
\begin{aligned}
\mathbb{E}_k\left[f(x^{k+1})\right] - f(x^k) &\leq -\sum_{j=1}^{d}\left(\mathbb{E}_k\left[\gamma_j^k \nabla_j f(x^k)\nabla_j f_{S_k}(x^k)\right] + \frac{L_j c_j^2}{2}\mathbb{E}_k\left[|\nabla_j f_{S_k}(x^k)|^2\right]\right) \\
&\leq \sum_{j=1}^{d}\left(\left[-c_j\left(\frac{1-c_jL_j}{1+c_jL_j}\right) + \frac{L_j c_j^2}{2}\right]|\nabla_j f(x^k)|^2 \right. \\
&\qquad\qquad \left. + \left[\frac{c_j^2 L_j}{1+c_jL_j} + \frac{L_j c_j^2}{2}\right]\sigma_j^2\right).
\end{aligned}
$$

If $c_j \leq \frac{1}{2L_j}$, we get

$$
\mathbb{E}_k\left[f(x^{k+1})\right] - f(x^k) \leq \sum_{j=1}^{d}\left(-\frac{c_j}{12}|\nabla_j f(x^k)|^2 + \frac{3L_j c_j^2}{2}\sigma_j^2\right).
$$

$\square$

We continue as follows

$$
\mathbb{E}_k\left[f(x^{k+1})\right] - f(x^k) \leq -\frac{c_{\min}}{12}\|\nabla f(x^k)\|^2 + \sum_{j=1}^{d}\frac{3L_j c_j^2}{2}\sigma_j^2. \tag{19}
$$

Taking full expectation and unrolling the recursion above for all iterations $\{0,\ldots,K-1\}$. Thus, we obtain

$$
\min_{0\leq k<K}\mathbb{E}\left[\|\nabla f(x^k)\|^2\right] \leq \frac{1}{K}\sum_{k=0}^{K-1}\mathbb{E}\left[\|\nabla f(x^k)\|^2\right] \leq \frac{12}{c_{\min}K}(f(x^0)-f^*) + \frac{18}{c_{\min}}\sum_{j=1}^{d}L_j c_j^2\sigma_j^2.
$$

If we choose each $c_j = \frac{c_{0,j}}{\sqrt{K}}$ such that $c_{0,j} \leq \frac{1}{2L_j}$ we ensure that $c_j \leq \frac{1}{2L_j}$ as well. Plugging this step-size into the bound we get

$$
\begin{aligned}
\min_{0\leq k<K}\mathbb{E}\left[\|\nabla f(x^k)\|^2\right] &\leq \frac{12}{\frac{c_{0,\min}}{\sqrt{K}}K}(f(x^0)-f^*) + \frac{18}{\frac{c_{0,\min}}{\sqrt{K}}}\sum_{j=1}^{d}L_j\sigma_j^2\frac{c_{0,j}^2}{K} \\
&\leq \frac{12}{c_{0,\min}\sqrt{K}}(f(x^0)-f^*) + \frac{18}{c_{0,\min}\sqrt{K}}\sum_{j=1}^{d}L_j\sigma_j^2 c_{0,j}^2,
\end{aligned}
$$

where $c_{0,\min} := \min_{j\in[d]} c_{0,j}$. If we choose $K = \mathcal{O}(\varepsilon^{-4})$ we get that

$$
\min_{0\leq k<K}\mathbb{E}\left[\|\nabla f(x^k)\|^2\right] = \mathcal{O}(1/\sqrt{K}) = \mathcal{O}(\varepsilon^2).
$$

## C.2 Convergence under PŁ-condition

**Theorem C.2.** *Let $f$ satisfies PŁ-condition and each $f_i$ satisfies Definition B.1. Assume that Assumption B.4 holds. Then the iterates of NGN-D (Algorithm 3) with step-size parameters $\{c_j\}_{j=1}^{d}$ such that $c_j \leq \min\{1/2L_j, 6/\mu\}$ satisfy*

$$
\mathbb{E}\left[f(x^K)-f^*\right] \leq (1-\mu c_{\min}/6)^K(f(x^0)-f^*) + \frac{9}{\mu c_{\min}}\sum_{j=1}^{d}L_j c_j^2\sigma_j^2, \tag{14}
$$

*where $c_{\min} := \min_{j\in[d]} c_j$. Moreover, if $c_j = \mathcal{O}(\varepsilon)$ for all $j \in [d]$ then after $K = \max\{\mathcal{O}(\varepsilon^{-1}), \mathcal{O}(1)\}\log\varepsilon^{-1}$ iterations we obtain $\mathbb{E}\left[f(x^K)-f^*\right] \leq \mathcal{O}(\varepsilon).$*

*Proof.* We obtain (19) and use Definition B.3

$$\mathbb{E}_k \left[ f(x^{k+1}) \right] - f(x^k) \leq -\frac{c_{\min}}{12} \|\nabla f(x^k)\|^2 + \sum_{j=1}^{d} \frac{3L_j c_j^2}{2} \sigma_j^2$$

$$\leq -\frac{\mu c_{\min}}{6} (f(x^k) - f^*) + \sum_{j=1}^{d} \frac{3L_j c_j^2}{2} \sigma_j^2$$

Subtracting $f^*$ from both sides of the inequality above and taking full expectation we obtain

$$\mathbb{E} \left[ f(x^{k+1}) - f^* \right] \leq (1 - \mu c_{\min}/6)\mathbb{E} \left[ f(x^k) - f^* \right] + \sum_{j=1}^{d} \frac{3L_j c_j^2}{2} \sigma_j^2.$$

Unrolling the recursion above for $\{0, \ldots, K-1\}$ iterations we derive

$$\mathbb{E} \left[ f(x^K) - f^* \right] \leq (1 - \mu c_{\min}/6)^K (f(x^0) - f^*) + \frac{1}{c_{\min}} \sum_{j=1}^{d} \underbrace{\frac{9L_j \sigma_j^2}{\mu}}_{A_j} c_j^2.$$

Now we follow the proof of Lemma A.3 in Garrigos and Gower [22]. Let us choose $c_j = \min\{1/2L_j, \varepsilon/2dA_j\}$. Together with the choice of $K \geq \max_{j \in [d]} \max \left\{ \frac{1}{\varepsilon} \frac{12A_j}{\mu}, \frac{12L_j}{\mu} \right\} \log \frac{2(f(x^0)-f^*)}{\varepsilon}$ we get

$$(1 - \mu c_{\min}/6)^K (f(x^0) - f^*) \leq \frac{\varepsilon}{2}.$$

Now we have two cases:

1. $c_{\min}$ does not depend on $\varepsilon$, then we have

$$\frac{1}{c_{\min}} A_j c_j^2 \leq \mathcal{O}(\varepsilon^2).$$

2. $c_{\min}$ does depend on $\varepsilon$, i.e. $c_{\min} = \mathcal{O}(\varepsilon)$, then we have

$$\frac{1}{c_{\min}} A_j c_j^2 \leq \mathcal{O}(\varepsilon).$$

Therefore, combining all together we get

$$\mathbb{E} \left[ f(x^K) - f^* \right] \leq \mathcal{O}(\varepsilon)$$

after $K \geq \max_{j \in [d]} \max \left\{ \frac{1}{\varepsilon} \frac{12A_j}{\mu}, \frac{12L_j}{\mu} \right\} \log \frac{2(f(x^0)-f^*)}{\varepsilon}$ iterations.

$\square$

# D  Convergence of NGN-M

## D.1  Convergence of NGN-M in Stochastic Setting

**Theorem 4.3.** *Let Assumptions 4.1, 4.2 hold. Let the step-size hyperparameter $c > 0$ and the momentum parameter $\beta = \frac{\lambda}{1+\lambda}$ be constants where $\lambda \leq \min\{cL, 0.5(1+cL)^{-1}(1+2cL)^{-1}\}$. Then the iterates of* NGN-M *(Algorithm 1) satisfy*

$$\mathbb{E} \left[ f(\overline{x}^{K-1}) - f(x^*) \right] \leq \frac{\|x^0 - x^*\|^2 (1+2cL)^2}{cK} + 8cL(1+2cL)^2 \sigma_{\text{int}}^2 + 2cL \max \{2cL - 1, 0\} \sigma_{\text{pos}}^2,$$

*where $\overline{x}^{K-1}$ is chosen uniformly at random from $\{x^0, \ldots, x^{K-1}\}$. Moreover, if we set $c = \mathcal{O}(1/\sqrt{K})$ then we obtain $\mathbb{E} \left[ f(\overline{x}^{K-1}) - f(x^*) \right] \leq \mathcal{O}(1/\sqrt{K})$.*

*Remark* D.1. In fact, if $\lambda \le \frac{1}{(1+cL)(1+2cL)}$, then it implies that $\lambda \le \frac{1}{cL}$ because $\frac{1}{x} > \frac{1}{(1+x)(1+2x)}$ for any $x > 0$.

*Proof.* To prove the convergence of NGN-M we consider IMA formulation Equation (5):

$$x^{-1} = z^0 = x^0, \quad z^{k+1} = z^k - \gamma_k \nabla f_{S_k}(x^k), \quad x^{k+1} = \frac{\lambda}{1+\lambda}x^k + \frac{1}{1+\lambda}z^{k+1},$$

where $\beta = \frac{\lambda}{1+\lambda}, z^{k+1} = x^{k+1} + \lambda(x^{k+1} - x^k)$.

At iteration $k = 0$ we have

$$z^1 = z^0 - \gamma_0 \nabla f_{S_0}(x^0) = x^0 - \gamma_0 \nabla f_{S_0}(x^0).$$

Therefore, we get

$$
\begin{aligned}
\|z^1 - x^*\|^2 \quad &= \quad \|z^0 - x^*\|^2 - 2\gamma_0 \langle \nabla f_{S_0}(x^0), z^0 - x^* \rangle + \gamma_0^2 \|\nabla f_{S_0}(x^0)\|^2 \\
&\overset{\text{Lem. } B.6}{\le} \quad \|z^0 - x^*\|^2 - 2\gamma_0 \langle \nabla f_{S_0}(x^0), x^0 - x^* \rangle + \frac{4cL}{1+2cL}\gamma_0(f_{S_0}(x^0) - f_{S_0}^*) \\
&\qquad + \frac{2c^2L}{1+cL} \max\left\{\frac{2cL-1}{2cL+1}, 0\right\} f_{S_0}^*.
\end{aligned}
\tag{20}
$$

Let $\gamma_0 = \rho + \widetilde{\gamma}_0$ where $\rho = \frac{c}{(1+cL)(1+2cL)}$. Then we have

$$
\begin{aligned}
\widetilde{\gamma}_0 &= \gamma_0 - \rho \\
&\overset{\text{Lem. } B.5}{\le} c - \frac{c}{(1+cL)(1+2cL)} \\
&= c\frac{1+3cL+2c^2L^2-1}{(1+cL)(1+2cL)} \\
&= c^2L\frac{3+3cL}{(1+cL)(1+2cL)} \\
&= \frac{3c^2L}{1+2cL}.
\end{aligned}
$$

Using the above we continue from (20)

$$
\begin{aligned}
\|z^1 - x^*\|^2 &\overset{\text{conv.}}{\le} \|z^0 - x^*\|^2 - 2\gamma_0(f_{S_0}(x^0) - f_{S_0}(x^*)) + \frac{4cL}{1+2cL}\gamma_0(f_{S_0}(x^0) - f_{S_0}^*) \\
&\qquad + \frac{2c^2L}{1+cL} \max\left\{\frac{2cL-1}{2cL+1}, 0\right\} f_{S_0}^* \\
&\le \|z^0 - x^*\|^2 - 2\rho(f_{S_0}(x^0) - f_{S_0}(x^*)) - 2\widetilde{\gamma}_0(f_{S_0}(x^0) - f_{S_0}^*) + 2\widetilde{\gamma}_0(f_{S_0}(x^*) - f_{S_0}^*) \\
&\qquad + \frac{4cL}{1+2cL}\gamma_0(f_{S_0}(x^0) - f_{S_0}^*) + \frac{2c^2L}{1+cL} \max\left\{\frac{2cL-1}{2cL+1}, 0\right\} f_{S_0}^* \\
&= \|z^0 - x^*\|^2 - 2\rho(f_{S_0}(x^0) - f_{S_0}(x^*)) - 2\left(\gamma_0 - \rho - \frac{2cL}{1+2cL}\gamma_0\right)(f_{S_0}(x^0) - f_{S_0}^*) \\
&\qquad + 2\widetilde{\gamma}_0(f_{S_0}(x^*) - f_{S_0}^*) + \frac{2c^2L}{1+cL} \max\left\{\frac{2cL-1}{2cL+1}, 0\right\} f_{S_0}^*.
\end{aligned}
\tag{21}
$$

Here we have

$$
\begin{aligned}
\gamma_0 - \rho - \frac{2cL}{1+2cL}\gamma_0 &= \frac{1}{1+2cL}\gamma_0 - \rho \\
&= \frac{1}{1+2cL}\gamma_0 - \frac{c}{(1+cL)(1+2cL)} \\
&\overset{Lem.B.5}{\ge} \frac{1}{1+2cL}\frac{c}{1+cL} - \frac{c}{(1+cL)(1+2cL)} \\
&= 0,
\end{aligned}
$$

$\widetilde{\gamma}_0 \le \frac{3c^2L}{1+2cL}$, and $f_{S_0}(x^0) - f_{S_0}^* \ge 0$. Hence, we get

$$\|z^1 - x^*\|^2 \le \|z^0 - x^*\|^2 - 2\rho(f_{S_0}(x^0) - f_{S_0}(x^*)) + \frac{6c^2L}{1+2cL}(f_{S_0}(x^*) - f_{S_0}^*)$$
$$+ \frac{2c^2L}{1+cL}\max\left\{\frac{2cL-1}{2cL+1}, 0\right\}f_{S_0}^*.$$

Rearranging terms and taking expectation we get

$$2\rho\mathbb{E}\left[f(x^0) - f(x^*)\right] \le \mathbb{E}\left[\|z^1 - x^*\|^2\right] - \|z^0 - x^*\|^2 + \frac{6c^2L}{1+2cL}\sigma_{\text{int}}^2$$
$$+ \frac{2c^2L}{1+cL}\max\left\{\frac{2cL-1}{2cL+1}, 0\right\}\sigma_{\text{pos}}^2. \tag{22}$$

Next, for $k > 0$ we can use the relation $z^k = x^k + \lambda(x^k - x^{k-1})$. We expand $\|z^{k+1} - x^*\|^2$

$$\|z^{k+1} - x^*\|^2 \quad = \quad \|z^k - x^*\|^2 - 2\gamma_k\langle\nabla f_{S_k}(x^k), z^k - x^*\rangle + \gamma_k^2\|\nabla f_{S_k}(x^k)\|^2$$

$$\overset{\text{Lem. } A.1}{=} \quad \|z^k - x^*\|^2 - 2\gamma_k\langle\nabla f_{S_k}(x^k), x^k - x^*\rangle - 2\gamma_k\lambda\langle\nabla f_{S_k}(x^k), x^k - x^{k-1}\rangle$$
$$+ \gamma_k^2\|\nabla f_{S_k}(x^k)\|^2$$

$$\overset{\text{conv.}}{\le} \quad \|z^k - x^*\|^2 - 2\gamma_k(f_{S_k}(x^k) - f_{S_k}(x^*)) - 2\gamma_k\lambda(f_{S_k}(x^k) - f_{S_k}(x^{k-1}))$$
$$+ \gamma_k^2\|\nabla f_{S_k}(x^k)\|^2$$

$$\overset{\text{Lem. } B.6}{\le} \quad \|z^k - x^*\|^2 - 2\gamma_k(f_{S_k}(x^k) - f_{S_k}(x^*)) - 2\gamma_k\lambda(f_{S_k}(x^k) - f_{S_k}(x^{k-1}))$$
$$+ \frac{4cL}{1+2cL}\gamma_k(f_{S_k}(x^k) - f_{S_k}^*) + \frac{2c^2L}{1+cL}\max\left\{\frac{2cL-1}{2cL+1}, 0\right\}f_{S_k}^*.$$

Let $\gamma_k = \rho + \widetilde{\gamma}_k$, where $\rho, \widetilde{\gamma}_k \ge 0$, and $\rho$ is a constant step-size independent of $S_k$ which will be defined later. Therefore, we have

$$\|z^{k+1} - x^*\|^2 \quad \le \quad \|z^k - x^*\|^2 - 2\rho(f_{S_k}(x^k) - f_{S_k}(x^*)) - 2\widetilde{\gamma}_k(f_{S_k}(x^k) - f_{S_k}(x^*))$$
$$- 2\gamma_k\lambda_k(f_{S_k}(x^k) - f_{S_k}^*) + 2\gamma_k\lambda(f_{S_k}(x^{k-1}) - f_{S_k}^*)$$
$$+ \frac{4cL}{1+2cL}\gamma_k(f_{S_k}(x^k) - f_{S_k}^*) + \frac{2c^2L}{1+cL}\max\left\{\frac{2cL-1}{2cL+1}, 0\right\}f_{S_k}^*$$

$$= \quad \|z^k - x^*\|^2 - 2\rho(f_{S_k}(x^k) - f_{S_k}(x^*)) - 2\widetilde{\gamma}_k(f_{S_k}(x^k) - f_{S_k}^*) + 2\widetilde{\gamma}_k(f_{S_k}(x^*) - f_{S_k}^*)$$
$$- 2\gamma_k\lambda(f_{S_k}(x^k) - f_{S_k}^*) + 2\gamma_k\lambda(f_{S_k}(x^{k-1}) - f_{S_k}^*)$$
$$+ \frac{4cL}{1+2cL}\gamma_k(f_{S_k}(x^k) - f_{S_k}^*) + \frac{2c^2L}{1+cL}\max\left\{\frac{2cL-1}{2cL+1}, 0\right\}f_{S_k}^*$$

$$= \quad \|z^k - x^*\|^2 - 2\rho(f_{S_k}(x^k) - f_{S_k}(x^*)) - 2\left(\widetilde{\gamma}_k + \gamma_k\lambda - \frac{2cL}{1+2cL}\gamma_k\right)(f_{S_k}(x^k) - f_{S_k}^*)$$
$$+ 2\widetilde{\gamma}_k(f_{S_k}(x^*) - f_{S_k}^*) + 2\gamma_k\lambda(f_{S_k}(x^{k-1}) - f_{S_k}^*)$$
$$+ \frac{2c^2L}{1+cL}\max\left\{\frac{2cL-1}{2cL+1}, 0\right\}f_{S_k}^*. \tag{23}$$

We need to find $\rho$ such that

$$\widetilde{\gamma}_k + \gamma_k\lambda - \frac{2cL}{1+2cL}\gamma_k \ge 0$$

Since $\widetilde{\gamma}_k = \gamma_k - \rho$, then we have

$$\gamma_k - \rho + \gamma_k\lambda - \frac{2cL}{1+2cL}\gamma_k \ge 0$$
$$\Leftrightarrow \gamma_k\left(1 + \lambda - \frac{2cL}{1+2cL}\right) \ge \rho.$$

The inequality above is satisfied if it is satisfied for the lower bound on $\gamma_k$ (which is $c/1+cL$), i.e.

$$\frac{c}{1+cL}\left(\frac{1}{1+2cL}+\lambda\right)\geq\rho.$$

We can take $\rho=\frac{c}{(1+cL)(1+2cL)}$ since $\lambda\geq 0$.

$$\begin{aligned}
\widetilde{\gamma}_k &= \gamma_k - \rho\\
&\leq c - \frac{c}{(1+cL)(1+2cL)}\\
&= c\frac{1+3cL+2c^2L^2-1}{(1+cL)(1+2cL)}\\
&\leq c^2L\frac{3+3cL}{(1+cL)(1+2cL)}\\
&= \frac{3c^2L}{1+2cL}.
\end{aligned}$$

Using the above, we get from (23)

$$\begin{aligned}
\|z^{k+1}-x^*\|^2 &\leq \|z^k-x^*\|^2 - 2\rho(f_{S_k}(x^k)-f_{S_k}(x^*)) + 2c\lambda(f_{S_k}(x^{k-1})-f_{S_k}(x^*))\\
&\quad + 2c\lambda(f_{S_k}(x^*)-f_{S_k}^*) + \frac{6c^2L}{1+2cL}(f_{S_k}(x^*)-f_{S_k}^*)\\
&\quad + \frac{2c^2L}{1+cL}\max\left\{\frac{2cL-1}{2cL+1},0\right\}f_{S_k}^*.
\end{aligned}$$

Taking expectations we get

$$\begin{aligned}
\mathbb{E}\left[\|z^{k+1}-x^*\|^2\right] &\leq \mathbb{E}\left[\|z^k-x^*\|^2\right] - 2\rho\mathbb{E}\left[f(x^k)-f(x^*)\right] + 2c\lambda\mathbb{E}\left[f(x^{k-1})-f(x^*)\right]\\
&\quad + \left(2c\lambda+\frac{6c^2L}{1+2cL}\right)\sigma_{\text{int}}^2 + \frac{2c^2L}{1+cL}\max\left\{\frac{2cL-1}{2cL+1},0\right\}\sigma_{\text{pos}}^2. \quad (24)
\end{aligned}$$

Rearranging terms we get

$$\begin{aligned}
2\rho\mathbb{E}\left[f(x^k)-f(x^*)\right] - 2c\lambda\mathbb{E}\left[f(x^{k-1})-f(x^*)\right] &\leq \mathbb{E}\left[\|z^k-x^*\|^2\right] - \mathbb{E}\left[\|z^{k+1}-x^*\|^2\right]\\
&\quad + \left(2c\lambda+\frac{6c^2L}{1+2cL}\right)\sigma_{\text{int}}^2\\
&\quad + \frac{2c^2L}{1+cL}\max\left\{\frac{2cL-1}{2cL+1},0\right\}\sigma_{\text{pos}}^2. \quad (25)
\end{aligned}$$

Combining Equation (22) and Equation (25) for iterations $\{1,\ldots,K-1\}$ we get

$$\begin{aligned}
&2\rho\mathbb{E}\left[f(x^0)-f(x^*)\right] + 2\rho\sum_{k=1}^{K-1}\mathbb{E}\left[f(x^k)-f(x^*)\right] - 2c\lambda\sum_{k=1}^{K-1}\mathbb{E}\left[f(x^{k-1})-f(x^*)\right]\\
&= 2\rho\sum_{k=0}^{K-1}\mathbb{E}\left[f(x^k)-f(x^*)\right] - 2c\lambda\sum_{k=0}^{K-2}\mathbb{E}\left[f(x^k)-f(x^*)\right]\\
&\leq (2\rho-2c\lambda)\sum_{k=0}^{K-1}\mathbb{E}\left[f(x^k)-f(x^*)\right]\\
&\leq \|z^0-x^*\|^2 + \frac{6c^2L}{1+2cL}\sigma_{\text{int}}^2 + \frac{2c^2L}{1+cL}\max\left\{\frac{2cL-1}{2cL+1},0\right\}\sigma_{\text{pos}}^2\\
&\quad + \left(2c\lambda+\frac{6c^2L}{1+2cL}\right)(K-1)\sigma_{\text{int}}^2 + (K-1)\cdot\frac{2c^2L}{1+cL}\max\left\{\frac{2cL-1}{2cL+1},0\right\}\sigma_{\text{pos}}^2\\
&\leq \|z^0-x^*\|^2 + \left(2c\lambda+\frac{6c^2L}{1+2cL}\right)K\sigma_{\text{int}}^2 + K\cdot\frac{2c^2L}{1+cL}\max\left\{\frac{2cL-1}{2cL+1},0\right\}\sigma_{\text{pos}}^2. \quad (26)
\end{aligned}$$

We need to ensure that $\rho - c\lambda > 0$ which is satisfied for $\lambda$ such that

$$\frac{\rho}{2} = \frac{c}{2(1+cL)(1+2cL)} > c\lambda$$
$$\Leftrightarrow 1 > 2\lambda(1+cL)(1+2cL).$$

Note that we also assume that $\lambda \leq cL$. Therefore, from (26) we get

$$\frac{1}{K}\sum_{k=0}^{K-1}\mathbb{E}\left[f(x^k) - f(x^*)\right] \leq \frac{\|z^0 - x^*\|^2}{2(\rho - c\lambda)K} + \frac{1}{2(\rho - c\lambda)}\left(2c\lambda + \frac{6c^2 L}{1+2cL}\right)\sigma_{\text{int}}^2$$
$$+ \frac{1}{2(\rho - c\lambda)}\frac{2c^2 L}{1+cL}\max\left\{\frac{2cL-1}{2cL+1}, 0\right\}\sigma_{\text{pos}}^2$$
$$\leq \frac{\|z^0 - x^*\|^2}{2(\rho - c\lambda)K} + \frac{8c^2 L}{2(\rho - c\lambda)}\sigma_{\text{int}}^2$$
$$+ \frac{1}{2(\rho - c\lambda)}\frac{2c^2 L}{1+cL}\max\left\{\frac{2cL-1}{2cL+1}, 0\right\}\sigma_{\text{pos}}^2. \tag{27}$$

Since $\rho - c\lambda \geq \frac{\rho}{2}$ and setting $\overline{x}^k$ be uniformly at random chosen from $\{x^0, \ldots, x^{K-1}\}$ we get

$$\mathbb{E}\left[f(\overline{x}^k) - f(x^*)\right] \leq \frac{\|z^0 - x^*\|^2}{\rho K} + \frac{8c^2 L}{\rho}\sigma_{\text{int}}^2 + \frac{1}{\rho}\frac{2c^2 L}{1+cL}\max\left\{\frac{2cL-1}{2cL+1}, 0\right\}\sigma_{\text{pos}}^2, \tag{28}$$

where we use the convexity of $f$ and Jensen's inequality. Plugging the value of $\rho = \frac{c}{(1+cL)(1+2cL)}$ inside we get

$$\mathbb{E}\left[f(\overline{x}^k) - f(x^*)\right] \leq \frac{\|z^0 - x^*\|^2}{cK}(1+cL)(1+2cL) + 8cL(1+cL)(1+2cL)\sigma_{\text{int}}^2$$
$$+ 2cL\max\{2cL-1, 0\}\sigma_{\text{pos}}^2. \tag{29}$$

Choosing $c = \mathcal{O}(1/\sqrt{K})$ we get

$$\mathbb{E}\left[f(\overline{x}^k) - f(x^*)\right] \leq \mathcal{O}\left(\frac{\|z^0 - x^*\|^2}{\sqrt{K}} + \frac{\sigma_{\text{int}}^2}{\sqrt{K}} + \frac{\sigma_{\text{pos}}^2}{\sqrt{K}}\max\{2cL-1, 0\}\right). \tag{30}$$

Therefore, if $K \geq \mathcal{O}(\varepsilon^{-2})$ then $\mathbb{E}\left[f(\overline{x}^k) - f(x^*)\right] \leq \mathcal{O}(\varepsilon)$. It remains to notice that $z^0 = x^0$ to derive the statement of the theorem. $\square$

### D.2 Convergence of NGN-M under Interpolation

In this section, we show that NGN-M provably converges for large momentum values $\beta$–including the default $\beta = 0.9$—provided the LR hyperparameter $c$ is chosen sufficiently small. This is natural: if the step-size is too large, the momentum term accumulates excessive past error, which the algorithm may be unable to correct, potentially causing divergence. In short, convergence can be ensured in two complementary ways: $(i)$ use a small momentum parameter while allowing any LR hyperparameter $c$ (Theorem 4.3); or $(ii)$ restrict the LR hyperparameter $c$ while permitting a large (near-1) momentum parameter $\beta$ (Theorem D.2).

**Theorem D.2.** *Let Assumption 4.1 hold. Assume that there exists a minimizer $x^*$ shared across all functions $f_i$, i.e. $f_i^* = f(x^*)$ for all $i \in [n]$. Let NGN-M is run with momentum parameter $\beta = \frac{\lambda}{1+\lambda}$, $\lambda \in (0, \Lambda], \Lambda \geq 9$, step-size $c \leq \min\left\{\frac{1}{4\Lambda L}, \frac{1}{2L}\right\}$. Then the iterates of NGN-M satisfy*

$$\frac{1}{K}\sum_{k=0}^{K-1}\mathbb{E}\left[f(x^k) - f(x^*)\right] \leq \frac{3(\|x^0 - x^*\|^2 + 1/2L(f(x^0) - f(x^*)))}{cK}.$$

*Remark* D.3. Note that the requirement $\Lambda \geq 9$ allows setting the momentum parameter $\beta = 0.9$, which is a default choice in practice.

*Proof.* The proof is similar to Theorem 4.3, but we take into account the additional interpolation assumption. We start with

$$\|z^{k+1} - x^*\|^2 \quad = \quad \|z^k - x^*\|^2 - 2\gamma_k\langle\nabla f_{S_k}(x^k), z^k - x^*\rangle + \gamma_k^2\|\nabla f_{S_k}(x^k)\|^2$$

$$\overset{\text{Lem. } A.1}{=} \quad \|z^k - x^*\|^2 - 2\gamma_k\langle\nabla f_{S_k}(x^k), x^k + \lambda(x^k - x^{k-1}) - x^*\rangle$$
$$+ \gamma_k^2\|\nabla f_{S_k}(x^k)\|^2$$

$$= \quad \|z^k - x^*\|^2 - 2\gamma_k\langle\nabla f_{S_k}(x^k), x^k - x^*\rangle - 2\gamma_k\lambda\langle\nabla f_{S_k}(x^k), x^k - x^{k-1}\rangle$$
$$+ \gamma_k^2\|\nabla f_{S_k}(x^k)\|^2$$

$$\overset{\text{Asm. } 4.1}{\leq} \quad \|z^k - x^*\|^2 - 2\gamma_k(f_{S_k}(x^k) - f_{S_k}(x^*)) - 2\gamma_k\lambda(f_{S_k}(x^k) - f_{S_k}(x^{k-1}))$$
$$+ 2L\gamma_k^2(f_{S_k}(x^k) - f_{S_k}^*)$$

$$\overset{\text{Interp.}}{=} \quad \|z^k - x^*\|^2 - 2\gamma_k(f_{S_k}(x^k) - f_{S_k}(x^*)) - 2\gamma_k\lambda(f_{S_k}(x^k) - f_{S_k}(x^{k-1}))$$
$$+ 2L\gamma_k^2(f_{S_k}(x^k) - f_{S_k}(x^*))$$

$$= \quad \|z^k - x^*\|^2 - 2\gamma_k(1 + \lambda_k - L\gamma_k)(f(x^k) - f(x^*))$$
$$+ 2\gamma_k\lambda_k(f(x^{k-1}) - f(x^*)). \quad (31)$$

Since $f_{S_k}(x^k) - f_{S_k}(x^*) \geq 0$, with the choice $c \leq \frac{1}{2L}$, we have

$$1 + \lambda - L\gamma_k \geq 1 + \lambda - Lc \geq \lambda + \frac{1}{2}.$$

Moreover, by Lemma B.5 we have $\gamma_k \geq \frac{c}{1+cL}$ and $\gamma_k \leq c$. Thus, we continue the bound (31) on the distance $\|z^{k+1} - x^*\|^2$ as follows

$$\|z^{k+1} - x^*\|^2 \leq \|z^k - x^*\|^2 - \frac{2c}{1+cL}(1/2 + \lambda)(f_{S_k}(x^k) - f_{S_k}(x^*))$$
$$+ 2c\lambda(f_{S_k}(x^{k-1}) - f_{S_k}(x^*)).$$

Taking the expectation, we obtain the final bound on the distance

$$\mathbb{E}\left[\|z^{k+1} - x^*\|^2\right] \leq \mathbb{E}\left[\|z^k - x^*\|^2\right] - \frac{2c}{1+cL}(1/2 + \lambda)\mathbb{E}\left[f(x^k) - f(x^*)\right]$$
$$+ 2c\lambda\mathbb{E}\left[f(x^{k-1}) - f(x^*)\right]. \quad (32)$$

Note that for $k = 0$, the bound is simpler, since the momentum term is zero

$$\mathbb{E}\left[\|z^1 - x^*\|^2\right] \leq \|z^0 - x^*\|^2 - \frac{c}{1+cL}(f(x^0) - f(x^*))$$
$$= \|x^0 - x^*\|^2 - \frac{c}{1+cL}(f(x^0) - f(x^*)), \quad (33)$$

where we use $z^0 = x^0$. Summing the bounds (32) and (33) for iterations $k \in \{0, \ldots, K-1\}$, we obtain

$$\frac{c}{1+cL}(f(x^0) - f(x^*)) + \frac{2c(1/2 + \lambda)}{1+cL}\sum_{k=1}^{K-1}\mathbb{E}\left[f(x^k) - f(x^*)\right] - \sum_{k=2}^{K-1}2c\lambda\mathbb{E}\left[f(x^{k-1}) - f(x^*)\right]$$

$$= \frac{c}{1+cL}(f(x^0) - f(x^*)) + \frac{2c(1/2 + \lambda)}{1+cL}\mathbb{E}\left[f(x^{K-1}) - f(x^*)\right]$$

$$+ 2c\left(\frac{1/2 + \lambda}{1+cL} - \lambda\right)\sum_{k=1}^{K-2}\mathbb{E}\left[f(x^k) - f(x^*)\right]$$

$$\leq \sum_{k=0}^{K-1}\left(\mathbb{E}\left[\|z^k - x^*\|^2\right] - \mathbb{E}\left[\|z^{k+1} - x^*\|^2\right]\right) + 2c\lambda(f(x^0) - f(x^*))$$

$$\leq \|x^0 - x^*\|^2 + 2c\lambda(f(x^0) - f(x^*)).$$

Note that

$$\frac{1/4 + \lambda}{1 + cL} \geq \lambda \Leftrightarrow \frac{1}{4(1 + cL)} > \frac{\lambda cL}{1 + cL} \Leftrightarrow \frac{1}{4} > \lambda cL,$$

where the last inequality holds by the choice of $c \leq \frac{1}{4\Lambda L} \leq \frac{1}{4\lambda L}$. Therefore, we obtain

$$\frac{c/2}{1 + cL} \sum_{k=0}^{K-1} \mathbb{E}\left[f(x^k) - f(x^*)\right] \leq \frac{c}{1 + cL}(f(x^0) - f(x^*)) + \frac{2c(1/2 + \lambda)}{1 + cL}\mathbb{E}\left[f(x^{K-1}) - f(x^*)\right]$$

$$+ 2c\frac{1/4}{1 + cL} \sum_{k=1}^{K-2} \mathbb{E}\left[f(x^k) - f(x^*)\right]$$

$$\leq \frac{c}{1 + cL}(f(x^0) - f(x^*)) + \frac{2c(1/2 + \lambda)}{1 + cL}\mathbb{E}\left[f(x^{K-1}) - f(x^*)\right]$$

$$+ 2c\left(\frac{1/2 + \lambda}{1 + cL} - \lambda\right) \sum_{k=1}^{K-2} \mathbb{E}\left[f(x^k) - f(x^*)\right]$$

$$\leq \|x^0 - x^*\|^2 + 2c\lambda(f(x^0) - f(x^*)).$$

Thus, we obtain the rate

$$\frac{1}{K} \sum_{k=0}^{K-1} \mathbb{E}\left[f(x^k) - f(x^*)\right] \leq \frac{2(1 + cL)(\|x^0 - x^*\|^2 + 2c\lambda(f(x^0) - f(x^*)))}{cK}$$

$$\leq \frac{3(\|x^0 - x^*\|^2 + 1/2L(f(x^0) - f(x^*)))}{cK},$$

where the last inequality holds by the choice of $c$.

$\square$

### D.3 Convergence of NGN-M with Decaying Step-size

**Lemma D.4.** *We have*

$$\sum_{k=0}^{K-1} \frac{1}{k + 1} \leq \log(K + 2), \quad \sum_{k=0}^{K-1} \frac{1}{\sqrt{k + 1}} \geq \frac{4}{5}\sqrt{K + 1}. \tag{34}$$

*Proof.* We refer to Lemma A.8 from Garrigos and Gower [22]. $\square$

To prove the convergence of NGN-M with decaying $c_k$ we consider IMA formulation (see Section A in the paper):

$$x^{-1} = z^0 = x^0, \quad z^{k+1} = z^k - \gamma_k \nabla f_{S_k}(x^k), \quad \gamma_k = \frac{c_k}{1 + \frac{c_k}{2f_{S_k}(x^k)}\|\nabla f_{S_k}(x^k)\|^2}$$

$$x^{k+1} = \frac{\lambda_{k+1}}{1 + \lambda_{k+1}}x^k + \frac{1}{1 + \lambda_{k+1}}z^{k+1},$$

where $c_k = \frac{c_0}{\sqrt{k+1}}, \lambda_0 = 0$.

**Theorem D.5.** *Assume that each $f_i$ is convex and $L$-smooth, and that Assumption 3.2 holds. Let the step-size hyperparameter is set $c_k = \frac{c_0}{\sqrt{k}}$, momentum parameter $\lambda_k \leq \min\{c_k L, 0.5(1 + c_k L)^{-1}(1 + 2c_k L)^{-1}\}$. Then the iterates of NGN-M satisfy*

$$\mathbb{E}\left[f(\hat{x}^{K-1}) - f(x^*)\right] \leq \frac{5(1 + c_0 L)(1 + 2c_0 L)\|x^0 - x^*\|^2}{4c_0\sqrt{K}}$$

$$+ 10Lc_0(1 + c_0 L)(1 + 2c_0 L)\sigma_{\text{int}}^2\frac{\log(K + 2)}{\sqrt{K}}$$

$$+ 5c_0 L(1 + c_0 L)\frac{\log(K + 2)}{2\sqrt{K}}\max\{2c_0 L - 1, 0\}\sigma_{\text{pos}}^2, \tag{35}$$

*where $\hat{x}^{K-1} = \sum_{k=0}^{K-1} \frac{\rho_k}{\sum_{k=0}^{K-1}\rho_k}x^k, \rho_k = \frac{c_k}{(1 + c_k L)(1 + 2c_k L)}$.*

*Proof.* At iteration $k = 0$ we have

$$z^1 = z^0 - \gamma_0 \nabla f_{S_0}(x^0) = x^0 - \gamma_0 \nabla f_{S_0}(x^0).$$

Therefore, we get

$$
\begin{aligned}
\|z^1 - x^*\|^2 \quad &= \quad \|z^0 - x^*\|^2 - 2\gamma_0 \langle \nabla f_{S_0}(x^0), z^0 - x^* \rangle + \gamma_0^2 \|\nabla f_{S_0}(x^0)\|^2 \\
&\overset{\text{Lem. B.6}}{\leq} \quad \|z^0 - x^*\|^2 - 2\gamma_0 \langle \nabla f_{S_0}(x^0), x^0 - x^* \rangle + \frac{4 c_0 L}{1 + 2 c_0 L} \gamma_0 (f_{S_0}(x^0) - f_{S_0}^*) \\
&\qquad + \frac{2 c_0^2 L}{1 + c_0 L} \max\left\{ \frac{2 c_0 L - 1}{2 c_0 L + 1}, 0 \right\} f_{S_0}^*.
\end{aligned}
\tag{36}
$$

Let $\gamma_0 = \rho_0 + \widetilde{\gamma}_0$ where $\rho_0 = \frac{c_0}{(1 + c_0 L)(1 + 2 c_0 L)}$. Then we have

$$
\begin{aligned}
\widetilde{\gamma}_0 &= \gamma_0 - \rho_0 \\
&\overset{\text{Lem. B.5}}{\leq} c_0 - \frac{c_0}{(1 + c_0 L)(1 + 2 c_0 L)} \\
&= c_0 \frac{1 + 3 c_0 L + 2 c_0^2 L^2 - 1}{(1 + c_0 L)(1 + 2 c_0 L)} \\
&= c_0^2 L \frac{3 + 3 c_0 L}{(1 + c_0 L)(1 + 2 c_0 L)} \\
&= \frac{3 c_0^2 L}{1 + 2 c_0 L}.
\end{aligned}
$$

Using the above we continue from (36)

$$
\begin{aligned}
\|z^1 - x^*\|^2 &\overset{\text{conv.}}{\leq} \|z^0 - x^*\|^2 - 2\gamma_0 (f_{S_0}(x^0) - f_{S_0}(x^*)) + \frac{4 c_0 L}{1 + 2 c_0 L} \gamma_0 (f_{S_0}(x^0) - f_{S_0}^*) \\
&\qquad + \frac{2 c_0^2 L}{1 + c_0 L} \max\left\{ \frac{2 c_0 L - 1}{2 c_0 L + 1}, 0 \right\} f_{S_0}^* \\
&\leq \|z^0 - x^*\|^2 - 2\rho_0 (f_{S_0}(x^0) - f_{S_0}(x^*)) - 2\widetilde{\gamma}_0 (f_{S_0}(x^0) - f_{S_0}^*) + 2\widetilde{\gamma}_0 (f_{S_0}(x^*) - f_{S_0}^*) \\
&\qquad + \frac{4 c_0 L}{1 + 2 c_0 L} \gamma_0 (f_{S_0}(x^0) - f_{S_0}^*) + \frac{2 c_0^2 L}{1 + c_0 L} \max\left\{ \frac{2 c_0 L - 1}{2 c_0 L + 1}, 0 \right\} f_{S_0}^* \\
&= \|z^0 - x^*\|^2 - 2\rho_0 (f_{S_0}(x^0) - f_{S_0}(x^*)) - 2\left( \gamma_0 - \rho_0 - \frac{2 c_0 L}{1 + 2 c_0 L} \gamma_0 \right) (f_{S_0}(x^0) - f_{S_0}^*) \\
&\qquad + 2\widetilde{\gamma}_0 (f_{S_0}(x^*) - f_{S_0}^*) + \frac{2 c_0^2 L}{1 + c_0 L} \max\left\{ \frac{2 c_0 L - 1}{2 c_0 L + 1}, 0 \right\} f_{S_0}^*.
\end{aligned}
\tag{37}
$$

Here we have

$$
\begin{aligned}
\gamma_0 - \rho_0 - \frac{2 c_0 L}{1 + 2 c_0 L} \gamma_0 &= \frac{1}{1 + 2 c_0 L} \gamma_0 - \rho_0 \\
&= \frac{1}{1 + 2 c L} \gamma_0 - \frac{c_0}{(1 + c_0 L)(1 + 2 c_0 L)} \\
&\overset{Lem.B.5}{\geq} \frac{1}{1 + 2 c_0 L} \frac{c_0}{1 + c_0 L} - \frac{c_0}{(1 + c_0 L)(1 + 2 c L)} \\
&= 0,
\end{aligned}
$$

$\widetilde{\gamma}_0 \leq \frac{3 c_0^2 L}{1 + 2 c_0 L}$, and $f_{S_0}(x^0) - f_{S_0}^* \geq 0$. Hence, we get

$$
\begin{aligned}
\|z^1 - x^*\|^2 &\leq \|z^0 - x^*\|^2 - 2\rho_0 (f_{S_0}(x^0) - f_{S_0}(x^*)) + \frac{6 c_0^2 L}{1 + 2 c_0 L} (f_{S_0}(x^*) - f_{S_0}^*) \\
&\qquad + \frac{2 c_0^2 L}{1 + c_0 L} \max\left\{ \frac{2 c_0 L - 1}{2 c_0 L + 1}, 0 \right\} f_{S_0}^*.
\end{aligned}
$$

Rearranging terms and taking the expectation we get

$$2\rho_0 \mathbb{E}\left[f(x^0) - f(x^*)\right] \leq \mathbb{E}\left[\|z^1 - x^*\|^2\right] - \|z^0 - x^*\|^2 + \frac{6c_0^2 L}{1 + 2c_0 L}\sigma_{\text{int}}^2$$

$$+ \frac{2c_0^2 L}{1 + c_0 L}\max\left\{\frac{2c_0 L - 1}{2c_0 L + 1}, 0\right\}\sigma_{\text{pos}}^2. \tag{38}$$

Next, for $k > 0$ we can use the relation $z^k = x^k + \lambda_k(x^k - x^{k-1})$. We expand $\|z^{k+1} - x^*\|^2$

$$
\begin{aligned}
\|z^{k+1} - x^*\|^2 &= &\|z^k - x^*\|^2 - 2\gamma_k\langle\nabla f_{S_k}(x^k), z^k - x^*\rangle + \gamma_k^2\|\nabla f_{S_k}(x^k)\|^2 \\
&= &\|z^k - x^*\|^2 - 2\gamma_k\langle\nabla f_{S_k}(x^k), x^k - x^*\rangle - 2\gamma_k\lambda_k\langle\nabla f_{S_k}(x^k), x^k - x^{k-1}\rangle \\
& &+ \gamma_k^2\|\nabla f_{S_k}(x^k)\|^2 \\
&\overset{\text{conv.}}{\leq} &\|z^k - x^*\|^2 - 2\gamma_k(f_{S_k}(x^k) - f_{S_k}(x^*)) - 2\gamma_k\lambda(f_{S_k}(x^k) - f_{S_k}(x^{k-1})) \\
& &+ \gamma_k^2\|\nabla f_{S_k}(x^k)\|^2 \\
&\overset{\text{Lem. B.6}}{\leq} &\|z^k - x^*\|^2 - 2\gamma_k(f_{S_k}(x^k) - f_{S_k}(x^*)) - 2\gamma_k\lambda_k(f_{S_k}(x^k) - f_{S_k}(x^{k-1})) \\
& &+ \frac{4c_k L}{1 + 2c_k L}\gamma_k(f_{S_k}(x^k) - f_{S_k}^*) + \frac{2c_k^2 L}{1 + c_k L}\max\left\{\frac{2c_k L - 1}{2c_k L + 1}, 0\right\}f_{S_k}^*.
\end{aligned}
$$

Let $\gamma_k = \rho_k + \widetilde{\gamma}_k$, where $\rho, \widetilde{\gamma}_k \geq 0$, and $\rho$ is a constant step-size independent of $S_k$ which will be defined later. Therefore, we have

$$
\begin{aligned}
\|z^{k+1} - x^*\|^2 &\leq &\|z^k - x^*\|^2 - 2\rho_k(f_{S_k}(x^k) - f_{S_k}(x^*)) - 2\widetilde{\gamma}_k(f_{S_k}(x^k) - f_{S_k}(x^*)) \\
& &- 2\gamma_k\lambda_k(f_{S_k}(x^k) - f_{S_k}^*) + 2\gamma_k\lambda(f_{S_k}(x^{k-1}) - f_{S_k}^*) \\
& &+ \frac{4c_k L}{1 + 2c_k L}\gamma_k(f_{S_k}(x^k) - f_{S_k}^*) + \frac{2c_k^2 L}{1 + c_k L}\max\left\{\frac{2c_k L - 1}{2c_k L + 1}, 0\right\}f_{S_k}^* \\
&= &\|z^k - x^*\|^2 - 2\rho(f_{S_k}(x^k) - f_{S_k}(x^*)) - 2\widetilde{\gamma}_k(f_{S_k}(x^k) - f_{S_k}^*) + 2\widetilde{\gamma}_k(f_{S_k}(x^*) - f_{S_k}^*) \\
& &- 2\gamma_k\lambda(f_{S_k}(x^k) - f_{S_k}^*) + 2\gamma_k\lambda(f_{S_k}(x^{k-1}) - f_{S_k}^*) \\
& &+ \frac{4c_k L}{1 + 2c_k L}\gamma_k(f_{S_k}(x^k) - f_{S_k}^*) + \frac{2c_k^2 L}{1 + c_k L}\max\left\{\frac{2c_k L - 1}{2c_k L + 1}, 0\right\}f_{S_k}^* \\
&= &\|z^k - x^*\|^2 - 2\rho(f_{S_k}(x^k) - f_{S_k}(x^*)) - 2\left(\widetilde{\gamma}_k + \gamma_k\lambda - \frac{2c_k L}{1 + 2c_k L}\gamma_k\right)(f_{S_k}(x^k) - f_{S_k}^*) \\
& &+ 2\widetilde{\gamma}_k(f_{S_k}(x^*) - f_{S_k}^*) + 2\gamma_k\lambda(f_{S_k}(x^{k-1}) - f_{S_k}^*) \\
& &+ \frac{2c_k^2 L}{1 + c_k L}\max\left\{\frac{2c_k L - 1}{2c_k L + 1}, 0\right\}f_{S_k}^*. \tag{39}
\end{aligned}
$$

We need to find $\rho_k$ such that

$$\widetilde{\gamma}_k + \gamma_k\lambda - \frac{2c_k L}{1 + 2c_k L}\gamma_k \geq 0$$

Since $\widetilde{\gamma}_k = \gamma_k - \rho_k$, then we have

$$\gamma_k - \rho_k + \gamma_k\lambda_k - \frac{2c_k L}{1 + 2c_k L}\gamma_k \geq 0$$

$$\Leftrightarrow \gamma_k\left(1 + \lambda_k - \frac{2c_k L}{1 + 2c_k L}\right) \geq \rho_k.$$

The inequality above is satisfied if it is satisfied for the lower bound on $\gamma_k$ (which is $c/1+cL$), i.e.

$$\frac{c_k}{1 + c_k L}\left(\frac{1}{1 + 2c_k L} + \lambda\right) \geq \rho.$$

We can take $\rho_k = \frac{c_k}{(1+c_kL)(1+2c_kL)}$ since $\lambda \geq 0$.

$$\widetilde{\gamma}_k = \gamma_k - \rho_k$$
$$\leq c_k - \frac{c_k}{(1+c_kL)(1+2c_kL)}$$
$$= c_k \frac{1 + 3c_kL + 2c_k^2L^2 - 1}{(1+c_kL)(1+2c_kL)}$$
$$\leq c_k^2 L \frac{3 + 3c_kL}{(1+c_kL)(1+2c_kL)}$$
$$= \frac{3c_k^2L}{1+2c_kL}.$$

Using the above, we get from (39)

$$\|z^{k+1} - x^*\|^2 \leq \|z^k - x^*\|^2 - 2\rho_k(f_{S_k}(x^k) - f_{S_k}(x^*)) + 2c_k\lambda_k(f_{S_k}(x^{k-1}) - f_{S_k}(x^*))$$
$$+ 2c_k\lambda_k(f_{S_k}(x^*) - f_{S_k}^*) + \frac{6c_k^2L}{1+2c_kL}(f_{S_k}(x^*) - f_{S_k}^*)$$
$$+ \frac{2c_k^2L}{1+c_kL}\max\left\{\frac{2c_kL-1}{2c_kL+1}, 0\right\}f_{S_k}^*.$$

Taking expectations, we get

$$\mathbb{E}\left[\|z^{k+1} - x^*\|^2\right] \leq \mathbb{E}\left[\|z^k - x^*\|^2\right] - 2\rho_k\mathbb{E}\left[f(x^k) - f(x^*)\right] + 2c_k\lambda_k\mathbb{E}\left[f(x^{k-1}) - f(x^*)\right]$$
$$+ \left(2c_k\lambda_k + \frac{6c_k^2L}{1+2c_kL}\right)\sigma_{\text{int}}^2 + \frac{2c_k^2L}{1+c_kL}\max\left\{\frac{2c_kL-1}{2c_kL+1}, 0\right\}\sigma_{\text{pos}}^2 \quad (40)$$

Rearranging terms, we get

$$2\rho_k\mathbb{E}\left[f(x^k) - f(x^*)\right] - 2c_k\lambda_k\mathbb{E}\left[f(x^{k-1}) - f(x^*)\right] \leq \mathbb{E}\left[\|z^k - x^*\|^2\right] - \mathbb{E}\left[\|z^{k+1} - x^*\|^2\right]$$
$$+ \left(2c_k\lambda_k + \frac{6c_k^2L}{1+2cL}\right)\sigma_{\text{int}}^2$$
$$+ \frac{2c_k^2L}{1+c_kL}\max\left\{\frac{2c_kL-1}{2c_kL+1}, 0\right\}\sigma_{\text{pos}}^2. \quad (41)$$

Combining (38) and (41) for iterations $\{1, \ldots, K-1\}$ we get

$$2\rho_0\mathbb{E}\left[f(x^0) - f(x^*)\right] + 2\sum_{k=1}^{K-1}\rho_k\mathbb{E}\left[f(x^k) - f(x^*)\right] - 2\sum_{k=1}^{K-1}c_k\lambda_k\mathbb{E}\left[f(x^{k-1}) - f(x^*)\right]$$

$$= 2\sum_{k=0}^{K-1}\rho_k\mathbb{E}\left[f(x^k) - f(x^*)\right] - 2\sum_{k=0}^{K-2}c_k\lambda_k\mathbb{E}\left[f(x^k) - f(x^*)\right]$$

$$\leq 2\sum_{k=0}^{K-1}(\rho_k - c_k\lambda_k)\mathbb{E}\left[f(x^k) - f(x^*)\right]$$

$$\leq \|z^0 - x^*\|^2 + \frac{6c_0^2L}{1+2c_0L}\sigma_{\text{int}}^2 + \frac{2c_0^2L}{1+c_0L}\max\left\{\frac{2c_0L-1}{2c_0L+1}, 0\right\}\sigma_{\text{pos}}^2$$

$$+ \sum_{k=1}^{K-1}\left(2c_k\lambda_k + \frac{6c_k^2L}{1+2c_kL}\right)\sigma_{\text{int}}^2 + \sum_{k=1}^{K-1}\frac{2c_k^2L}{1+c_kL}\max\left\{\frac{2c_kL-1}{2c_kL+1}, 0\right\}\sigma_{\text{pos}}^2. \quad (42)$$

Note that choosing $\lambda_k = \min\left\{c_kL, 0.5(1+c_kL)^{-1}(1+2c_kL)^{-1}\right\}$ ensures that $\frac{\rho_k}{2} \geq c_k\lambda_k$. Indeed, we have

$$\frac{\rho_k}{2} = \frac{c_k}{2(1+c_kL)(1+2c_kL)} > c_k\lambda_k$$
$$\Leftrightarrow 1 > 2\lambda_k(1+c_kL)(1+2c_kL).$$

Therefore, from (42) and the facts that $\lambda_0 = 0$ and $\lambda_k \leq c_k L$ we get

$$\sum_{k=0}^{K-1} \rho_k \mathbb{E}\left[f(x^k) - f(x^*)\right] \leq \|z^0 - x^*\|^2 + \sum_{k=0}^{K-1}\left(2c_k\lambda_k + \frac{6c_k^2 L}{1 + 2c_k L}\right)\sigma_{\text{int}}^2$$

$$+ \sum_{k=0}^{K-1} \frac{2c_k^2 L}{1 + c_k L}\max\left\{\frac{2cL - 1}{2cL + 1}, 0\right\}\sigma_{\text{pos}}^2$$

$$\leq \|z^0 - x^*\|^2 + 8L\sigma_{\text{int}}^2 \sum_{k=0}^{K-1} c_k^2$$

$$+ \sum_{k=0}^{K-1} 2c_k^2 L \max\left\{\frac{2c_k L - 1}{2c_k L + 1}, 0\right\}\sigma_{\text{pos}}^2. \tag{43}$$

We have by Lemma D.4

$$\sum_{k=0}^{K-1} \rho_k = \sum_{k=0}^{K-1} \frac{c_k}{(1 + c_k L)(1 + 2c_k L)} \geq \sum_{k=0}^{K-1} \frac{c_k}{(1 + c_0 L)(1 + 2c_0 L)} \geq \frac{4c_0\sqrt{K}}{5(1 + c_0 L)(1 + 2c_0 L)},$$

$$\sum_{k=0}^{K-1} c_k^2 \overset{\text{Lem } D.4}{\leq} c_0^2 \log(K + 2), \tag{44}$$

$$\sum_{k=0}^{K-1} c_k^2 \max\left\{\frac{2c_k L - 1}{2c_k L + 1}, 0\right\} \leq \sum_{k=0}^{K-1} c_k^2 \max\left\{\frac{2c_0 L - 1}{2c_0 L + 1}, 0\right\} \leq c_0^2 \log(K + 2)\max\left\{\frac{2c_0 L - 1}{2c_0 L + 1}, 0\right\}.$$

Therefore, using (44), $z^0 = x^0$ in (43) and dividing both sides in (43) by $\sum_{k=0}^{K-1} \rho_k$ we derive

$$\sum_{k=0}^{K-1} \frac{\rho_k}{\sum_{k=0}^{K-1} \rho_k}\mathbb{E}\left[f(x^k) - f(x^*)\right] \leq \frac{\|x^0 - x^*\|^2}{\sum_{k=0}^{K-1} \rho_k} + 8Lc_0^2\sigma_{\text{int}}^2 \frac{\log(K + 2)}{\sum_{k=0}^{K-1} \rho_k}$$

$$+ 2c_0^2 L\frac{\log(K + 2)}{\sum_{k=0}^{K-1} \rho_k}\max\left\{\frac{2c_0 L - 1}{2c_0 L + 1}, 0\right\}\sigma_{\text{pos}}^2. \tag{45}$$

With an lower bound on $\sum_{k=0}^{K-1}$ and Jensen's inequality we conclude that

$$\mathbb{E}\left[f(\hat{x}^{K-1}) - f(x^*)\right] \leq \frac{5(1 + c_0 L)(1 + 2c_0 L)\|x^0 - x^*\|^2}{4c_0\sqrt{K}}$$

$$+ 10Lc_0(1 + c_0 L)(1 + 2c_0 L)\sigma_{\text{int}}^2 \frac{\log(K + 2)}{\sqrt{K}}$$

$$+ 5c_0 L(1 + c_0 L)\frac{\log(K + 2)}{2\sqrt{K}}\max\{2c_0 L - 1, 0\}\sigma_{\text{pos}}^2, \tag{46}$$

where $\hat{x}^{K-1} = \sum_{k=0}^{K-1} \frac{\rho_k}{\sum_{k=0}^{K-1} \rho_k}x^k$.

$\square$

# E    Stability of NGN-M on a Simple Problem

We consider 1D convex functions of the form $f(x) = Lx^2(1 + p^2(x))$ that satisfy the following assumption.

**Assumption E.1.** There exists a constant $C$ such that $C(1 + p^2(x)) \geq xp(x)p'(x)$.

Note that $1 + p^2(x) \geq 1$ and $\deg(1 + p^2(x)) = \deg(xp(x)p'(x))$. Therefore, this assumption is mild.
*Remark* E.2. For example, the function $f(x) = x^2(1 + x^2)$ (i.e., $p(x) = x$) is convex and satisfies Assumption E.1 with $C = 1$.

*Remark* E.3. Let $p(x) = \sum_{j=0}^{m} a_j x^j$. Then for large values of $x$ in magnitude, $p(x) \sim a_m x^m, p'(x) \sim m a_m x^{m-1}$. Therefore, the constant $C$ should be expected of order $C \approx m$, where $m = \deg(p(x))$.

The function $f(x)$ is non-negative for any $x \in \mathbb{R}$ and its minimum $f^* = 0$ is attained at $x = 0$ by design. Let us compute a step of NGN-M on this problem

$$
\begin{aligned}
x^{k+1} &= x^k - (1-\beta) \frac{c}{1 + \frac{c}{2f(x^k)}(f'(x^k))^2} f'(x^k) + \beta(x^k - x^{k-1}) \\
&= x^k - (1-\beta) \frac{2Lc(1 + p^2(x^k) + x^k p(x^k) p'(x^k))}{1 + \frac{4L^2 c[x^k]^2}{2L[x^k]^2(1+p^2(x^k))}(1 + p^2(x^k) + x^k p(x^k) p'(x^k))^2} x^k + \beta(x^k - x^{k-1}) \\
&= x^k - (1-\beta) \underbrace{\frac{2Lc(1 + p^2(x^k) + x^k p(x^k) p'(x^k))}{1 + \frac{2Lc}{1+p^2(x^k)}(1 + p^2(x^k) + x^k p(x^k) p'(x^k))^2}}_{:= \hat{\gamma}_k} x^k + \beta(x^k - x^{k-1}). \quad (47)
\end{aligned}
$$

Note that the convexity of $f$ implies that

$$
\begin{aligned}
f(0) &\geq f(x) + f'(x)(0 - x) \\
0 &\geq Lx^2(1 + p^2(x)) - 2Lx^2(1 + p^2(x) + xp(x)p'(x)) \\
0 &\geq -Lx^2(1 + p^2(x)) - 2Lx^3 p(x) p'(x) \\
xp(x)p'(x) &\geq -\frac{1}{2}(1 + p^2(x)). \quad (48)
\end{aligned}
$$

In particular, (48) implies that $1 + p^2(x) + xp(x)p'(x) \geq \frac{1}{2}(1 + p^2(x)) > 0$. Therefore, we can obtain lower and upper bounds on $\hat{\gamma}_k$.

**Lemma E.4.** *Let Assumption E.1 hold with a constant $C > 0$ and $f(x) = x^2(1 + p^2(x))$ be convex. Let $c \geq \frac{1}{2L}$. Then we have $\hat{\gamma}_k \in \left[\frac{1}{2(1+C)}, 2\right]$.*

*Proof.* Indeed, the upper bound on $\hat{\gamma}_k$ follows from the following inequality

$$
\begin{aligned}
\hat{\gamma}_k &= \frac{2Lc(1 + p^2(x^k) + x^k p(x^k) p'(x^k))}{1 + \frac{2Lc}{1+p^2(x^k)}(1 + p^2(x^k) + x^k p(x^k) p'(x^k))^2} \\
&\leq \frac{2Lc(1 + p^2(x^k) + x^k p(x^k) p'(x^k))}{\frac{2Lc}{1+p^2(x^k)}(1 + p^2(x^k) + x^k p(x^k) p'(x^k))^2} \\
&= \frac{1 + p^2(x^k)}{1 + p^2(x^k) + x^k p(x^k) p'(x^k)} \leq 2, \quad (49)
\end{aligned}
$$

due to (48). The lower bound can be obtained as follows

$$
\begin{aligned}
\hat{\gamma}_k &= \frac{2Lc(1 + p^2(x^k) + x^k p(x^k) p'(x^k))}{1 + \frac{2Lc}{1+p^2(x^k)}(1 + p^2(x^k) + x^k p(x^k) p'(x^k))^2} \\
&= \frac{2Lc(1 + p^2(x^k) + x^k p(x^k) p'(x^k))(1 + p^2(x^k))}{(1 + p^2(x^k)) + 2Lc(1 + p^2(x^k) + x^k p(x^k) p'(x^k))^2} \\
&\geq \frac{2Lc(1 + p^2(x^k) + x^k p(x^k) p'(x^k))(1 + p^2(x^k))}{2(1 + p^2(x^k) + x^k p(x^k) p'(x^k)) + 2Lc(1 + p^2(x^k) + x^k p(x^k) p'(x^k))^2} \\
&= \frac{Lc(1 + p^2(x^k))}{1 + Lc(1 + p^2(x^k) + x^k p(x^k) p'(x^k))} \\
&= \frac{Lc(1 + p^2(x^k))}{1 + Lc(1 + p^2(x^k) + C(1 + p^2(x^k)))} \\
&\geq \frac{Lc(1 + p^2(x^k))}{2Lc(1 + C)(1 + p^2(x^k))} = \frac{1}{2(1 + C)} \quad (50)
\end{aligned}
$$

$\square$

The update rule of NGN-M can be rewritten as

$$x^{k+1} = x^k - (1-\beta)\hat{\gamma}_k x^k + \beta(x^k - x^{k-1}). \tag{51}$$

Let us consider the joint dynamics of $w^k := ([x^k]^\top, [x^{k-1}]^\top)^\top \in \mathbb{R}^{2d}$. We have that

$$w^k = \begin{pmatrix} x^k \\ x^{k-1} \end{pmatrix} = \begin{pmatrix} x^k - (1-\beta)\hat{\gamma}_k x^k + \beta(x^k - x^{k-1}) \\ x^{k-1} \end{pmatrix}$$

$$= \begin{pmatrix} \mathbf{I} - (1-\beta)\hat{\gamma}_k \mathbf{I} + \beta\mathbf{I} & -\beta\mathbf{I} \\ \mathbf{I} & \mathbf{0} \end{pmatrix} \begin{pmatrix} x^k \\ x^{k-1} \end{pmatrix} = \mathbf{G} w^{k-1}, \tag{52}$$

where

$$\mathbf{G} := \begin{pmatrix} \mathbf{I} - (1-\beta)\hat{\gamma}_k \mathbf{I} + \beta\mathbf{I} & -\beta\mathbf{I} \\ \mathbf{I} & \mathbf{0} \end{pmatrix}. \tag{53}$$

Now we are ready to prove the convergence of NGN-M on this simple problem for any value $c \geq \frac{1}{2}$.

**Theorem E.5.** *Let $f(x) = x^2(1 + p^2(x))$ be convex and Assumption E.1 holds. Let $\beta \geq \frac{(2(1+C)-1)^2}{(2(1+C)+1)^2}$ and $c \geq \frac{1}{2L}$. Then the iterates of NGN-M on $f(x)$ converge to the minimum $f^* = 0$.*

*Proof.* We follow the standard proof of SGD with Polyak momentum [71]. At this stage, we need to estimate the eigenvalues of $\mathbf{G}$. To do so, we will proceed with a permutation matrix $\Pi$[4] which transforms the matrix $\mathbf{G}$ to the block-diagonal matrix as

$$\mathbf{G} = \begin{pmatrix} \mathbf{G}_1 & 0 & \dots & 0 \\ \dots & \dots & \dots & \dots \\ 0 & 0 & \dots & \mathbf{G}_d \end{pmatrix}, \tag{54}$$

where

$$\mathbf{G}_i := \begin{pmatrix} 1 + \beta - (1-\beta)\hat{\gamma}_k & -\beta \\ 1 & 0 \end{pmatrix} \tag{55}$$

Since the matrix $\mathbf{G}$ is a block-diagonal matrix, we have $\|\mathbf{G}\| \leq \max_i \|\mathbf{G}_i\|$. Therefore, the problem is now simplified to bounding the spectral radii of the individual blocks $\mathbf{G}_i$, for $i = 1, 2, \dots, d$. The two eigenvalues $u_1$ and $u_2$ of $\mathbf{G}_i$ are the roots of the quadratic

$$q(u) := u^2 - (1 + \beta - (1-\beta)\hat{\gamma}_k)u + \beta = 0, \tag{56}$$

which take different values depending on the discriminant $\Delta := (1 + \beta - (1-\beta)\hat{\gamma}_k)^2 - 4\beta$. Let us find the values of $\beta$ when the discriminant is negative. We need to satisfy the inequality

$$(1 + \beta - (1-\beta)\hat{\gamma}_k)^2 - 4\beta \leq 0 \Leftrightarrow (1+\beta)^2 + (1-\beta)^2\hat{\gamma}_k^2 - 2(1+\beta)(1-\beta)\hat{\gamma}_k - 4\beta \leq 0$$

$$\Leftrightarrow (1-\beta)^2 + (1-\beta)^2\hat{\gamma}_k^2 - 2(1+\beta)(1-\beta)\hat{\gamma}_k \leq 0$$

$$\Leftrightarrow (1-\beta)(1 + \hat{\gamma}_k^2) \leq 2(1+\beta)\hat{\gamma}_k$$

$$\Leftrightarrow \frac{1 + \hat{\gamma}_k^2}{2\hat{\gamma}_k} \leq \frac{1+\beta}{1-\beta}. \tag{57}$$

Since the function $\frac{1+y^2}{2y}$ for $y \in \left[\frac{1}{2(1+C)}, 2\right]$ attains the maximum $\frac{4(1+C)^2+1}{4(1+C)}$ at $y = \frac{1}{2(1+C)}$, then we satisfy the last inequality, and consequently the discriminant is non-positive, if we choose

$$\frac{4(1+C)^2 + 1}{4(1+C)} \leq \frac{1+\beta}{1-\beta}. \tag{58}$$

---

[4] The permutation matrix $\Pi$ is defined as $\Pi_{ij} = \begin{cases} 1 & i \text{ odd}, j = i \\ 1 & i \text{ even}, j = 2n + i \\ 0 & \text{else} \end{cases}$. Note that permutation matrices preserve eigenvalues.

The above inequality is satisfied for $\beta \in \left[\frac{(2(1+C)-1)^2}{(2(1+C)+1)^2}, 1\right)$. Therefore, we obtain that for such choice of $\beta$ we have $\Delta_i \leq 0$ for all $i \in [d]$. Therefore, the zeros of the quadratic $q(u)$ are complex, and are equal in absolute value

$$|u_1| = |u_2| = \sqrt{\beta} < 1. \tag{59}$$

This gives us that $\|\mathbf{G}_i\| \leq \sqrt{\beta} < 1$. Therefore, the algorithm converges for any value of $\beta$ in this range.

It remains to use Lemma 11 from Foucart [20] which says that for a given matrix $\mathbf{A} \in \mathbb{R}^{d \times d}$, and $\epsilon > 0$, there exists a matrix norm $\|\cdot\|$ such that

$$\|\mathbf{A}\| \leq \rho(\mathbf{A}) + \epsilon, \tag{60}$$

where $\rho(\mathbf{A}) = \max\{|\lambda| : \lambda \text{ eigenvalue of } \mathbf{A}\}$ (spectral radius of $\mathbf{A}$).

Asymptotically [5] (as $k \to \infty$, one can show (see Theorem 12 in [20]) that

$$\|w^k\|_2 = \mathcal{O}(\rho(\mathbf{G})^k), \tag{61}$$

where $\rho(\mathbf{G}) \leq \sqrt{\beta} < 1$ in our analysis. Therefore, NGN-M with hyperparameters $c \geq \frac{1}{2}$ and $\beta \geq \frac{1}{0}$ converges. $\qquad\square$

*Remark* E.6. For example, NGN-M converges on $f(x) = x^2(1 + x^2)$ for any $c \geq \frac{1}{2}$ and $\beta \geq \frac{9}{25}$.

Theorem E.5 shows that NGN-M remains stable even with an arbitrarily large step-size hyperparameter $c$. Thanks to the adaptive nature of NGN step-size, the actual update scale is automatically shrunk when necessary, preserving convergence. Importantly, this is possible with a choice of momentum parameter $\beta$ close to 1, which extends the results of Section 4. We acknowledge that our current analysis is restricted to the special convex class of 1D functions $f(x) = x^2(1 + p^2(x))$ satisfying Assumption E.1. Extending such stability guarantees to wider function classes with large momentum $\beta$ remains a significant open challenge.

To support the theoretical result, we test the performance of NGN-M and GDM (Gradient Descent with Momentum) on the problem $f(x) = x^2(1 + x^2)$, which is convex and satisfies Assumption E.1; see Figure E.1. We run both algorithms, varying the step-size hyperparameter in $\{10^{-4}, \ldots, 10^4\}$. We run algorithms for $10^5$ iterations. We stop training if the loss reaches a threshold $10^{-15}$ or exceeds $10^{10}$ for the first time. We observe that $(i)$ for small step-size hyperparameters, both methods converge but do not reach the threshold $10^{-15}$; $(ii)$ NGN-M reaches the threshold even for extremely large values of the step-size hyperparameter while GDM diverges. $(iii)$ the fastest convergence of GDM is achieved with the step-size hyperparameter $10^{-2}$ after 691 iterations while the fastest convergence of NGN-M is achieved with $c = 10^1$ after 269 iterations. In other details, NGN-M achieves faster convergence and much more stable to the choice of the step-size hyperparameter. These results align well with our theoretical analysis.

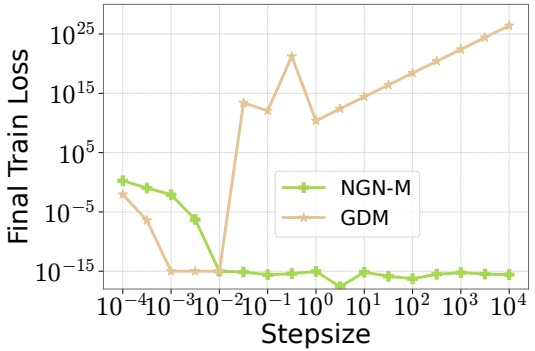

Figure E.1: Comparison of SGDM and NGN-M when minimizing a function $f(x) = x^2 + x^4$.

---

[5]A non-asymptotic version of the analysis can be derived using Theorem 5 by [91]

# F   How to Derive Diagonal NGN-based Step-size?

Here we provide derivations of how combine NGN and diagonal step-size following Section 3.3 for completeness.

We consider the following model

$$p^k = \operatorname{argmin}_{p \in \mathbb{R}^d} \left[ f_{\mathbf{\Sigma}_k, c}(x^k + p) := (r(x^k) + \nabla r(x^k)^\top p)^2 + \frac{1}{2c} \|p\|_{\mathbf{\Sigma}_k}^2 \right], \tag{62}$$

where $r(x) = \sqrt{f(x)}$. We compute the gradient of RHS of (62) w.r.t. $p$ and equal it to zero:

$$\nabla_p f_{\mathbf{\Sigma}_k, c}(x^k + p) = 2 \left( r(x^k) + \nabla r(x^k)^\top p \right) \nabla r(x^k) + \frac{1}{c} \mathbf{\Sigma}_k p$$
$$= \left( 2\nabla r(x^k) \nabla r(x^k)^\top + \frac{1}{c} \mathbf{\Sigma}_k \right) p + 2r(x^k) \nabla r(x^k).$$

Therefore, we have

$$p^k = -\left( 2\nabla r(x^k) \nabla r(x^k)^\top + \frac{1}{c} \mathbf{\Sigma}_k \right)^{-1} 2r(x^k) \nabla r(x^k).$$

Using Shermann-Morrison formula $(\mathbf{A} + uv^\top)^{-1} = \mathbf{A}^{-1} - \frac{\mathbf{A}^{-1} uv^\top \mathbf{A}^{-1}}{1 + u^\top \mathbf{A}^{-1} v}$ with $\mathbf{A} = {}^1\!/c \mathbf{\Sigma}_k$ we derive

$$p^k = -\left( c\mathbf{\Sigma}_k^{-1} - \frac{2c^2 \mathbf{\Sigma}_k^{-1} \nabla r(x^k) \nabla r(x^k)^\top \mathbf{\Sigma}_k^{-1}}{1 + 2c\nabla r(x^k)^\top \mathbf{\Sigma}_k^{-1} \nabla r(x^k)} \right) 2r(x^k) \nabla r(x^k)$$
$$= -2cr(x^k) \left( 1 - \frac{2c\nabla r(x^k)^\top \mathbf{\Sigma}_k^{-1} \nabla r(x^k)}{1 + 2c\nabla r(x^k) \mathbf{\Sigma}_k^{-1} \nabla r(x^k)} \right) \mathbf{\Sigma}_k^{-1} \nabla r(x^k)$$
$$= -\frac{2cr(x^k)}{1 + 2c\nabla r(x^k) \mathbf{\Sigma}_k^{-1} \nabla r(x^k)} \mathbf{\Sigma}_k^{-1} \nabla r(x^k).$$

Now we plug-in $r(x^k) = \sqrt{f(x^k)}$ and $\nabla r(x^k) = \frac{1}{2\sqrt{f(x^k)}} \nabla f(x^k)$ and obtain

$$p^k = -\frac{2c\sqrt{f(x^k)}}{1 + 2c\frac{1}{4f(x^k)} \nabla f(x^k)^\top \mathbf{\Sigma}_k^{-1} \nabla f(x^k)} \frac{1}{2\sqrt{f(x^k)}} \mathbf{\Sigma}_k^{-1} \nabla f(x^k)$$
$$= \frac{c}{1 + \frac{c}{2f(x^k)} \|\nabla f(x^k)\|_{\mathbf{\Sigma}_k^{-1}}^2} \mathbf{\Sigma}_k^{-1} \nabla f(x^k).$$

## F.1   Design Comparison of NGN-MDv1 and NGN-MDv2

The derivations in (3) are used to provide an intuition of how one can add a diagonal step-size into NGN by choosing the regularization matrix $\mathbf{\Sigma}_k$. By choosing $\mathbf{\Sigma}_k = \mathbf{D}_k$ we recover the update direction of NGN-MDv1. In this case, we have only one global NGN step-size in front of $\mathbf{D}_k$. The design of NGN-MDv2 follows a more straightforward intuition. In particular, it can be seen as a direct extension of NGN to diagonal case by replacing the squared gradient norm $\|\nabla f_{S_k}(x^k)\|^2$ by the squared partial derivative $(\nabla_j f_{S_k}(x^k))^2$ for each parameter $j \in [d]$.

The main difference in comparison with Adam is the order in which the preconditioning and momentum is applied. In both NGN-MDv1 and NGN-MDv2 we average the preconditioned updates $\mathbf{\Sigma}_k^{-1} \nabla f_{S_k}(x^k)$, i.e. we first apply preconditioning and momentum later. In contrast, in Adam the stochastic gradients are averaged to construct new momentum term, and then the momentum is preconditioned. In other words, the momentum is applied first and then it is followed by preconditioning. We believe this change might be one of the reasons behind the step-size hyperparameter resilience as well.

In practice, we found out that the tuned performance of NGN-MDv1 is slightly better than that of NGN-MDv2. Moreover, NGN-MDv1 demonstrates higher resilience to the choice of the step-size hyperparameter than NGN-MDv2.

Table 5: Train time of Adam and NGN-MDv1 when training language models.

| Model | Method | Time per Iteration (sec) | Time per Optimizer Update (sec) |
|---|---|---|---|
| 70M | AdamW | $1.63 \pm 0.01$ | $0.0048 \pm 0.0002$ |
| | NGN-MDv1 | $1.65 \pm 0.01$ | $0.0130 \pm 0.0002$ |
| 160M | AdamW | $3.33 \pm 0.03$ | $0.0088 \pm 0.0003$ |
| | NGN-MDv1 | $3.37 \pm 0.02$ | $0.0239 \pm 0.0003$ |
| 410M | AdamW | $8.41 \pm 0.06$ | $0.0838 \pm 0.0009$ |
| | NGN-MDv1 | $8.68 \pm 0.06$ | $0.2154 \pm 0.0007$ |

## F.2 Computation Cost of NGN-MD

Implementing any version of NGN-MD in practice might be slightly more computationally expensive. However, we highlight that computing a step of NGN-MD does not involve matrix-vector operations since the preconditioner is a diagonal matrix, and the matrix notation is used only for the convenience of presentation. The additional computation cost that we have in NGN-MDv1 is the computation of $\|\nabla f_{S_k}(x^k)\|^2_{\mathbf{D}_k^{-1}}$. This can naïvely be done by one additional pass over the gradient and summing the terms $\frac{1}{(\mathbf{D}_k)_j}(\nabla_j f_{S_k}(x^k))^2$ for $j \in [d]$. This operation does not require additional matrix multiplication. However, it can be computed more efficiently while updating $\mathbf{D}_k$. The rest of the NGN-MDv1 implementation does not add any significantly costly operations in comparison with Adam.

We compare in Table 5 the time per iteration and optimizer update when training language models from Section 5 using AdamW and NGN-MDv1. We notice that our naive implementation of NGN-MDv1 is about 2.5 times slower than PyTorch's AdamW. This is expected since our algorithm requires two passes over the gradient. Nevertheless, in this setting training time is dominated by forward and backward computations, keeping NGN-MDv1 competitive with AdamW. Moreover, as noted above, this overhead can be largely eliminated by computing the weighted gradient concurrently with the second-momentum $v^k$ update. We do not aim to provide the most efficient implementation of NGN-MDv1 as the primary goal of our work is to highlight the stability advantages that NGN step-size brings in the training of neural networks.

### F.2.1 Distributed Training

In a vanilla DDP implementation [50], computing the weighted gradient norm $\|\nabla f_{S_k}(x^k)\|^2_{\mathbf{D}_k^{-1}}$ is straightforward since gradients are replicated across devices. We only require an additional all-reduce to synchronize $f_{S_k}(x^k)$ across devices, which is, however, a lightweight communication (just a single float) and, in principle, can even be overlapped with the backward pass.

However, with more sophisticated types of parallelism, like Tensor Parallel [83] or ZeRO-2 [75], computing the weighted gradient norm introduces additional communication, as gradients are sharded across devices. This could still be implemented efficiently by accumulating squared gradient entries in each device and all-reducing only a single float, but it will, nevertheless, result in a computation and communication overhead for NGN-MDv1. We acknowledge that our methods might not be scalable to large distributed training, and adjustments are needed to make NGN-MDv1 work in this case. Nonetheless, we believe that our findings offer useful insights toward designing more stable optimization algorithms.

## G   How to add weight decay to NGN-MDv1?

Regularization techniques serve a fundamental purpose in minimizing generalization error. Orthogonal to their role for generalization, modern deep learning tasks often benefit from the use of weight decay [98]. Despite its widespread application, the role of weight decay is poorly understood. Andriushchenko et al. [2] suggested that it might provide implicit regularization by stabilizing the loss in over-parameterized neural networks and helping to balance the bias-variance tradeoff that leads to lower training loss in under-parameterized networks. However, even in the case of SGD, there is still uncertainty regarding how the weight decay mechanism should be incorporated, as various implementations may exist [102].

We propose two ways of adding weight decay to NGN-MDv1. The first variant follows the approach of [55], adding decoupled weight decay $\lambda$:

$$x^{k+1} = x^k - \lambda c x^k - (1 - \beta_1)\boldsymbol{\Sigma}_k^{-1}\nabla f_{S_k}(x^k) + \beta_1(x^k - x^{k-1}). \tag{63}$$

In this update rule, the weight is added separately from the update direction $\boldsymbol{\Sigma}_k^{-1}\nabla f_{S_k}(x^k)$. We call the resulting algorithm (63) Dec-NGN-MDv1, that stands for decoupled NGN-MDv1.

### G.1 Combining NGN-MDv1 and Weight Decay Regularization

We now discuss how to combine NGN-MDv1 and weight decay, following the idea that weight decay should perform weight regularization.

We consider the following model

$$f_{\boldsymbol{\Sigma}_k,\lambda}(x^k + p) := (r(x^k) + \nabla r(x^k)^\top p)^2 + \frac{1}{2c}\|p\|_{\boldsymbol{\Sigma}_k}^2 + \frac{\lambda}{2}\|x^k + p\|_{\boldsymbol{\Sigma}_k}^2.$$

By taking the gradient of $f_{\boldsymbol{\Sigma}_k,\lambda}$ w.r.t. $p$ we get

$$0 = 2(r(x^k) + \nabla r(x^k)^\top p)\nabla r(x^k) + \frac{1}{c}\boldsymbol{\Sigma}_k p + \lambda\boldsymbol{\Sigma}_k(x^k + p)$$

$$= \left(2\nabla r(x^k)\nabla r(x^k)^\top + \frac{1}{c}\boldsymbol{\Sigma}_k + \lambda\boldsymbol{\Sigma}_k\right)p + 2r(x^k)\nabla r(x^k) + \lambda\boldsymbol{\Sigma}_k x^k.$$

Therefore, we get

$$p^k = -\left(2\nabla r(x^k)\nabla r(x^k)^\top + \frac{1}{c}\boldsymbol{\Sigma}_k + \lambda\boldsymbol{\Sigma}_k\right)^{-1}(2r(x^k)\nabla r(x^k) + \lambda\boldsymbol{\Sigma}_k x^k).$$

Using Sherman-Morrison formula $(\mathbf{A} + uv^\top)^{-1} = \mathbf{A}^{-1} - \frac{\mathbf{A}^{-1}uv^\top\mathbf{A}^{-1}}{1 + u^\top\mathbf{A}^{-1}v}$ with $\mathbf{A} = (\lambda + 1/c)\boldsymbol{\Sigma}_k$ and $u = v = \sqrt{2}\nabla r(x^k)$ we get that

$$\left(2\nabla r(x^k)\nabla r(x^k)^\top + \frac{1}{c}\boldsymbol{\Sigma}_k + \lambda\boldsymbol{\Sigma}_k\right)^{-1}$$

$$= \frac{c}{1 + \lambda c}\boldsymbol{\Sigma}_k^{-1} - \frac{\frac{2c^2}{(1+\lambda c)^2}\boldsymbol{\Sigma}_k^{-1}\nabla r(x^k)\nabla r(x^k)^\top\boldsymbol{\Sigma}_k^{-1}}{1 + \frac{2c}{1+\lambda c}\nabla r(x^k)\boldsymbol{\Sigma}_k^{-1}\nabla r(x^k)}.$$

Therefore, we have

$$p^k = -\left(\frac{c}{1 + \lambda c}\boldsymbol{\Sigma}_k^{-1} - \frac{\frac{2c^2}{(1+\lambda c)^2}\boldsymbol{\Sigma}_k^{-1}\nabla r(x^k)\nabla r(x^k)^\top\boldsymbol{\Sigma}_k^{-1}}{1 + \frac{2c}{1+\lambda c}\nabla r(x^k)\boldsymbol{\Sigma}_k^{-1}\nabla r(x^k)}\right)(2r(x^k)\nabla r(x^k) + \lambda\boldsymbol{\Sigma}_k x^k)$$

$$= -\frac{2cr(x^k)}{1 + \lambda c}\left(1 - \frac{\frac{2c}{1+\lambda c}\nabla r(x^k)^\top\boldsymbol{\Sigma}_k^{-1}\nabla r(x^k)}{1 + \frac{2c}{1+\lambda c}\nabla r(x^k)\boldsymbol{\Sigma}_k^{-1}\nabla r(x^k)}\right)\boldsymbol{\Sigma}_k\nabla r(x^k)$$

$$\quad - \frac{\lambda c}{1 + \lambda c}x^k + \frac{\frac{2c^2\lambda}{1+\lambda c}\boldsymbol{\Sigma}_k^{-1}\nabla r(x^k)\nabla r(x^k)^\top x^k}{1 + \frac{2c}{1+\lambda c}\nabla r(x^k)\boldsymbol{\Sigma}_k^{-1}\nabla r(x^k)}$$

$$= -\frac{2cr(x^k)}{1 + \lambda c}\frac{1}{1 + \frac{2c}{1+\lambda c}\nabla r(x^k)\boldsymbol{\Sigma}_k^{-1}\nabla r(x^k)}\boldsymbol{\Sigma}_k^{-1}\nabla r(x^k)$$

$$\quad - \frac{\lambda c}{1 + \lambda c}x^k + \frac{\frac{2c^2\lambda}{1+\lambda c}\boldsymbol{\Sigma}_k^{-1}\nabla r(x^k)\nabla r(x^k)^\top x^k}{1 + \frac{2c}{1+\lambda c}\nabla r(x^k)\boldsymbol{\Sigma}_k^{-1}\nabla r(x^k)}.$$

---
**Algorithm 4** NGN-MDv1W
---

1: **Input:** $x^0 \in \mathbb{R}^d$, step-size parameter $c > 0$, momentum parameters $\beta_1, \beta_2 \in [0, 1)$, weight decay parameter $\lambda \geq 0$, stabilization parameter $\varepsilon > 0$
2: **for** $k = 0, 1, \ldots, K - 1$ **do**
3:      Sample a batch $S_k \subseteq [n]$ and compute $f_{S_k}$ and $\nabla f_{S_k}(x^k)$
4:      Compute $v^k = \beta_2 v^{k-1} + (1 - \beta_2)(\nabla f_{S_k}(x^k) \odot \nabla f_{S_k}(x^k))$
5:      Compute $\mathbf{D}_k = \mathrm{diag}(\varepsilon \mathbf{I} + \sqrt{v^k / (1 - \beta_2^k)})$
6:      Compute

$$\gamma_k = \frac{\frac{c}{(1+\lambda c)} \left[ 1 - \frac{c\lambda}{2 f_{S_k}(x^k)} \nabla f_{S_k}(x^k)^\top x^k \right]_+}{1 + \frac{c}{2 f_{S_k}(x^k)(1+\lambda c)} \|\nabla f_{S_k}(x^k)\|^2_{\mathbf{D}_k^{-1}}}$$

7:      Update $x^{k+1} = \frac{1}{1+\lambda c} x^k - (1 - \beta_1) \gamma_k \mathbf{D}_k^{-1} \nabla f_{S_k}(x^k) + \beta_1 (x^k - x^{k-1})$
8: **end for**
    $[\cdot]_+$ denotes $\max\{0, \cdot\}$.

---

Using the connection $\nabla r(x^k) = \frac{1}{2\sqrt{f(x^k)}} \nabla f(x^k)$ and $r(x^k) = \sqrt{f(x^k)}$ we get

$$p^k = -\frac{2c\sqrt{f(x^k)}}{1 + \lambda c} \frac{1}{1 + \frac{2c}{4f(x^k)(1+\lambda c)} \nabla f(x^k)^\top \boldsymbol{\Sigma}_k^{-1} \nabla f(x^k)} \boldsymbol{\Sigma}_k^{-1} \frac{1}{2\sqrt{f(x^k)}} \nabla f(x^k)$$

$$- \frac{c\lambda}{1+\lambda c} x^k + \frac{\frac{2c^2\lambda}{4f(x^k)(1+\lambda c)} \boldsymbol{\Sigma}_k^{-1} \nabla f(x^k) \nabla f(x^k)^\top x^k}{1 + \frac{2c}{4(1+\lambda c)f(x^k)} \nabla f(x^k)^\top \boldsymbol{\Sigma}_k^{-1} \nabla f(x^k)}$$

$$= -\frac{c/(1+\lambda c)}{1 + \frac{c}{2f(x^k)(1+\lambda c)} \|\nabla f(x^k)\|^2_{\boldsymbol{\Sigma}_k^{-1}}} \boldsymbol{\Sigma}_k \nabla f(x^k) - \frac{c\lambda}{1+\lambda c} x^k$$

$$+ \frac{c\lambda}{1+\lambda c} \frac{\frac{c}{2f(x^k)} \nabla f(x^k)^\top x^k}{1 + \frac{c}{2f(x^k)(1+\lambda c)} \|\nabla f(x^k)\|^2_{\boldsymbol{\Sigma}_k^{-1}}} \boldsymbol{\Sigma}_k^{-1} \nabla f(x^k).$$

To summarize, the update of NGN-Dv1W is the following

$$x^{k+1} = x^k + p^k$$

$$= \frac{1}{1+\lambda c} x^k + \frac{c\lambda}{1+\lambda c} \frac{\frac{c}{2f(x^k)} \nabla f(x^k)^\top x^k}{1 + \frac{c}{2f(x^k)(1+\lambda c)} \|\nabla f(x^k)\|^2_{\boldsymbol{\Sigma}_k^{-1}}} \boldsymbol{\Sigma}_k^{-1} \nabla f(x^k)$$

$$- \frac{c/(1+\lambda c)}{1 + \frac{c}{2f(x^k)(1+\lambda c)} \|\nabla f(x^k)\|^2_{\boldsymbol{\Sigma}_k^{-1}}} \boldsymbol{\Sigma}_k^{-1} \nabla f(x^k)$$

$$= \frac{1}{1+\lambda c} x^k - \frac{\frac{c}{1+\lambda c} \left(1 - \frac{c\lambda}{2f(x^k)} \nabla f(x^k)^\top x^k\right)}{1 + \frac{c}{2f(x^k)(1+\lambda c)} \|\nabla f(x^k)\|^2_{\boldsymbol{\Sigma}_k^{-1}}} \boldsymbol{\Sigma}_k^{-1} \nabla f(x^k). \tag{64}$$

To prevent the step-size next to $\boldsymbol{\Sigma}_k^{-1} \nabla f(x^k)$ from being negative, the final update has the form

$$x^{k+1} = \frac{1}{1+\lambda c} x^k - \frac{\frac{c}{1+\lambda c} \left[ 1 - \frac{c\lambda}{2f(x^k)} \nabla f(x^k)^\top x^k \right]_+}{1 + \frac{c}{2f(x^k)(1+\lambda c)} \|\nabla f(x^k)\|^2_{\boldsymbol{\Sigma}_k^{-1}}} \boldsymbol{\Sigma}_k^{-1} \nabla f(x^k), \tag{65}$$

where $[\cdot]_+ := \max\{\cdot, 0\}$. Now we can add momentum on top and obtain the following update of NGN-MDv1W

$$x^{k+1} = \frac{1}{1+\lambda c} x^k - \frac{\frac{c}{1+\lambda c} \left[ 1 - \frac{c\lambda}{2f(x^k)} \nabla f(x^k)^\top x^k \right]_+}{1 + \frac{c}{2f(x^k)(1+\lambda c)} \|\nabla f(x^k)\|^2_{\boldsymbol{\Sigma}_k^{-1}}} \boldsymbol{\Sigma}_k^{-1} \nabla f(x^k) + \beta(x^k - x^{k-1}). \tag{66}$$

This combination of NGN-MDv1 and weight decay is summarized in Algorithm 4. We highlight that now the weight decay is incorporated inside the adaptive step-size as well as regularizing the coefficient next to $x^k$.

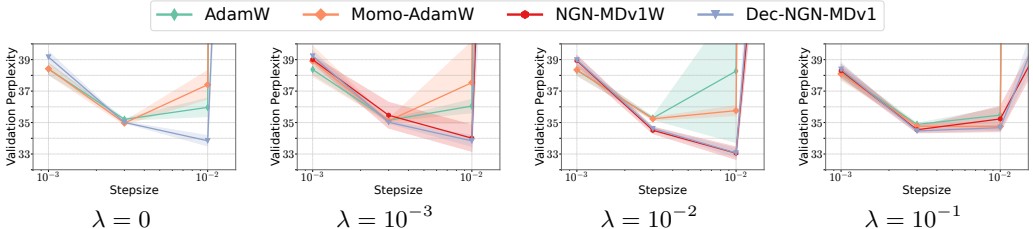

Figure G.1: Adding weight decay when pretraining a 70M Transformer++. When properly tuned, a value of weight decay $> 0$ enhances the performance of all algorithms. NGN-MDv1 retains his characteristic stability, and achieves smaller perplexity in all scenarios.

## G.2 Empirical Validation of the Proposed Combinations

Having two possible ways of adding weight decay to NGN-MDv1, we test them on pretraining a 70M transformer on language modeling. The validation perplexity at the end of training is reported in Figure G.1. We note that when weight decay is turned off, both NGN-MDv1W and Dec-NGN-MDv1 reduce to NGN-MDv1.

First, we observe that when weight decay is properly tuned, all algorithms improve over the baseline case with no weight decay, which is consistent with the observation of Xiao [98] and Andriushchenko et al. [2] on AdamW. We also note that Dec-NGN-MDv1 and NGN-MDv1W require a smaller weight decay value compared to the other algorithms. Finally, the stability and performance of NGNMDv1 are preserved by both variations, allowing training with larger LR, and significantly improving over AdamW and Momo-Adam.

We do not observe a substantial difference between the two proposed modifications of NGN-MDv1 for this task. We remark however that these two versions serve substantially different purposes, and pretraining language models might not be the most representative task to evaluate the effect of adding regularization.

## H  Additional Experiments on Toy Problems

### H.1  Additional Experiments on the Problem with Many Minima

Now, we provide a simple example of minimizing a function

$$f(x) = (\sin(1 + \cos(-\pi + x)) - 0.2x)^2 + (\sin(1 + \cos(\pi - x)) + 0.2x)^4 \qquad (67)$$

that has many sharp sub-optimal local and flat global minima. We compare the performance of NGN-M and SGDM varying the step-size hyperparameter in $\{10^0, 10^1, 10^2, 10^3\}$ and the starting point in $[-20, 20]$ with a step $4/30$[6]. Based on the results in Figure H.1 (right), we conclude that $(i)$ for small step-sizes, both methods likely get stuck at sub-optimal local minima and reach the global minima only if they are initialized close enough to it; $(ii)$ for large step-sizes, we observe less runs of SGDM reaching the global minima; $(iii)$ in contrast, for NGN-M with large step-sizes, we observe more runs reaching the global minima. This is possible due to the adaptive nature of the NGN step-size that forces NGN-M to converge to the flatness of the global minima.

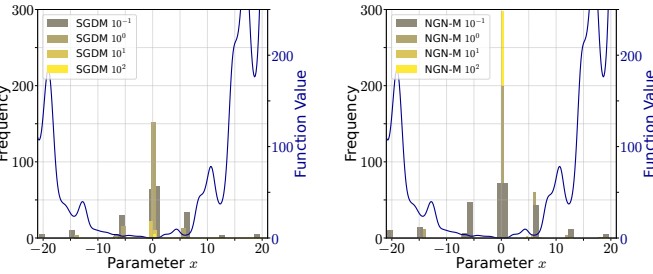

Figure H.1: Comparison of SGDM and NGN-M when minimizing function in (67).

### H.2  Comparison on Rosenbrock Function

Now we present the results where we compare NGN-M and SGDM when minimizing the Rosenbrock function. We report the trajectories of optimizers and training dynamics in Figure H.2 and Figure H.3.

We observe that NGN-M converges for all values of $c$, indicating its high resilience to the choice of step-size hyperparameter. In contrast, SGDM already diverges for the step-size hyperparameter $10^{-2}$. This can be explained by the adaptive nature of NGN step-size, which decreases the effective step-size of NGN-M for a more stable convergence. This is especially evident from the trajectories of algorithms. Indeed, NGN-M effectively moves in the complex valley of the Rosenbrock function, adapting to the local curvature.

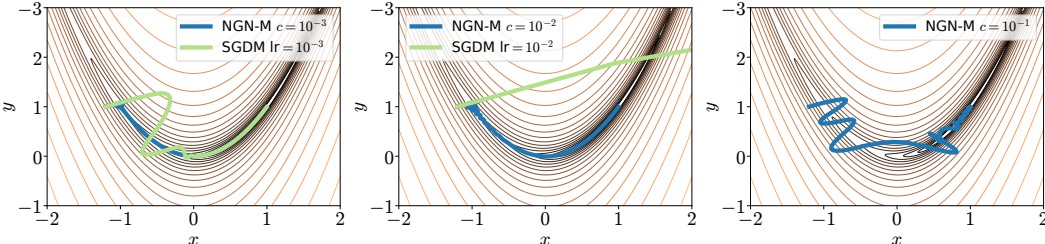

Figure H.2: Trajectories of NGN-M and SGDM when minimizing the Rosenbrock function and varying the step-size hyperparameter.

### H.3  Comparison on Quadratic Function with Theoretical Step-size

Next, we run NGN-M with theoretical choice of step-size hyperparameter $c = 1/\sqrt{K}$ and $c_k = 1/\sqrt{k}$ (see Theorem 4.3 and Theorem D.5 for more details) against fixed choices $c \in \{10^{-3}, 10^{-4}\}$. The

---

[6]This step is chosen small enough so that the initial point can be close to any local minima within $[-20, 20]$.

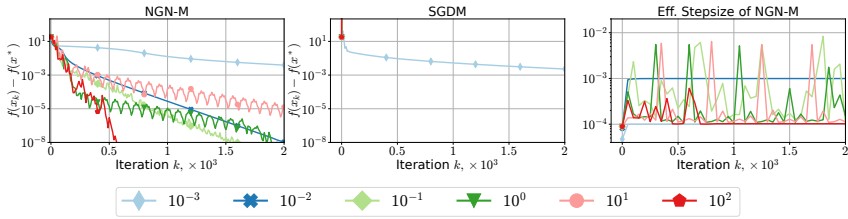

Figure H.3: Training dynamics of NGN-M and SGDM when minimizing the Rosenbrock function and varying the step-size hyperparameter.

comparison is made on quadratic function $f(x) = \frac{1}{2}\|(\mathbf{A} + r\mathbf{I})x - y\|^2$, where $\mathbf{A} \in \mathbb{R}^{400 \times 400}$ and $y \in \mathbb{R}^{400}$ are sampled from standard normal distribution. The constant $r$ controls the condition number of the problem.

We test the performance of NGN-M varying the condition number of the problem and the number of iterations; see Figure H.4. We observe that in all the cases, the choice $1/\sqrt{k}$ leads to faster convergence, supporting our theoretical claims. The choice $1/\sqrt{K}$ demonstrates competitive performance as well, but it is slightly pessimistic at the beginning of training. In contrast, the choice $c \in \{10^{-3}, 10^{-4}\}$, which is a default value in practice, is too small and does not lead to fast convergence.

These experiments demonstrate that when the problem satisfies all assumptions needed in the analysis, the choice of the step-size hyperparameter $c$ given by the convergence theorems is a good starting point in practice and can serve as a baseline when tuning $c$.

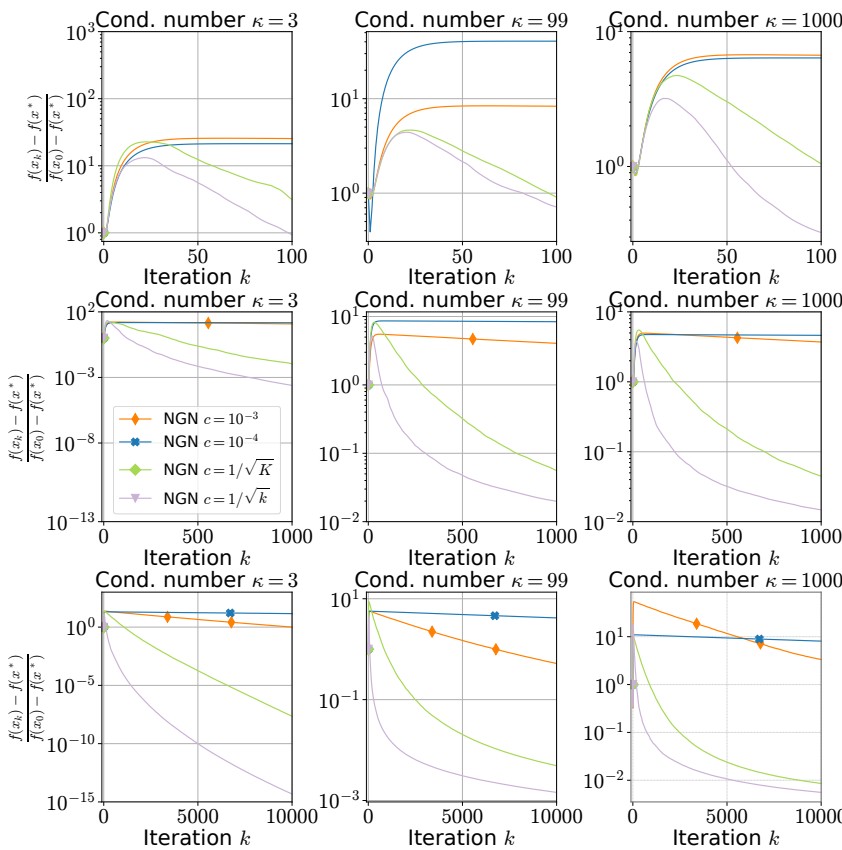

Figure H.4: Training dynamics of NGN-M with several choices of the step-size hyperparameter varying the condition number of the quadratic problem.

# I  Additional Experiments and Training Details

## I.1  Training Details

The detailed experiment setup with hyperparameters and training details is presented in Table 6. We provide links to the exact model architectures used in our experiments (the links are clickable) as well as links to the tables and figures for each workload. We demonstrate the results averaged across 3 different random seeds for small and middle-range size experiments. We use standard values of momentum parameters $(\beta_1, \beta_2) = (0.9, 0.999)$ if the opposite is not specified. The step-size hyperparameter is tuned across powers of 10 (for some workloads we add additional values of the step-size hyperparameter shown in the step-size resilience plots). We use PyTorch [69] implementation of Adam. The implementation of MomSPS, Momo, Momo-Adam are provided in the corresponding papers. Finally, when employing SGD-M, we set dampening equal to 0.9.

For vision transformers experiments, we follow the setup of Schaipp et al. [79], and use Pytorch Image Models codebase [95]. We train a `vit_tiny_patch16_224` for 200 epochs on Imagenet1k, using a cosine LR schedule with a linear warmup of 5 epochs. Differently than Schaipp et al. [79], we train in `bfloat16`, instead of `float16`, and do not employ weight decay regularization.

For pre-training Transformers on Causal Language Modeling, we build upon the nanoGPT [43] implementation, augmenting it with Rotational Positional Embedding [87], RMSNorm [101], and SwiGLU [82]. We call this enhanced version Transformer++. Models are trained with a batch size of 256, context length of 2048 tokens, vocabolary size of 50280 and make use of GPT-Neox tokenizer [5]. We adopt an enhanced training recipe, made popular by large language models such as LLaMa [89]. These modifications include: training in `bfloat16`; employing a linear LR warm-up for 10% of the training steps, followed by cosine annealing to $10^{-5}$; omitting biases from linear layers; using $(\beta_1, \beta_2) = (0.9, 0.95)$ for all algorithms; clipping gradient norms above 1; no weight tying between embedding and last linear layer. All models are trained on SlimPajama-627B [86], a cleaned and deduplicated version of RedPajama We report validation perplexity on a separate subset of Slim-Pajama consisting of 10M tokens. The total compute is estimated following Kaplan et al. [41], where the estimated number of floating-point operations (FLOPs) is 6 × Number of Parameters × Number of Tokens.

Experiments of small and middle size are performed on 1xRTX 4090. We perform ImageNet32 experiments on 2xA100-40GB, and ImageNet1k experiments on 4xA100-SXM4-40GB. For pretraining Transformers on Language Modeling, we employ 8xH100-HBM3-80GB GPUs. With multiple devices in use, we employ Distributed Data Parallel to parallelize the training process.

## I.2  Comparison Algorithms that Support Momentum

In the main paper, we provided the test performance only. Now we additionally illustrate the performance of algorithms w.r.t. training loss convergence. Figure I.1 demonstrates that NGN-M is the most robust algorithm for the choice of the step-size hyperparameter from this perspective as well. In Figure I.1, we additionally demonstrate the performance of the algorithms on (VGG16 [84], CIFAR10) and (MLP, MNIST) workloads where NGN-M matches the performance of the state-of-the-art algorithms in this setting and archives higher resilience to the step-size hyperparameter choice. The best performance results are reported in Table 7 and showcase that NGN-M always matches the performance of other optimizers or improves it.

## I.3  Comparison of Algorithms that Support Momentum and Diagonal Step-size

Next, we illustrate the performance of the algorithms that support both momentum and diagonal step-size. According to the results in Figures I.2 and I.3, NGN-MDv1 achieves the best resilience to the step-size hyperparameter choice among all considered algorithms. Again, NGN-MDv1 is the most stable algorithm to the choice of step-size hyperparameter w.r.t. training loss convergence. Its best performance is competitive to that of other algorithms but the step-size hyperparameter range that gives such performance is wider.

Moreover, we support our claims about stability on additional workloads such as (VGG16, CIFAR10) (in Figure I.1), (MLP, MNIST), (LSTM [32], PTB [59]), and (Transformer [43], Tiny Shakespeare [42]) workloads. We observe that NGN-MDv1 attains higher robustness to the choice of the step-size

Table 6: Summary of experiment setup with all the details on hyperparameters used in each case.

| Model | Dataset | Performance Results | Stability Results | Effective Stepsize Results | Epochs / Iterations | Batch Size | Comments |
|---|---|---|---|---|---|---|---|
| Resnet20 | CIFAR10 | Tab. 7, 8, 9 | Fig. 2, I.1, I.2, I.5 | Fig. I.9, I.10, I.6 | 50 | 128 | |
| Resnet110 | CIFAR100 | Tab. 7, 8 | Fig. 2, I.1, I.2, I.5 | | 100 | 128 | |
| VGG16 | CIFAR10 | Tab. 7, 8 | Fig. I.1, I.2 | | 50 | 128 | |
| MLP | MNIST | Tab. 7, 8 | Fig. I.1, I.3 | | 10 | 128 | 2 hidden layers of size 100 |
| ViT | CIFAR10 | Tab. 7, 8 | Fig. 2, I.1, I.2, I.5 | Fig. 5, I.9, I.10, I.7 | 200 | 512 | |
| LSTM | PTB | Tab. 8, 9 | Fig. I.3 | | 150 | 20 | # layers 3 |
| LSTM | Wikitext-2 | Tab. 8, 9 | Fig. I.8 | | 150 | 20 | # layers 3 |
| Transformer | Rotten Tomatoes | Tab. 8, 9 | Tab. I.8 | | 2000 | 16 | # heads 8 # layers 24 |
| Transformer | Tiny Shakespeare | Tab. 8, 9 | Fig. I.3, I.8 | | 2000 | 16 | # heads 8 # layers 24 |
| Resnet18 | ImageNet32 | Tab. 7, 8, | Fig. I.4 | | 45 | 128 | constant learning rate schedule; no weight decay |
| Resnet18 | ImageNet1k | Tab. 7, 8 | Fig. 2, I.4 | | 90 | 256 | learning rate decay every 30 epochs by 0.1 no weight decay |
| ViT-Tiny | ImageNet1k | Tab. 8 | Fig. 3 | | 200 | 512 | cosine learning rate schedule with linear warm-up for 5 epochs no weight decay, bfloat16 |
| 70M Transformer++ | SlimPajama-627B | Tab. 8, 5 | Fig. 4, G.1, I.14 | | 2400 | 256 | dim=512, # heads 8 # layers 6, context length 2048 $(\beta_1, \beta_2) = (0.9, 0.95)$, bfloat16 clipping norm 1, linear warm-up for 10% of iterations |
| 70M Transformer++ | FineWeb | Tab. 10, 11, 12, 13 | | | 4800 | 128 | dim=512, # heads 8 # layers 6, context length 2048 $(\beta_1, \beta_2) = (0.9, 0.95)$, bfloat16 clipping norm 1, linear warm-up for 10% of iterations |
| 160M Transformer++ | SlimPajama-627B | Tab. 8, 5 | Fig. 4, I.14 | Fig. I.11, I.12, I.13 | 4800 | 256 | dim=768, # heads 12 # layers 12, context length 2048 $(\beta_1, \beta_2) = (0.9, 0.95)$, bfloat16 clipping norm 1, linear warm-up for 10% of iterations |
| 410M Transformer++ | SlimPajama-627B | Tab. 8, 5 | Fig. 4, I.14 | | 13500 | 256 | dim=1024, # heads 16 # layers 24, context length 2048 $(\beta_1, \beta_2) = (0.9, 0.95)$, bfloat16 clipping norm 1, linear warm-up for 10% of iterations |
| 1B Transformer++ | SlimPajama-627B | Tab. 8 | Fig. 4, I.14 | | 13500 | 256 | dim=2048, # heads 8 # layers 16, context length 2048 $(\beta_1, \beta_2) = (0.9, 0.95)$, bfloat16 clipping norm 1, linear warm-up for 10% of iterations |

Table 7: The best validation score (with one standard deviation across 3 runs; accuracy for computer vision tasks; perplexity for NLP tasks) for the best learning rate choice for each method that supports momentum.

| Model | Dataset | NGN | SGDM | NGN-M | MomSPS | Momo | ALR-SMAG |
|---|---|---|---|---|---|---|---|
| Resnet20 | CIFAR10 | $88.30_{\pm 0.20}$ | $85.42_{\pm 0.70}$ | $88.76_{\pm 0.05}$ | $87.20_{\pm 0.38}$ | $88.86_{\pm 0.14}$ | $88.88_{\pm 0.19}$ |
| Resnet110 | CIFAR100 | $64.76_{\pm 0.26}$ | $57.16_{\pm 2.06}$ | $64.98_{\pm 0.29}$ | $63.37_{\pm 0.71}$ | $64.81_{\pm 0.33}$ | $64.73_{\pm 1.81}$ |
| VGG16 | CIFAR10 | $90.21_{\pm 0.10}$ | $89.67_{\pm 0.43}$ | $90.42_{\pm 0.06}$ | $87.26_{\pm 0.21}$ | $90.43_{\pm 0.17}$ | $90.49_{\pm 0.35}$ |
| MLP | MNIST | $98.04_{\pm 0.07}$ | $97.63_{\pm 0.10}$ | $97.97_{\pm 0.08}$ | $97.73_{\pm 0.09}$ | $97.97_{\pm 0.04}$ | $97.64_{\pm 0.06}$ |
| ViT | CIFAR10 | $83.34_{\pm 0.24}$ | $83.74_{\pm 0.11}$ | $84.95_{\pm 0.29}$ | $83.77_{\pm 0.27}$ | $85.47_{\pm 0.27}$ | $85.54_{\pm 0.39}$ |
| Resnet18 | ImageNet32 | 48.63 | 48.56 | 48.29 | N/A | 48.68 | N/A |
| Resnet18 | ImageNet1k | 67.00 | 66.73 | 67.12 | N/A | 67.09 | N/A |
| Transformer | Tiny Shakespeare | $9.27_{\pm 0.19}$ | $8.73_{\pm 0.13}$ | $7.67_{\pm 0.12}$ | N/A | $8.80_{\pm 0.19}$ | N/A |
| Transformer | Rotten Tomatoes | $9.01_{\pm 0.22}$ | $8.75_{\pm 0.04}$ | $7.12_{\pm 0.03}$ | N/A | $8.65_{\pm 0.03}$ | N/A |
| LSTM | Wikitext-2 | $75.33_{\pm 0.15}$ | $82.07_{\pm 0.16}$ | $75.51_{\pm 0.22}$ | N/A | $76.09_{\pm 0.40}$ | N/A |

hyperparameter. Finally, the performance results on (LSTM, Wikitext-2 [58]) and (Transformer, Rotten Tomatoes [68]) are reported in Table 8. The results demonstrate competitive performance of NGN-MDv1 against other benchmarks across all considered workloads.

## I.4  Additional ImageNet Experiments

Now we turn to the experiments involving training Resnet18 on ImageNet1k and ImageNet32. In Figure I.4 we provide the train loss curves and results on (Resnet18, ImageNet32) workload that demonstrate that NGN-M and NDN-MDv1 attain better resilience to the step-size hyperparameter

Table 8: The best validation score (with one standard deviation; accuracy for computer vision tasks; perplexity for NLP tasks) for the best learning rate choice for each method that supports diagonal step-sizes and momentum.

| Model | Dataset | Adam | Momo-Adam | NGN-MDv1 | NGN-MDv2 | Lion | Adabelief | Adabound |
|---|---|---|---|---|---|---|---|---|
| Resnet20 | CIFAR10 | $86.96_{\pm0.70}$ | $89.41_{\pm0.36}$ | $89.53_{\pm0.11}$ | $87.80_{\pm0.16}$ | $88.09_{\pm0.27}$ | $87.47_{\pm0.48}$ | $85.00_{\pm0.56}$ |
| Resnet110 | CIFAR100 | $64.12_{\pm0.94}$ | $67.10_{\pm0.53}$ | $66.10_{\pm0.45}$ | $64.33_{\pm0.40}$ | $61.85_{\pm0.77}$ | $65.32_{\pm0.43}$ | $61.28_{\pm0.39}$ |
| VGG16 | CIFAR10 | $90.26_{\pm0.23}$ | $90.95_{\pm0.28}$ | $90.64_{\pm0.18}$ | $90.07_{\pm0.37}$ | N/A | N/A | N/A |
| MLP | MNIST | $97.44_{\pm0.19}$ | $97.96_{\pm0.10}$ | $98.10_{\pm0.06}$ | $97.67_{\pm0.17}$ | N/A | N/A | N/A |
| ViT | CIFAR10 | $85.96_{\pm0.23}$ | $85.74_{\pm0.12}$ | $85.65_{\pm0.10}$ | $86.56_{\pm0.11}$ | $86.89_{\pm0.19}$ | $85.05_{\pm0.47}$ | $80.32_{\pm0.47}$ |
| Transformer | Rotten Tomatoes | $6.80_{\pm0.07}$ | $6.81_{\pm0.05}$ | $6.90_{\pm0.05}$ | $6.83_{\pm0.05}$ | N/A | N/A | N/A |
| Transformer | Tiny Shakespeare | $6.80_{\pm0.06}$ | $6.80_{\pm0.05}$ | $6.89_{\pm0.06}$ | $6.82_{\pm0.05}$ | N/A | N/A | N/A |
| LSTM | PTB | $70.95_{\pm0.08}$ | $71.09_{\pm0.05}$ | $70.84_{\pm0.20}$ | $71.37_{\pm0.17}$ | N/A | N/A | N/A |
| LSTM | Wikitext-2 | $81.49_{\pm1.49}$ | $82.23_{\pm0.64}$ | $75.24_{\pm0.21}$ | $81.99_{\pm0.78}$ | N/A | N/A | N/A |
| Resnet18 | ImageNet32 | 48.11 | 48.09 | 48.06 | 47.55 | N/A | N/A | N/A |
| Resnet18 | ImageNet1k | 67.17 | 67.06 | 67.15 | 67.32 | N/A | N/A | N/A |
| ViT-Tiny | ImageNet1k | $71.05_{\pm0.16}$ | $71.22_{\pm0.36}$ | $71.345_{\pm0.22}$ | N/A | N/A | N/A | N/A |
| Transformer++ 70M | SlimPajama-627B | $34.38_{\pm0.12}$ | $34.96_{\pm0.11}$ | $33.84_{\pm0.33}$ | N/A | N/A | N/A | N/A |
| Transformer++ 160M | SlimPajama-627B | $24.03_{\pm0.02}$ | $24.29_{\pm0.10}$ | $23.32_{\pm0.06}$ | N/A | N/A | N/A | N/A |
| Transformer++ 410M | SlimPajama-627B | $16.65_{\pm0.03}$ | $17.07_{\pm0.05}$ | $16.48_{\pm0.03}$ | N/A | N/A | N/A | N/A |
| Transformer++ 1B | SlimPajama-627B | 13.09 | N/A | 13.11 | N/A | N/A | N/A | N/A |

choice than competitors not only from the train loss point of view as well. The best performance of algorithms is provided in Table 7 and 8. According to them, both NGN-M and NGN-M achieve competitive performance against considered benchmarks.

## I.5 Additional Comparison against Lion, Adabelief, Adabound

This section compares algorithms from Section 5. Moreover, we include the comparison against Lion [9], Adabound [56], and Adabelief [109]. The results are presented in Table 8.

We observe that NGN-MDv1 and NGN-MDv2 both achieve competitive performance across various Deep Learning workloads. In Figure I.5, we observe that Lion, Adabound and Adabelief algorithms do not match always the performance of NGN-MDv1 and Adam: Adabelief has worse performance on (Resnet20, CIFAR10) workload; Adabound has worse performance on (Resnet20, CIFAR10), (Resnet110, CIFAR100), and (ViT, CIFAR10) workloads; Lion has worse performance on (Resnet110, CIFAR100) workload. Moreover, their resilience to the step-size hyperparameter choice is lower than that of NGN-MDv1. To summarize, NGN-M and NGN-MDv1 are the most robust algorithms to the choice of step-size hyperparameter.

## I.6 Comparison of Adaptive Step-sizes of Adam, Momo-Adam, and NGN-MDv1

Next, we conduct experiments to compare the adaptive step-size of Adam, Momo-Adam, and NGN-MDv1. Note that ResNet20 model consists of 3 base blocks, and each block has 3 convolution layers. In Figure I.6 we plot the average adaptive step-size of the layers $j \in$ {layer1.0.conv1, layer2.0.conv1, layer3.0.conv1} of ResNet20 that corresponds to the first convolution layer within each base block. Similarly, in Figure I.7 we plot the average adaptive step-size of the layers $j \in$ {layer0.0.fn.to_qkv, layer3.0.fn.to_qkv, layer5.0.fn.to_qkv} that corresponds to the attention layers of the first, fourth, and sixth base blocks.

Since the adaptivity of Adam is only in the second-order momentum applied as a normalization, in our experiment we compare the following quantities

$$\frac{\gamma}{(\mathbf{D}_k)_{(j)}} \text{ for Adam}, \qquad \frac{\tau_k}{(\mathbf{D}_k)_{(j)}} \text{ for Momo-Adam}, \qquad \frac{\gamma_k}{(\mathbf{D}_k)_{(j)}} \text{ for NGN-MDv1}, \qquad (68)$$

where $\gamma$ is the step-size hyperparameter of Adam.

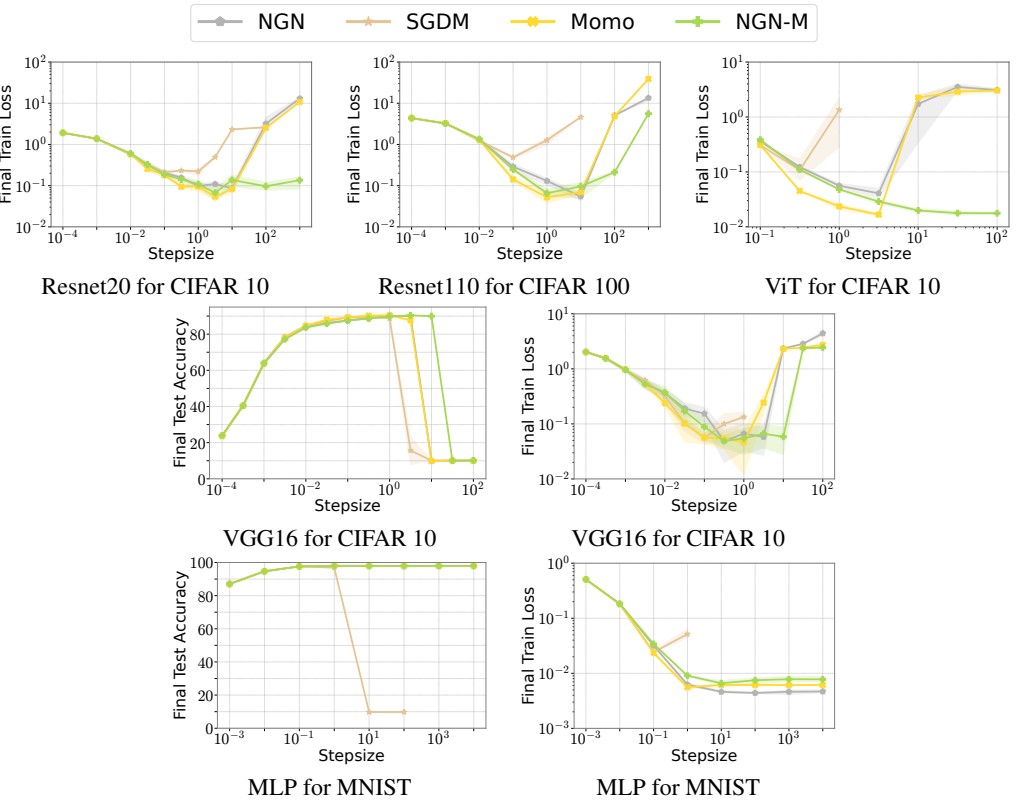

Figure I.1: Stability performance of algorithms supporting momentum varying step-size hyper-arameter ($c$ for NGN and NGN-M, $\alpha_0$ for Momo, and step-size for SGDM). We observe that NGN-M achieves the training loss close to the best possible for a wider range of the step-size hyperparameter.

Let us first describe the results for ResNet20 in Figure I.6. We observe that NGN-MDv1 tends to set smaller effective step-size compared to two other algorithms. This is especially visible for the large step-size hyperparameter values where the adaptive step-size of NGN-MDv1 is by several orders in magnitude smaller than that of Adam and Momo-Adam. In contrast, the coordinate-wise adaptive step-size of Momo-Adam is mostly follow that of Adam. Considering that the stability performance of NGN-MDv1 is much higher for this task, this happens mainly due to the fact that the adaptation mechanism of NGN-MDv1 step-size is more conservative than that of Momo-Adam.

Now we switch to the results on ViT model in Figure I.7. Here both Momo-Adam and NGN-MDv1 tend to utilize smaller effective coordinate-wise step-size, by several orders in magnitude smaller than that of Adam. However, the adaptation mechanism of NGN-MDv1 is still more conservative than that of Momo-Adam, especially for large step-size hyperparameters. We also highlight that in this experiment the best performance of NGN-MDv1 is achieved with $c = 10^{-3}$. When we vary the step-size hyperparameter $c$, the effective coordinate-wise step-size does not change dramatically, especially for layers.0.0.fin.to_qkv layer.

## I.7 Extended Comparison of Momentum-based Algorithms on NLP Tasks

We switch to comparison of NGN-M, Momo, NGN, and SGDM on NLP tasks. In particular, we consider the training of Transformer (based on NanoGPT) on the Tiny Shakespeare and Rotten Tomatoes datasets and LSTM on the Wikitext-2 dataset from Appendix I.3. We report the results in Figure I.8 while the best performance is shown in Table 7. First, note that all algorithms do not match the best performance of those that incorporate diagonal step-size and momentum (see Table 8). Such results are expected since the training of NLP models has significantly different coordinate-wise conditioning. Nonetheless, NGN-M algorithm achieves better resilience to the step-

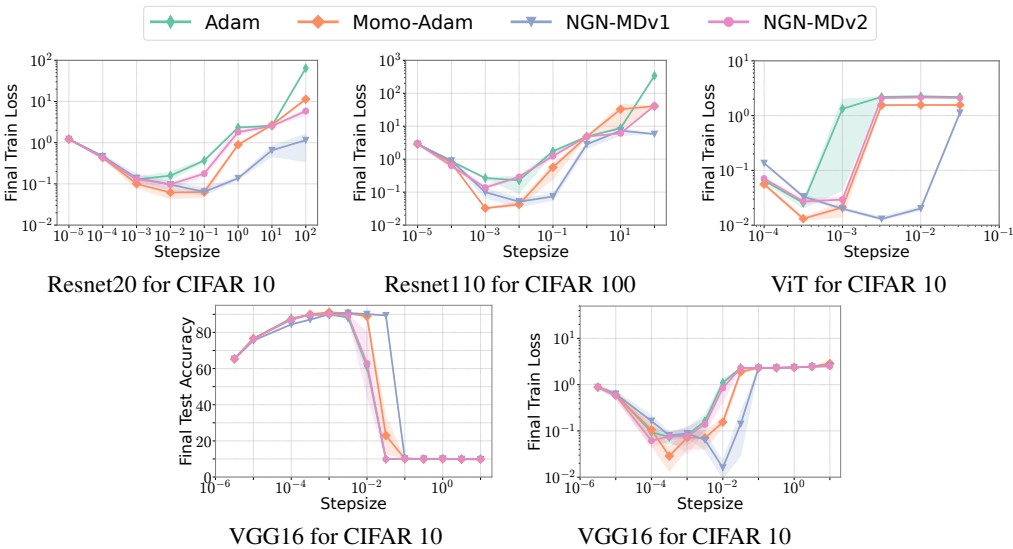

Figure I.2: Stability performance of algorithms supporting momentum and diagonal step-size varying step-size hyperparameter ($c$ for NGN-MDv1 and NGN-MDv2, $\alpha_0$ for Momo-Adam, and step-size for Adam). We observe that NGN-MDv1 achieves the training loss close to the best possible for a wider range of the step-size hyperparameter.

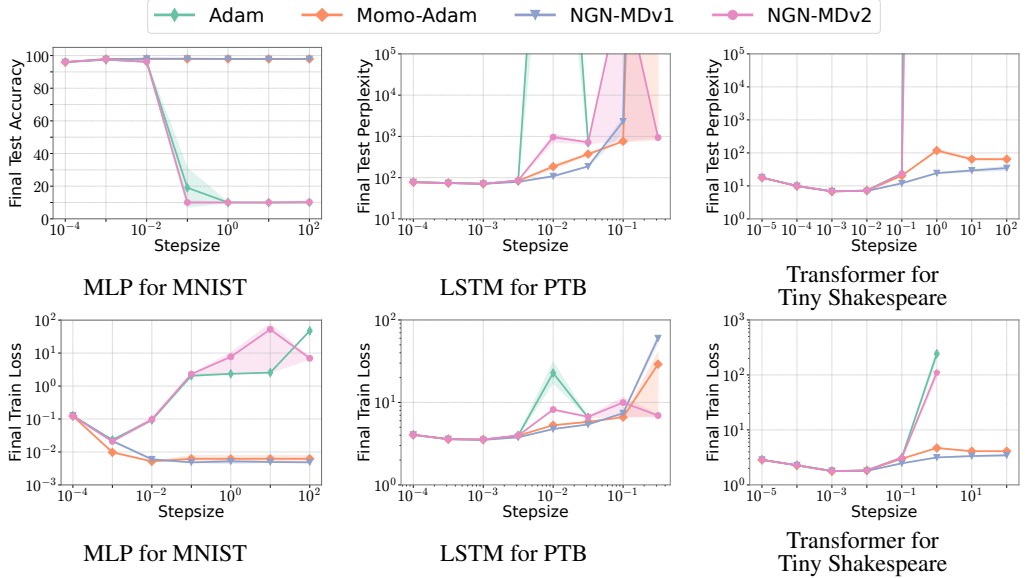

Figure I.3: Stability performance of algorithms supporting momentum and diagonal step-size varying step-size hyperparameter ($c$ for NGN-MDv1 and NGN-MDv2, $\alpha_0$ for Momo-Adam, and step-size for Adam). We observe that NGN-MDv1 achieves the training loss close to the best possible for a wider range of the step-size hyperparameter.

size hyperparameter choice, especially in the training of Transformer models. Therefore, NGN-M across various model architectures and task domains.

## I.8 Comparison of Algorithms with Diagonal Step-size

Now we compare algorithms with diagonal step-size such as NGN-D, Adagrad [18], and RMSprop [44]. Since NGN-D requires to find constants $\{c_j\}_{j=1}^d$ where $d$ is the size of the model. Finding sufficiently good constants $c_j$ might be a challenging task since $d$ is a large number. Therefore, we

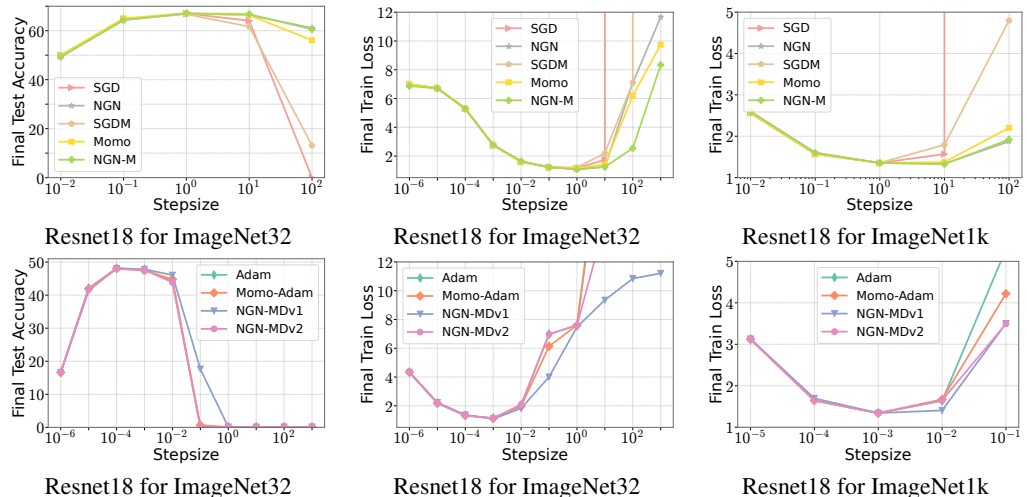

Figure I.4: Stability performance of algorithms supporting momentum (**first row**), and momentum with diagonal step-size (**second row**) varying step-size hyperparameter ($c$ for NGN, NGN-M, NGN-MDv1, and NGN-MDv2, $\alpha_0$ for Momo and Momo-Adam, and step-size for SGD, SGDM, and Adam).

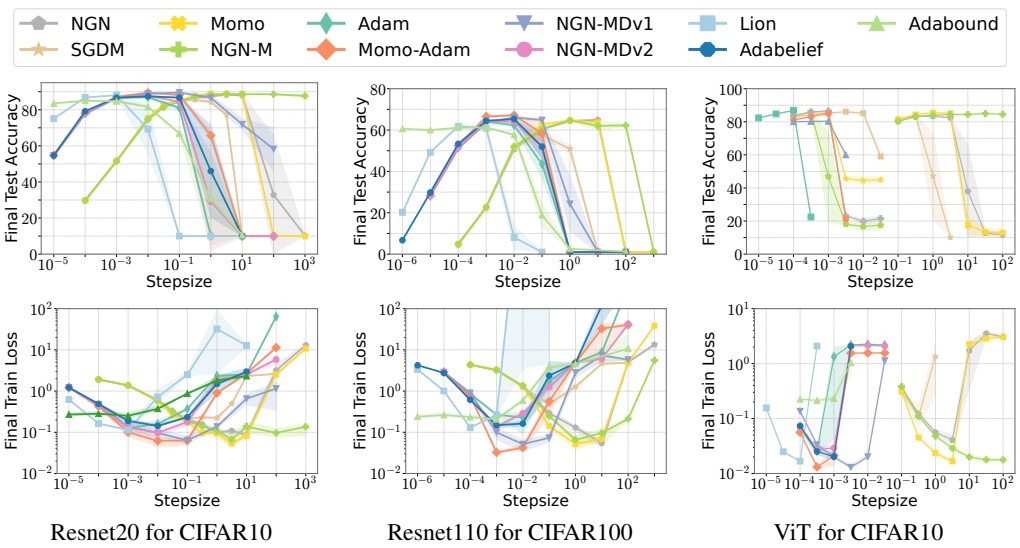

Figure I.5: Stability performance of various optimizers for (Resnet20, CIFAR10), (Resnet110, CIFAR100), (ViT, CIFAR10) workloads.

use RMSprop preconditioner $\mathbf{D}_k$ to set them as $c_j = c/(\mathbf{D}_k)_{(j)}$. We leave the exploration of how to set constants $c_j$ properly for future research.

For each method, we tune its learning rate hyperparameter over the powers of 10: $\{10^{-4}, \ldots, 10^2\}$ and present the best performance averaged across 3 random seeds in Table 9. We observe that NGN-D performs similarly to RMSprop. NGN-D has slightly worse performance on (LSTM, PTB) dataset but significantly better on (LSTM, Wikitext-2) workload. Besides, Adagrad always has the worst performance. Moreover, these algorithms do not have high resilience to the choice of hyperparameter. Therefore, we omit their comparison from this perspective.

### I.9 Effective Step-size of NGN-M, Momo, NGN-MDv1, and Momo-Adam

Next, we compare the effective step-size applied throughout the training with NGN-M, Momo, NGN-MDv1, and Momo-Adam in Figures I.9 and I.10. First, both NGN-M and Momo perform a warm-up

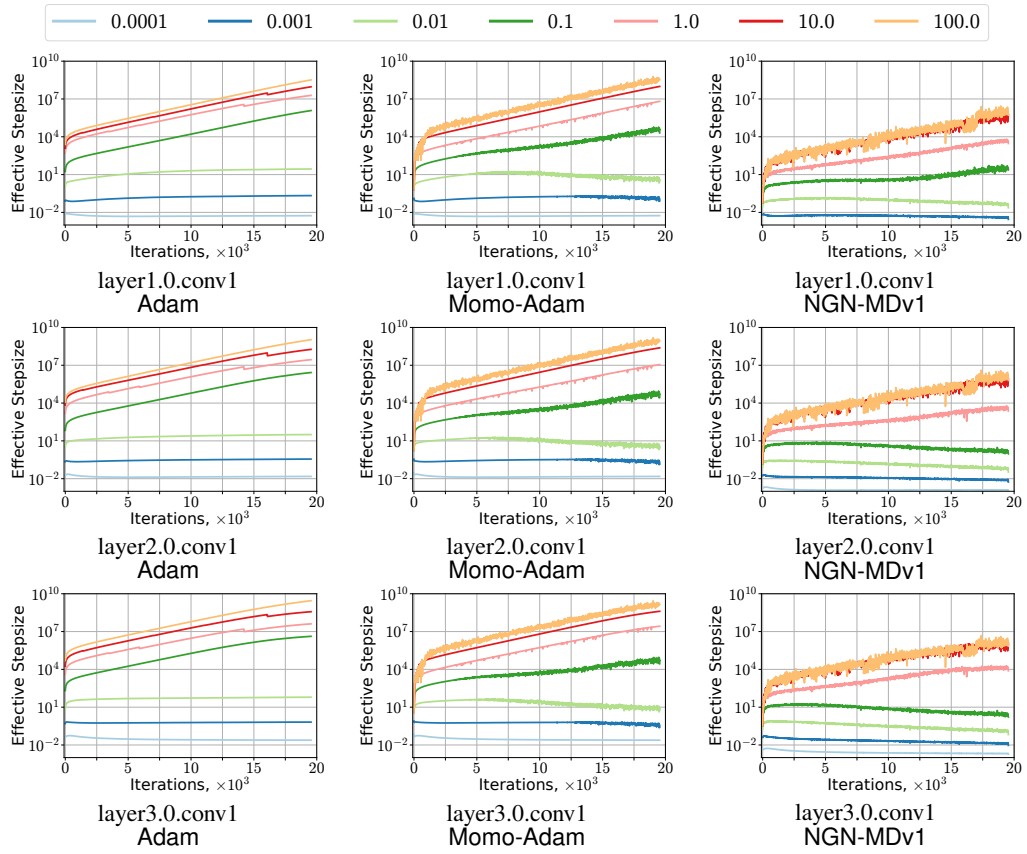

Figure I.6: The adaptive stepsize of Adam (**first column**), Momo-Adam (**second column**), and NGN-MDv1 (**third column**) algorithms in training ResNet20 model on CIFAR10 dataset. We plot the average stepsize $\frac{\gamma}{(\mathbf{D}_k)_{(j)}}$ (for Adam), $\frac{\tau_k}{(\mathbf{D}_k)_{(j)}}$ (for Momo-Adam), and $\frac{\gamma_k}{(\mathbf{D}_k)_{(j)}}$ (for NGN-MDv1) for the first convolution layer within each of 3 base blocks of ResNet20 architecture varying the step-size hyperparameter of the algorithms ($c$ for NGN-M and NGN, $\alpha_0$ for Momo, and learning rate parameter for Adam).

Table 9: The best validation score (with one standard deviation; accuracy for image classification; perplexity for language modeling) for the best learning rate choice for each method that supports diagonal step-sizes.

| Model | Dataset | Adagrad | RMSprop | NGN-D |
|---|---|---|---|---|
| Resnet20 | CIFAR10 | $85.90_{\pm 0.30}$ | $86.71_{\pm 0.64}$ | $86.98_{\pm 0.15}$ |
| Transformer | Rotten Tomatoes | $7.77_{\pm 0.02}$ | $6.87_{\pm 0.05}$ | $6.92_{\pm 0.03}$ |
| Transformer | Tiny Sheaksper | $7.77_{\pm 0.05}$ | $7.00_{\pm 0.13}$ | $6.90_{\pm 0.05}$ |
| LSTM | PTB | $99.24_{\pm 2.13}$ | $69.00_{\pm 0.17}$ | $71.54_{\pm 0.11}$ |
| LSTM | Wikitext-2 | $113.19_{\pm 4.36}$ | $79.48_{\pm 0.45}$ | $75.44_{\pm 0.12}$ |

in the beginning: the effective step-size increases at the beginning of the training. Then we observe the main difference between the two algorithms above: effective step-size of Momo for sufficiently large step-size hyperparameter is not adaptive within some part of the training, it always hits the upper bound. Consequently, during that part of the training Momo reduces to SGDM. In contrast, the effective step-size of NGN-M is always adaptive: it gradually decreases after a short warm-up. This trend is similar to the state-of-the-art learning rate schedulers used in practice. Similar observations can be made in comparison of NGN-MDv1 and Momo-Adam.

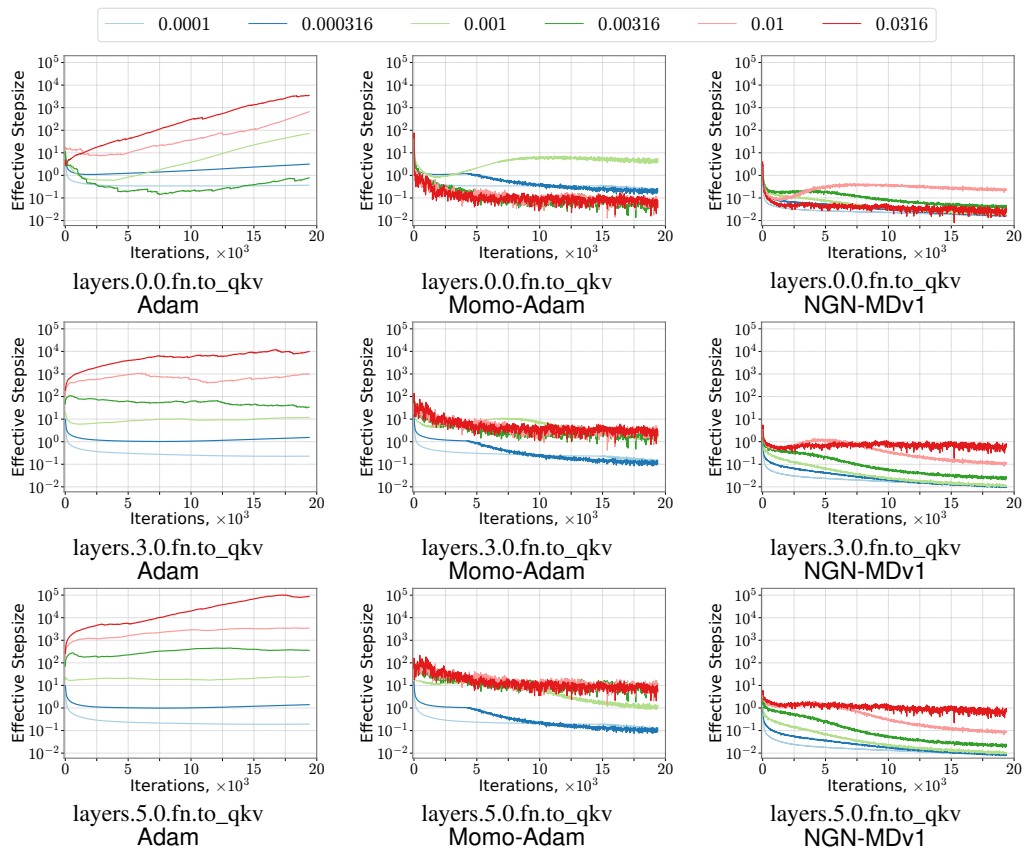

Figure I.7: The adaptive stepsize of Adam (**first column**), Momo-Adam (**second column**), and NGN-MDv1 (**third column**) algorithms in training ViT model on CIFAR10 dataset. We plot the average stepsize $\frac{\gamma}{(\mathbf{D}_k)_{(j)}}$ (for Adam), $\frac{\tau_k}{(\mathbf{D}_k)_{(j)}}$ (for Momo-Adam), and $\frac{\gamma_k}{(\mathbf{D}_k)_{(j)}}$ (for NGN-MDv1) for the attention layer within each of the first, fourth, and sixth base blocks of ViT architecture varying the step-size hyperparameter of the algorithms ($c$ for NGN-M and NGN, $\alpha_0$ for Momo, and learning rate parameter for Adam).

## I.10 Effective Updates in Training Language Models

In this section, we demonstrate the magnitude of updates when training 160M language model with Adam and NGN-MDv1 and varying the step-size hyperparameter across different layers of the model: see the results in Figures I.11 to I.13. We demonstrate that NGN-MDv1 is a more conservative algorithm: the effective update is smaller than that of Adam due to the adaptive nature of the step-size. This is especially evident when training 160M language model with a step-size hyperparameter 0.03: The updates of Adam become considerably larger than the update of NGN-MDv1. This property is a key factor behind the difference in training dynamics: NGN-MDv1 can stabilize at a significantly lower training loss.

## I.11 Training Dynamics in Training Language Models

Now we report the training dynamics in the training language across all tested sizes.

## I.12 Ablation Study of Momentum Parameters

In this section, we study the sensitivity of NGN-MDv1 and Adam to the choice of the learning rate and momentum hyperparameters, when training 70M language model on FineWeb dataset [70]. To do that, we fix $\beta_1 = 0.9$ (or $\beta_2 = 0.95$) and make a sweep over the learning rate hyperparameter and

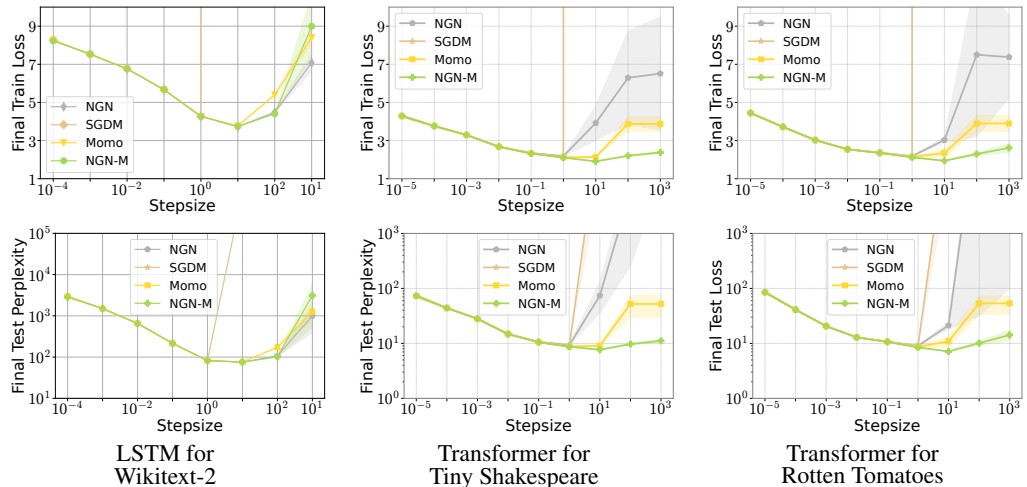

Figure I.8: Stability performance of algorithms supporting momentum and diagonal step-size varying step-size hyperparameter ($c$ for NGN-M and NGN, $\alpha_0$ for Momo, and step-size for SGDM). We observe that NGN-M achieves the training loss close to the best possible for a wider range of the step-size hyperparameter.

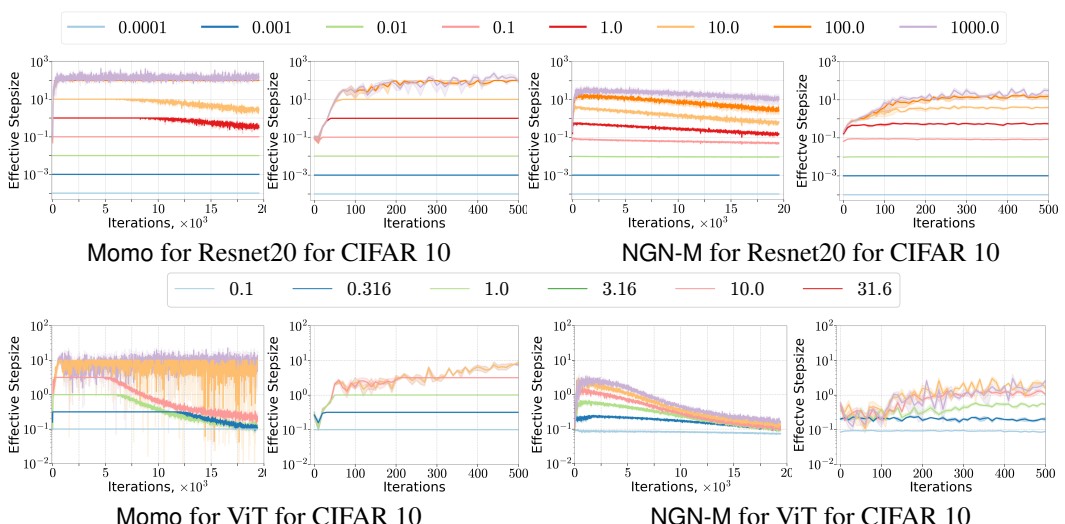

Figure I.9: The step-size of Momo and NGN-M during the training. We demonstrate the step-sizes $\tau_k$ for Momo and $\gamma_k$ for NGN-M varying step-size parameters $\alpha_0$ for Momo and $c$ for NGN-M.

$\beta_2$ (or learning rate hyperparameter and $\beta_1$). We report the final test perplexity averaged over 3 runs for each set of hyperparameters.

We summarize our findings from Table 10, Table 11, Table 12, and Table 13 as follows:

- Low lr (3e-3): NGN-MDv1 and Adam show similar sensitivity to changes in both $\beta_1$ and $\beta_2$.

- Moderate lr (1e-2): NGN-MDv1 is noticeably more robust than Adam to extremes of $\beta_1$, while both optimizers perform similarly across $\beta_2$ (though Adam's performance degrades slightly at $\beta_2 = 0.999$).

- High lr (3e-2): Both methods suffer when $\beta_1$ is small (or $\beta_2$ is large), but NGN-MDv1 recovers lower perplexity at larger $\beta_1$ values (smaller $\beta_2$ values), whereas Adam fails to reach comparable performance.

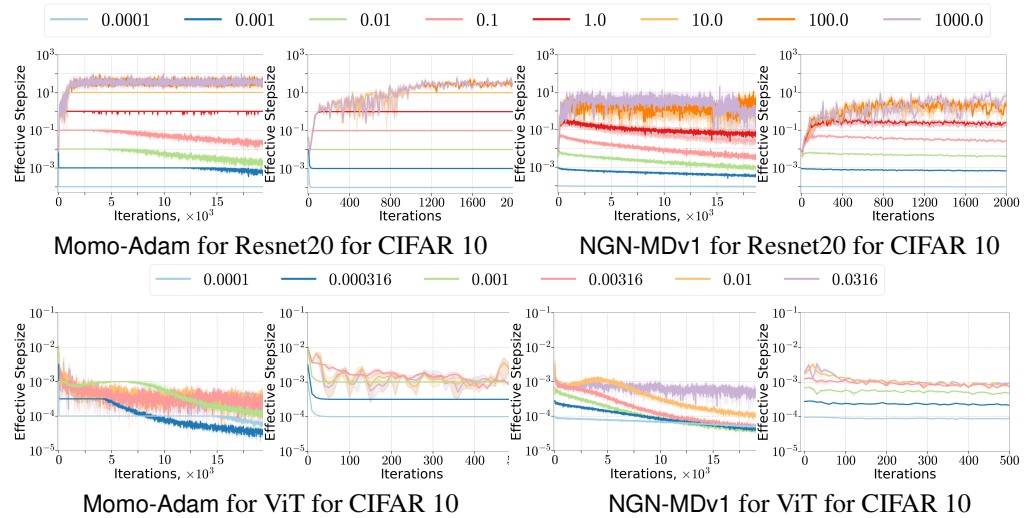

Figure I.10: The step-size of Momo-Adam and NGN-MDv1 during the training. We demonstrate the step-sizes $\tau_k$ for Momo-Adam and $\gamma_k$ for NGN-MDv1 varying step-size parameters $\alpha_0$ for Momo and $c$ for NGN-MDv1.

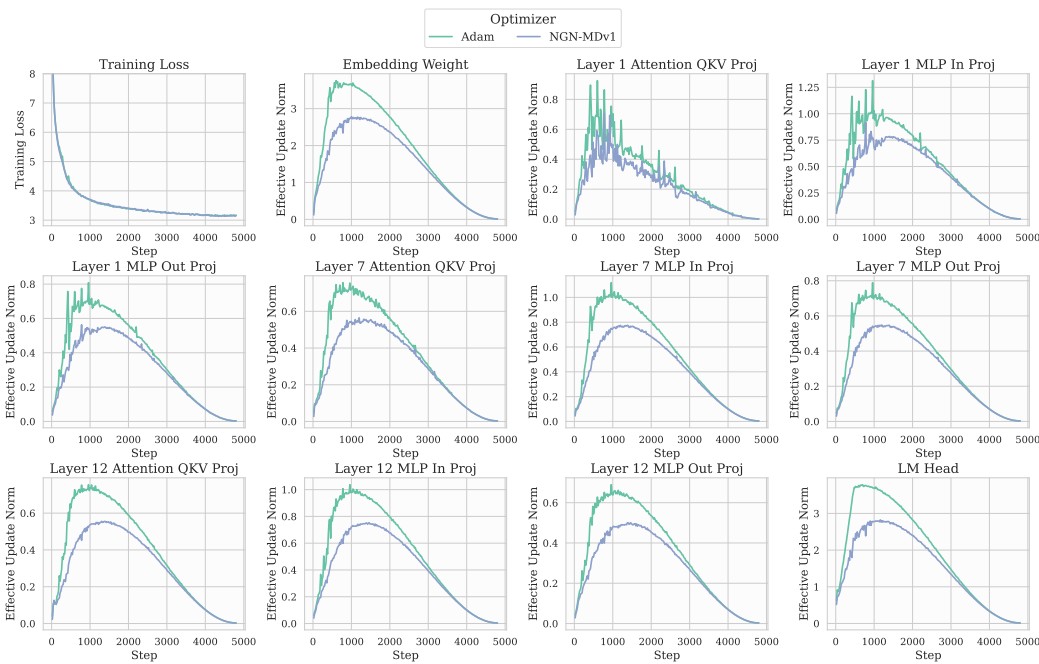

Figure I.11: Magnitude of updates when training 160M language model with Adam and NGN-MDv1 and step-size hyperparameter 0.003.

To conclude, NGN-MDv1 demonstrates greater robustness to changes in momentum parameters at high lr, and consistently attains lower perplexity than Adam, even when both methods' performance deteriorates (we refer to the cases when both algorithms cannot achieve perplexity around 50).

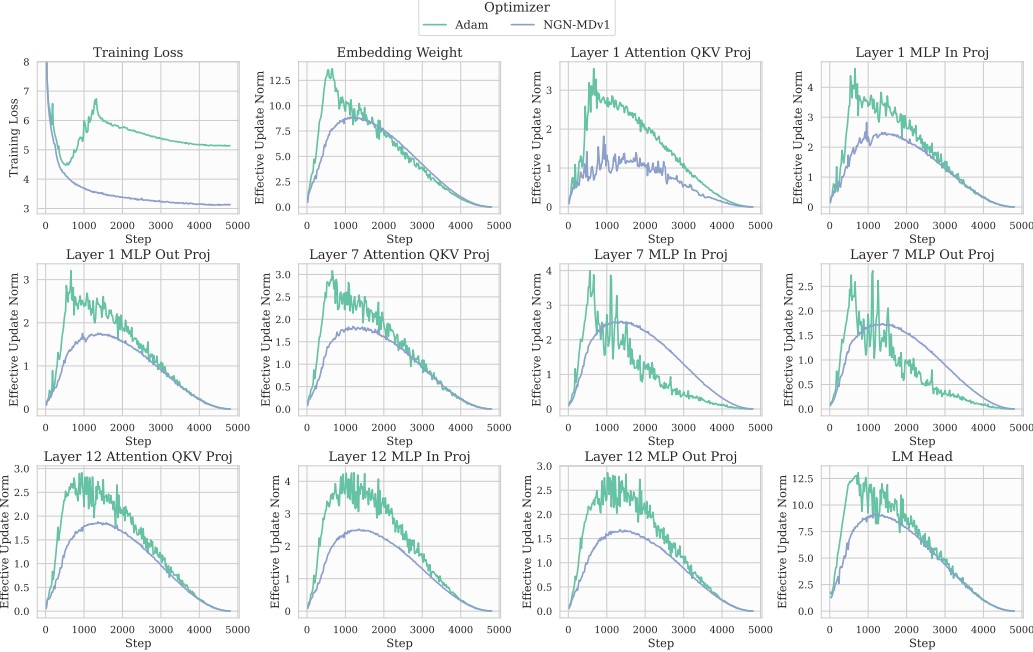

Figure I.12: Magnitude of updates when training 160M language model with Adam and NGN-MDv1 and step-size hyperparameter 0.01.

Table 10: Test perplexity of NGN-MDv1 when varying the learning rate and $\beta_1$ hyperparameters when training 70M language model on the FineWeb dataset.

| lr | $\beta_1 = 0.6$ | $\beta_1 = 0.8$ | $\beta_1 = 0.9$ | $\beta_1 = 0.99$ |
|---|---|---|---|---|
| 3e-4 | $49.9 \pm 0.2$ | $47.4 \pm 0.2$ | $47.0 \pm 0.2$ | $49.7 \pm 0.3$ |
| 1e-3 | $41.5 \pm 0.2$ | $39.9 \pm 0.2$ | $38.6 \pm 0.1$ | $40.2 \pm 0.3$ |
| 3e-3 | $40 \pm 1$ | $36.9 \pm 0.3$ | $35.9 \pm 0.1$ | $37.2 \pm 0.1$ |
| 1e-2 | $54 \pm 16$ | $37 \pm 2$ | $34.7 \pm 0.3$ | $35.9 \pm 0.1$ |
| 3e-2 | $278 \pm 6$ | $129 \pm 2$ | $34.6 \pm 0.1$ | $35.6 \pm 0.1$ |

Table 11: Test perplexity of Adam when varying the learning rate and $\beta_1$ hyperparameters when training 70M language model on the FineWeb dataset.

| lr | $\beta_1 = 0.6$ | $\beta_1 = 0.8$ | $\beta_1 = 0.9$ | $\beta_1 = 0.99$ |
|---|---|---|---|---|
| 3e-4 | $49.4 \pm 0.2$ | $46.5 \pm 0.1$ | $46.2 \pm 0.3$ | $57 \pm 1$ |
| 1e-3 | $41.4 \pm 0.2$ | $39.6 \pm 0.1$ | $38.5 \pm 0.1$ | $45.0 \pm 0.2$ |
| 3e-3 | $40.7 \pm 0.1$ | $37.0 \pm 0.1$ | $36.0 \pm 0.1$ | $220 \pm 70$ |
| 1e-2 | $160 \pm 60$ | $41 \pm 2$ | $36 \pm 2$ | $210 \pm 110$ |
| 3e-2 | $420 \pm 20$ | $340 \pm 50$ | $320 \pm 60$ | $330 \pm 130$ |

Table 12: Test perplexity of NGN-MDv1 when varying the learning rate and $\beta_2$ hyperparameters when training 70M language model on the FineWeb dataset.

| lr | $\beta_2 = 0.6$ | $\beta_2 = 0.8$ | $\beta_2 = 0.9$ | $\beta_2 = 0.95$ | $\beta_2 = 0.999$ |
|---|---|---|---|---|---|
| 3e-4 | $51.8 \pm 0.6$ | $49.2 \pm 0.4$ | $47.8 \pm 0.3$ | $47.0 \pm 0.2$ | $47.0 \pm 0.2$ |
| 1e-3 | $42.6 \pm 0.3$ | $40.5 \pm 0.1$ | $39.3 \pm 0.2$ | $38.6 \pm 0.1$ | $38.9 \pm 0.1$ |
| 3e-3 | $39.4 \pm 0.2$ | $37.5 \pm 0.2$ | $36.3 \pm 0.1$ | $35.9 \pm 0.1$ | $36.5 \pm 0.4$ |
| 1e-2 | $37.8 \pm 0.2$ | $35.9 \pm 0.1$ | $35.1 \pm 0.3$ | $34.7 \pm 0.3$ | $35.0 \pm 0.3$ |
| 3e-2 | $37.8 \pm 0.3$ | $35.8 \pm 0.1$ | $34.9 \pm 0.1$ | $34.6 \pm 0.1$ | $250 \pm 50$ |

Table 13: Test perplexity of Adam when varying the learning rate and $\beta_2$ hyperparameters when training 70M language model on the FineWeb dataset.

| lr | $\beta_2 = 0.6$ | $\beta_2 = 0.8$ | $\beta_2 = 0.9$ | $\beta_2 = 0.95$ | $\beta_2 = 0.999$ |
|---|---|---|---|---|---|
| 3e-4 | $46.1 \pm 0.2$ | $46.6 \pm 0.1$ | $46.5 \pm 0.2$ | $46.2 \pm 0.3$ | $46.5 \pm 0.1$ |
| 1e-3 | $38.8 \pm 0.1$ | $39.0 \pm 0.2$ | $38.9 \pm 0.1$ | $38.5 \pm 0.1$ | $39.5 \pm 0.6$ |
| 3e-3 | $38.8 \pm 0.3$ | $36.3 \pm 0.1$ | $36.1 \pm 0.2$ | $36.0 \pm 0.1$ | $36.7 \pm 0.8$ |
| 1e-2 | $35.4 \pm 0.2$ | $35.0 \pm 0.1$ | $34.9 \pm 0.3$ | $36 \pm 2$ | $41 \pm 3$ |
| 3e-2 | $550 \pm 250$ | $120 \pm 80$ | $160 \pm 5$ | $210 \pm 60$ | $500 \pm 20$ |

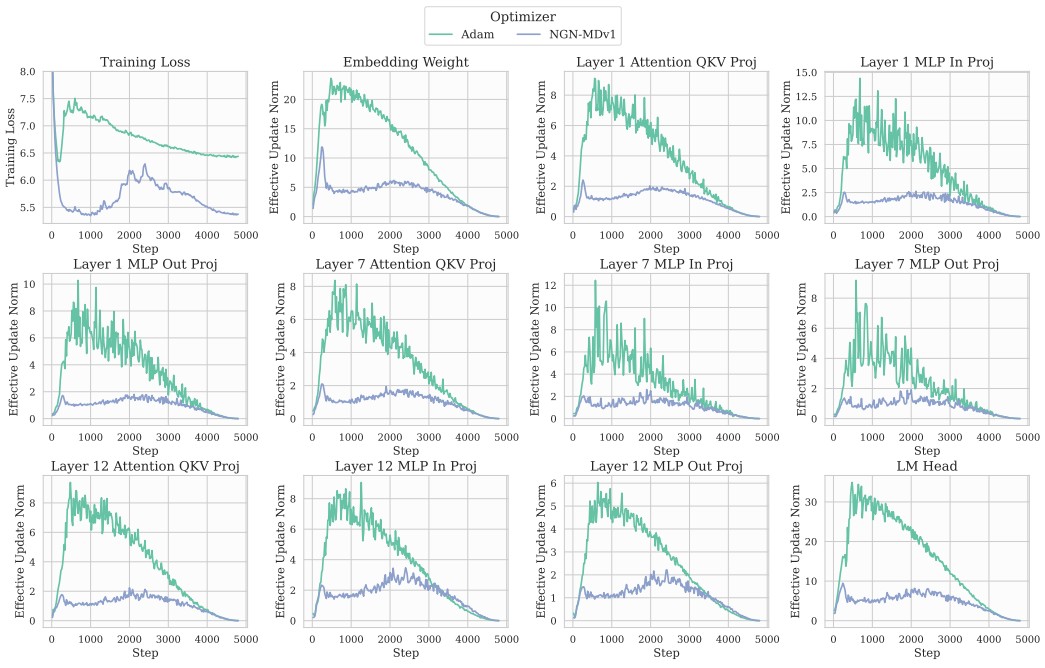

Figure I.13: Magnitude of updates when training 160M language model with Adam and NGN-MDv1 and step-size hyperparameter 0.03.

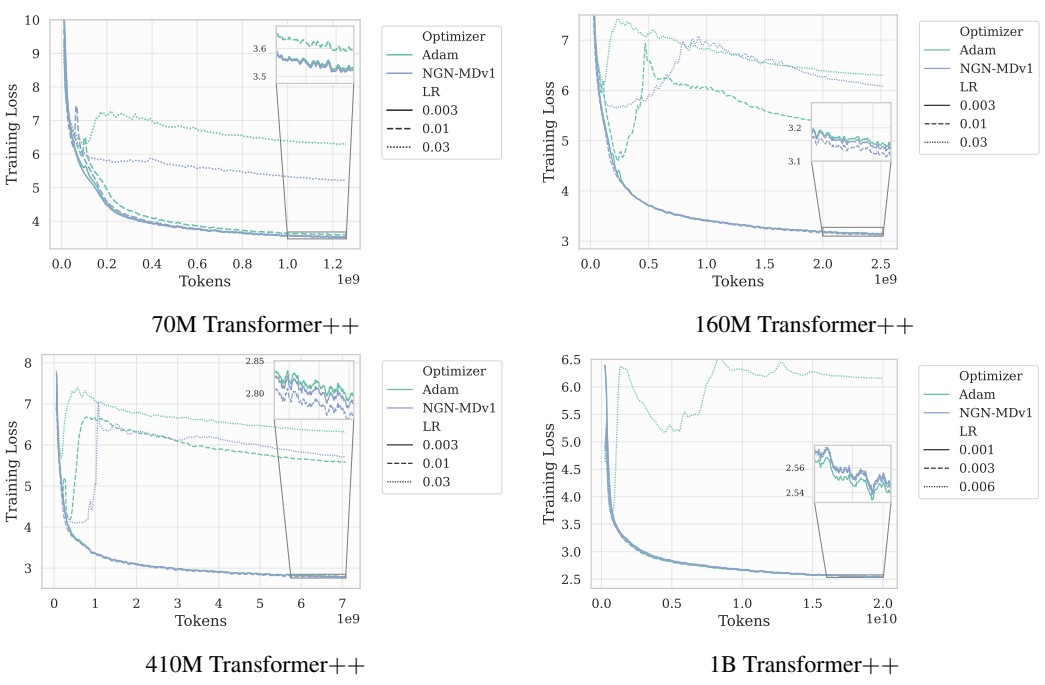

Figure I.14: Training dynamics when training language model at different sizes.

