# OpenReview forum: "Enhancing Optimizer Stability: Momentum Adaptation of The NGN Step-size"
_NeurIPS.cc/2025/Conference — NeurIPS 2025 poster_

### Official Review · Reviewer_R39v · 2025-06-16

**Clarity:** 3
**Significance:** 2
**Originality:** 3
**Rating:** 3
**Confidence:** 4

**Summary:**

This paper introduces new optimization algorithms, NGN-M and NGN-MD, which enhance the recently proposed NGN (Non-negative Gauss-Newton) step-size method by incorporating momentum and coordinate-wise (diagonal) updates. The primary goal is to improve the stability and robustness of the optimizer with respect to the step-size hyperparameter, a common issue with adaptive optimizers like Adam, especially in training large models like Transformers.

The paper introduces NGN-M: A momentum-based version of the NGN step-size. The authors provide a theoretical analysis showing that NGN-M achieves a convergence rate of O(1/ K) in the convex setting without requiring restrictive assumptions like bounded gradients or the interpolation condition.

The paper also introduces NGN-MD: A variant that integrates a coordinate-wise diagonal preconditioning strategy with momentum, making it suitable for training large-scale models like Transformers. Two versions, NGN-MDv1 and NGN-MDv2, are proposed.

Empirical results demonstrate that NGN-M and NGN-MD are significantly more robust to the choice of the step-size hyperparameter compared to state-of-the-art optimizers like Adam, SGDM, and Momo. They achieve comparable or better performance while maintaining stability over a much wider range of learning rates.

**Questions:**

1.  Many recent studies have used the affine gradient noise assumption. The paper claims existing literature relying on bounded gradient is not true.

2. Minor typos.

2.1 Page 6, Line 190: "MomSPS and ALR-SMAG were shown to converge up to a non-vanishing neighborhood of the solution only  " - The word "only" seems misplaced and the footnote reference is superscripted. Suggest rephrasing for clarity, e.g., "...were shown to converge only to a non-vanishing neighborhood...".

**Ethical Concerns:**

["NO or VERY MINOR ethics concerns only"]

**Final Justification:**

The authors' response resolved my comments on the lack of non-convex analysis and clarified the related hyperparameter choices.

However, the paper lacks at least a strong baseline to demonstrate the efficacy of the proposed method in training nowadays LLMs. This point is quite important to me given that this is a paper positioning as a fundamental optimizer design, which should demonstrate its efficacy in most challenging and real optimization tasks.

**Limitations:**

yes

**Quality:**

3

**Strengths And Weaknesses:**

**Strength**

1. Well-Motivated Algorithms: The paper extends the NGN step-size to incorporate momentum and diagonal adaptive learning, which is important for NGN able to apply to state-of-the-art deep learning models.  The proposed combination of the NGN step-size with momentum and diagonal preconditioning is a logical and well-justified extension of prior work.


2. Comprehensive Empirical Evaluation: The experiments are extensive, covering a wide range of models (CNNs, ViTs, LLMs), datasets (from small to very large-scale), and comparisons against relevant popular optimizers. This provides evidence for the practical utility of the proposed methods. The stability plots (Figures 2, 3, and 4) are particularly effective at illustrating the main claim of the paper—that NGN-M and NGN-MD are significantly more robust to the step-size hyperparameter than their counterparts.

**Cons**

Theoretically, the analysis does not cover nonconvex objectives, which has been widely studied in recent convergence analysis of Adam. Moreover, their assumption on gradient noise is affine, rather than bounded.

It is of benefit to study the hyperparameter sensitivity under a clean setup: no learning rate scheduling, no weight decay and no extremely large batch. However, the paper lacks at least a strong baseline to demonstrate the efficacy of the proposed method in training nowadays LLMs. For example, the Llama 2, with standard weight decay, 4M batch size and standard learning rate scheduling, the default setting in available torchtitan codebase. This is a significant gap, as NGN-MD is positioned as the more practical algorithm for large models.

---

> ### Author Rebuttal · Authors · 2025-07-30
>
> We thank the reviewer for valuable comments.
>
> ***Answer to W1:*** In Section C, ***we provide a nonconvex convergence analysis for NGN‑D***, which may have been overlooked by the reviewer—we kindly ask them to revisit this section. Moreover, that analysis readily extends to the ***more general ABC noise model*** [1], which covers the affine noise case mentioned by the reviewer as a special instance. We will include this broader analysis in the revised manuscript.
>
> We also agree that developing a nonconvex theory for NGN‑M would strengthen the analysis, and we mention this explicitly in the conclusion as an important direction for future study. While convexity is typically considered a strong assumption, especially for neural networks training, a growing body of empirical evidence shows that ***convex convergence bounds often predict the behavior of large‑scale language models quite accurately*** [1], and that neural loss landscapes frequently ***exhibit convex‑like regions around the minimizer*** [3-5]. Finally, our theoretical contributions significantly advance prior Polyak‐stepsize adaptive methods by ***relaxing assumptions*** and ***tightening convergence guarantees***, and we validate these findings through a comprehensive empirical evaluation.
>
> [1] Khaled and Richtarik. Better theory for SGD in the nonconvex world. TMLR, 2023.
>
> [2] Schaipp et al., The surprising agreement between convex optimization theory and learning-rate scheduling for large model training. ICML 2025
>
> [3] Tran et al., Empirical tests of optimization assumptions in deep learning. arXiv preprint arXiv: 2407.01825.
>
> [4] Islamov, et al. Loss landscape characterization of neural networks without over-parametrization. NeurIPS 2024
>
> [5] Guille-Escuret et al. No wrong turns: The simple geometry of neural networks optimization paths. arXiv preprint arXiv: 2306.11922, 2023.
>
> ***Answer to W2:*** We highlight that the results demonstrated in Figure 2 ***do not use any lr scheduling and weight decay***, as the reviewer suggests, since they are not required to achieve strong performance in those settings. We demonstrate an improved stability of NGN-M and NGN-MDv1 to the lr hyperparameter. Moreover, we do not claim that the proposed adaptive scheme of setting lr hyperparameter replaces other techniques such as warm-up or weight decay. In contrast, ***all of them can be combined together for a further improvement in robustness*** of the algorithms since warm-up and weight decay are known to improve sensitivity of algorithms to the choice of the lr hyperparameter [3]. Therefore, for training tasks where warm-up and weight decay are necessary, we ***employ standard training procedures*** that are summarized in Section J.1 and Table 3.
>
> The empirical part of the work involves testing the proposed algorithms on ***10 datasets***, including both image and text source of data; ***6 types of architectures*** (MLP, VGG, Resnet, LSTM, ViT, Transformer) with the largest ***1B model***. In all the cases, the proposed algorithms achieve competitive performance with an increased robustness to the lr hyperparameter, which is the main focus of the work. In our view, this ***comprehensive set of experiments sufficiently supports the validity of our claims***. While we agree that scaling to even larger models and datasets would strengthen the empirical evidence, such experiments are not feasible within the rebuttal period (a single run of the 1B model with our compute resources takes 11 hours on 8xH100-80Gb). Demanding larger-scale experiments would exceed our available compute resources and, more broadly, raises concerns about fairness since such a requirement tends to favor institutions with disproportionate access to large-scale GPU infrastructure, rather than evaluating contributions on the basis of methodological innovation.
>
> [3] Wortsman, Mitchell, et al. "Small-scale proxies for large-scale transformer training instabilities." arXiv preprint arXiv:2309.14322 (2023).
>
> ***Answer to Q1:*** We would like to clarify this point. In point 1 of the contributions, we primarily focus on the literature of adaptive methods with a Polyak stepsize, where zero interpolation (see [45, 53, 79]) and bounded gradient (see [66]) assumptions were used to demonstrate convergence. We are aware of other adaptive algorithms (e.g., Adam) that have been analyzed under the affine gradient noise assumption; however, they are outside the scope of that comparison. We will clarify this inaccuracy in the revised version of the paper. Moreover, as it was stated above NGN-D algorithm can be analyzed under a more relaxed ABC assumption on the gradient noise: see the answer to W1.
>
> ***Answer to Q2:*** We thank the reviewer for pointing this out. We will revise the writing accordingly.

---

> > ### Author Response · Authors · 2025-08-05
> > **Reminder about Authors-Reviewers discussion**
> >
> > Dear Reviewer,
> >
> > Thank you for your valuable feedback. In the rebuttals, we have clarified both theoretical and empirical results, added relevant references to support our claims, and highlighted the significance and novelty of our results. Therefore, we would like to initiate the discussion. If there are any remaining concerns, we would be glad to address them.

---

### Official Review · Reviewer_HqrG · 2025-06-16

**Clarity:** 2
**Significance:** 3
**Originality:** 3
**Rating:** 4
**Confidence:** 3

**Summary:**

The paper addresses the sensitivity of momentum-based optimizers to hyperparameter variance to improve the stability of neural network training. Specifically, the authors adapt the Non-negative Gauss-Newton (NGN) stepsize method to the optimizers incorporating momentum. Empirical experiments show the proposed optimizers, NGN-M and NGN-MD, achieve more stable performance when facing parameter variance, and comparable performance to traditional optimizers like Adam.

**Questions:**

Apart from the weaknesses noted above, I would appreciate if the authors could answer the following questions:

1. When comparing stability across optimizers (NGN-M, NGN-MDv1, NGN-MDv2, Momo, momo-Adam, SGDM and Adam), is it fair to adjust $c$ and $\alpha_0$ for the comparison?

2. The discussion of NGN-MD is relatively shorter than that of NGN-M. Are there any drawbacks to using NGN-MD, such as increased training time or other trade-offs?

3. In Related Work, the authors focus on the Polyak step-size framework, but there are other methods aiming to improve training stability (e.g., [1–3]). Could you briefly compare those approaches with the PS-based method and highlight the advantages of adopting the Polyak framework?

[1] Zhang, H. R., Li, D., & Ju, H. (2023). Noise stability optimization for finding flat minima: A hessian-based regularization approach. arXiv preprint arXiv:2306.08553.

[2] Gomes, D. M., Zhang, Y., Belilovsky, E., Wolf, G., & Hosseini, M. S. (2024). AdaFisher: Adaptive Second Order Optimization via Fisher Information. arXiv preprint arXiv:2405.16397.

[3] Li, X., Luo, J., Zheng, Z., Wang, H., Luo, L., Wen, L., ... & Xu, S. (2024). On the Performance Analysis of Momentum Method: A Frequency Domain Perspective. arXiv preprint arXiv:2411.19671.

**Ethical Concerns:**

["NO or VERY MINOR ethics concerns only"]

**Final Justification:**

My concerns raised during review were (1) clarity issues, (2) missing experiments, (3) comparison fairness, (4) broader related work needed. In the rebuttal, the authors address (1), (3), (4). While I still think that RL would be helpful for a more comprehensive evaluation of their method. However, after the rebuttal, I think it is an optional choice considering the response given by the authors and the large experiments the authors have conducted. My opinion is that the strengths of the paper outweigh its weaknesses.

**Limitations:**

Yes

**Paper Formatting Concerns:**

It would be better to insert a page break before the paper checklist so that it starts on its own page.

**Quality:**

3

**Strengths And Weaknesses:**

**Strengths**
- The authors naturally extends the NGN to momentum-based optimizers to address the hyperparameter sensitivity issue, where the motivation is well justified.
- The authors conduct extensive experiments and theory analysis to back up and prove the effectiveness of their method.

**Weaknesses**
-  **Captions, notation, and equations lack clarity**
   - The caption of Table 1 defines $x^*$ but never uses it.
   - $S_k$ appears in the NGN-M equations without explanation (though can be inferred from later texts).
   - Figure 1 labels subfigures as “left”, “second left”, “third and fourth”, yet line 123 refers to “left” when it seems to mean “second left.” Also, NGN-M isn’t identified as Version 1 until the paragraph’s final sentence, causing confusion.
   - Equations such as $f^* = \arg\min_x f(x)\in\mathbb{R}$ are nonstandard.

   I would encourage the authors to review all notations and equations for consistency and clarity or it would be difficult to follow.

-  **Lack of Reinforcement Learning(RL) experiments**
   RL training is well-known for its high sensitivity to hyperparameters, so it would be valuable to know whether NGN-M/NGN-MD deliver similar stability gains in that domain, which is of great importance. Considering the limited time, even a brief comparison on one or two RL tasks would make the evaluation more convincing.

---

> ### Author Rebuttal · Authors · 2025-07-30
>
> We thank the reviewer for valuable comments.
>
> ***Answer to W1:*** We thank the reviewer for a careful read of our paper. We will incorporate all proposed changes in the revised version of the work. The notation for $f^* \in \mathbb{R}$ means that the objective is bounded from below, i.e., $f^* > -\infty$.
>
> ***Answer to W2:*** Thank you for the suggestion to apply our results to reinforcement learning (RL). However, we would like to emphasize that our current experimental focus on the standard classification setting is deliberate: sensitivity to hyperparameters is already a well-documented and significant challenge in this domain. By demonstrating the robustness of our method in such a widely studied and sensitive setting, we believe our work already makes a significant contribution. But we agree that RL is an exciting future direction due to its high sensitivity, and we will mention this in the paper and consider it as follow-up work, thank you again for pointing this out.
>
> ***Answer to Q1:*** In our view, it is a fair comparison. Momo and NGN-M (Momo-Adam and NGN-MDv1) are based on the SPS (Stochastic Polyak Stepsize) and the NGN stepsizes, respectively, and they have hyperparameters $\alpha_0$ and $c$. These hyperparameters play a similar role: ***they adjust the effective lr hyperparameter*** during training, not allowing it to be larger than $\alpha_0$ or $c$, respectively. We therefore believe it is fair to adjust these parameters and would be happy to discuss this further if the reviewer thinks otherwise.
>
> ***Answer to Q2:*** NGN-MD is an extension of the NGN-M algorithm. Therefore, all the discussion on the lr hyperparameter sensitivity of NGN-M applies to NGN-MD as well. Moreover, the derivation of the NGN-M update rule can be seen as a special case of NGN-MDv1 with $\mathbf{\Sigma}_k=\mathbf{I}.$ From an implementation point of view, using a general diagonal preconditioner $\mathbf{\Sigma}_k$ leads to per-parameter adaptive lr. In the case of RMSprop preconditioning, performing one step of training increases since we need to compute the second-order momentum $v^k$. To link, the difference in training time of NGN-M and NGN-MDv1 is similar to that between SGD and Adam. However, the robustness of NGN-MDv1 also depends on the preconditioning. In the cases when the preconditioning is sensitive to the changes in the loss landscape or hyperparameters, NGN-MDv1 might be less robust to the choice of the lr hyperparameter than NGN-M.
>
> ***Answer to Q3:*** [1] This paper studies how to set momentum parameters $\beta_1$ and $\beta_2$ based on frequency domain analysis. However, we did not read any claim about the robustness of their approach to changes of the hyperparameters. Based on Figure 4 in [1], the ***hyperparameters should be chosen in a specific way***, which may limit their robustness in practice. Therefore, ***we ask the reviewer to clarify which part of the work is related to ours***. Nonetheless, their idea on changing the momentum parameter is orthogonal to ours, which makes the comparison difficult. But their momentum scheduling can be combined with our NGN-M algorithm, which would potentially lead to improved results.
>
> [2] Thank you for the reference, and we will cite this work in the revision. [2] introduces AdaFisher, a ***novel way to choose diagonal preconditioning*** by leveraging the composite structure of neural networks. Specifically, the authors derive a diagonal ***approximation to the Fisher Information Matrix*** (FIM) and use it to rescale each parameter update. This curvature‐aware scaling yields more accurate local stepsizes and, in their experiments, substantially reduces sensitivity to the lr choice when training ResNet‑50 on CIFAR‑100. It therefore remains an open question whether the same stability gains carry over to other architectures and datasets. Therefore, the robustness of their approach is underexplored. However, their new ***AdaFisher preconditioning can be incorporated into NGN-MDv1***, which potentially leads to ***better curvature approximation of AdaFisher and improved stability of NGN-MDv1*** to the choice of the lr hyperparameter.
>
> [3] The approach is based on computing the update direction as $\frac{1}{2}(\nabla f(x+u) + \nabla f(x-u)),$ where $u$ is sampled from some distribution. Such an approach leads to ***convergence to flatter minima*** where the trace of the Hessian is minimized, i.e., they study another. This is an alternative to the SAM algorithm, which minimizes the maximum eigenvalue of the Hessian. However, this approach is again orthogonal to ours. While the method potentially might lead to flatter minima, the efficiency of the algorithm is tested; it is still sensitive to the choice of the lr hyperparameter: see Figure 4 in [3], which is the main focus of our work. However, the main disadvantage of this approach is the necessity of computing at least two backpropagations per step, which increases both time and memory consumption during the training when the model is sufficiently large. Thus, ***we ask the reviewer to clarify which part of the work is related to ours*** since this work seems not to study robustness to the choice of hyperparameters.

---

> > ### Comment · Reviewer_HqrG · 2025-08-03
> >
> > Thank you for the detailed explanations and clarifications. My major concerns about comparison fairness and the tradeoff of methods have been well addressed. Regarding Q3: my intent in suggesting additional related work was to include studies on training robustness beyond Polyak-based frameworks for a more comprehensive comparison. Based on the authors’ response, I agree that [1] and [3] are not directly related to hyperparameter sensitivity (e.g., learning rates), which is the specific focus of this paper. Instead, [1] addresses robustness via flatness, and [3] provides a frequency-domain analysis that guides parameter settings for overall performance. I appreciate the clarification and will increase my score accordingly.

---

> > > ### Author Response · Authors · 2025-08-03
> > > **Response to the Reviewer HqrG**
> > >
> > > We would like to thank the reviewer for engaging in the discussion and for increasing the score.

---

### Official Review · Reviewer_3pMo · 2025-07-02

**Clarity:** 3
**Significance:** 2
**Originality:** 2
**Rating:** 4
**Confidence:** 3

**Summary:**

It extends the NGN step-size to consider momentum. Besides proposing the momentum-based version, NGN-M, it proves the standard convergence under less restrictive assumptions. Additionally, it conducts experiments to show the enhanced robustness to the choice of the step-size hyperparameter while delivering performance that is comparable to or surpasses other state-of-the-art optimizers.

**Questions:**

1. Can you conduct experiments on larger models and datasets? e.g.， pretrained MoE LLMs
2. Can you add LR scheduler and weight decay in experiments?
3. Can you give the variances of different step sizes for all compared methods?

**Ethical Concerns:**

["NO or VERY MINOR ethics concerns only"]

**Final Justification:**

Since my main concerns have been addressed, I plan to maintain my positive score.

**Limitations:**

yes

**Quality:**

3

**Strengths And Weaknesses:**

Strengths:
1. It provides a theoretical analysis and proof for the explanation.
2. It conducts comprehensive experiments on different tasks and models.
3. Good introduction on the background and related work to show the position in the literature.

Weakness:
1. The proposed algorithm is the extension of NGN, so the solution itself seems to be incremental.
2. The robustness is not well quantified by measuring some metrics like performance variance.
3. Since learning rate schedulers and weight decay are widely adopted in practice, it is a pity that the experiment settings ignore such practical factors. Therefore, it leaves room for the claim to be questioned.

---

> ### Author Rebuttal · Authors · 2025-07-30
>
> ***Answer to W1:*** We agree that NGN-M builds upon the NGN method, but this extension is far from trivial. In Section 3.1, we compare two variants of coupling momentum and the NGN stepsize. We demonstrate that ***the second variant*** (Adam-type momentum) ***does not preserve the non-divergence feature of NGN*** in the convex case. Moreover, in the convex setting, the coupling should lead to acceleration if $c$ is small enough. Hence, the integration of momentum into NGN is not straightforward and requires a careful analysis, which we provide in the paper.
>
> ***Answer to W2:*** We thank the reviewer for this insightful comment, as it allowed us to rethink how to better demonstrate the robustness of the proposed method, and eventually improve our work further. We can quantify the robustness in a similar way to how it was done in [1]. We define $\ell\_{\gamma}$ to be the final metric (e.g., train loss or negative test accuracy that we need to minimize) achieved when training a model with the lr hyperparameter $\gamma\in[a,b]$, while $\ell(x_0)$ is the metric at initialization. Moreover, we define the best possible metric $\ell^\*\_{\gamma_*}$ that can be achieved by training a model with $\gamma\in[a,b],$ i.e., $\ell^\*\_{\gamma_*} = \min\_{\gamma\in[a,b]}\ell\_{\gamma}.$ The lr sensitivity is then defined as $\mathbb{E}\_{\gamma\in[a,b]}[\min\\{\ell\_{\gamma}, \ell(x_0)\\} - \ell^\*\_{\gamma_*}]$. We measure this quantity for various tested algorithms and tasks, approximating the expectation by averaging $\min\\{\ell\_{\gamma}, \ell(x_0)\\} - \ell^\*\_{\gamma_*}$ over the grid of values for $\gamma$. This metric ***captures the maximum deviation from optimal performance***. Across all experiments, both NGN‑M and NGN‑MDv1 show consistently lower sensitivity to the lr hyperparameter than the other baselines, quantitatively confirming our claims.
>
>
>
> | Resnet20, CIFAR10 | SGDM | NGN | Momo | NGN-M | Adam | Momo-Adam | NGN-MDv1 | NGN-MDv2 |
> | -------- | -------- | -------- | -------- | -------- |-------- |-------- | -------- |-------- |
> | Test accuracy LR sensitivity     | 0.22     | 0.10     | 0.13 | ***0.03*** | 0.18 | 0.08 | ***0.04*** | 0.15|
>
> Here, $\gamma \in \{10^{-2}, 3\cdot 10^{-2}, \dots, 3\cdot 10^0, 10, 10^1, 10^2\}$ for the SGDM, NGN, Momo, NGN-M and $\gamma \in \{10^{-4}, \dots, 10^0\}$ for Adam, Momo-Adam, and NGN-MD.
>
>
> | Resnet110, CIFAR100 | SGDM | NGN | Momo | NGN-M | Adam | Momo-Adam | NGN-MDv1 | NGN-MDv2 |
> | -------- | -------- | -------- | -------- | -------- |-------- |-------- | -------- |-------- |
> | Test accuracy LR sensitivity     | 0.25 | 0.17 | 0.16 | ***0.05*** | 0.19 | 0.18 | ***0.10*** | 0.18 |
>
> Here, $\gamma \in \{10^{-2}, \dots, 10^2\}$ for the SGDM, NGN, Momo, NGN-M and $\gamma \in \{10^{-4}, \dots, 10^0\}$ for Adam, Momo-Adam, and NGN-MD.
>
> | ViT, CIFAR10 | SGDM | NGN | Momo | NGN-M | Adam | Momo-Adam | NGN-MDv1 | NGN-MDv2 |
> | -------- | -------- | -------- | -------- | -------- |-------- |-------- | -------- |-------- |
> | Test accuracy LR sensitivity     | 0.28 | 0.19 | 0.24 | ***0.01*** | 0.41 | 0.21 | ***0.06*** | 0.33 |
>
> Here, $\gamma \in \{10^{-1}, 3\cdot 10^{-1}, 10^0, 3\cdot 10^0, 10^1, 3\cdot 10^1\}$ for the SGDM, NGN, Momo, NGN-M and $\gamma \in \{10^{-4}, 3\cdot 10^{-4}, \dots, 10^{-2}, 3\cdot 10^{-2}\}$ for Adam, Momo-Adam, and NGN-MD.
>
>
> | Test perplexity LR sensitivity | 70M | 160M | 410M | 1B |
> | -------- | -------- | -------- | -------- | -------- |
> | Adam | 78.6 | 85.2 | 78.6 | 96.3 |
> | NGN-MDv1 | 25.4 | 30.9 | 25.4 | 15.5 |
>
> Here, for 70M, 160M, 410M models, $\gamma \in \{10^{-3}, 1.7\cdot 10^{-3}, 3\cdot10^{-3}, 5.5\cdot 10^{-3}, 10^{-2}, 3\cdot10^{-2}\}$. For 1B model, $\gamma \in \{10^{-3}, 3\cdot10^{-3}, 5.5\cdot 10^{-3}, 10^{-2}, 3\cdot10^{-2}\}$.
>
> [1] Wortsman, Mitchell, et al. "Small-scale proxies for large-scale transformer training instabilities." arXiv preprint arXiv:2309.14322 (2023).
>
> ***Answer to W3 and Q2:*** We respectfully disagree with this claim. Learning rate scheduling and weight decay are not required to achieve strong performance in our core experiments, as shown in Figure 2. When training Resnet18 on Imagenet1k, we use learning rate decay every 30 epochs by 0.1 without weight decay, while when training ViT on Imagenet1k, we use cosine annealing with warm-up and weight decay. For language modeling, we use linear warm-up followed by cosine annealing. To summarize, for each training task we ***employ standard training procedures*** that are summarized in Section J.1 and Table 3. Moreover, we provide a discussion on how weight decay can be added to NGN-MD in Section H and provide additional ablations by varying the weight decay parameter $\lambda$ in Figure H.1.
>
> ***Answer to Q1:*** The empirical part of the work involves testing the proposed algorithms on ***10 datasets***, including both image and text source of data; ***6 types of architectures*** (MLP, VGG, Resnet, LSTM, ViT, Transformer) with the largest ***1B model***. In all the cases, the proposed algorithms achieve competitive performance with an increased robustness to the lr hyperparameter, which is the main focus of the work. In our view, this ***comprehensive set of experiments sufficiently supports the validity of our claims***. While we agree that scaling to even larger models and datasets would strengthen the empirical evidence, such experiments are not feasible within the rebuttal period (a single run of a 1B model with our compute resources takes 11 hours on 8xH100-80Gb). Demanding larger-scale experiments would exceed our available compute resources, and, more broadly, raises concerns about fairness since such a requirement tends to favor institutions with disproportionate access to large-scale GPU infrastructure, rather than evaluating contributions on the basis of methodological innovation.
>
> ***Answer to Q3:*** We thank the reviewer for this comment. Note that Figures 5 and J.9 do show the standard deviation. However, since the stepsize changes rapidly, this cannot be observed in Figure 5. The reviewer still can observe the variance in the 2nd and 4th columns of Figure J.9. The standard deviation is relatively small. We will update the plots by using exponential smoothing so that the variance can be better observed in all figures.

---

> > ### Comment · Reviewer_3pMo · 2025-08-02
> >
> > Thank you for your detailed explanation and clarification. Since my main concerns have been addressed,  I plan to maintain my positive score.

---

> > > ### Author Response · Authors · 2025-08-03
> > > **Response to the Reviewer 3pMo**
> > >
> > > We would like to thank the reviewer for engaging in the discussion and for a positive evaluation of our work. We are happy that the main concerns are addressed, and answering them has allowed us to improve our work.

---

### Official Review · Reviewer_ST2q · 2025-07-03

**Clarity:** 3
**Significance:** 3
**Originality:** 3
**Rating:** 4
**Confidence:** 4

**Summary:**

Summary:

This paper introduces {NGN-M}, a momentum-enhanced variant of the {Non-negative Gauss-Newton step-size} (NGN) method by Orvieto & Xiao. It also proposes {NGN-MD}, which incorporates coordinate-wise diagonal step-size adaptation similar to the classic Adam. The main goal is to improve optimizer {stability} with minimal hyperparameter tuning, especially the learning rate. Key contributions include: (1) a theoretical convergence guarantee for NGN-M at the standard $\mathcal{O}(1/\sqrt{K})$ rate in convex stochastic optimization {without} requiring restrictive assumptions like interpolation or bounded gradients (which prior adaptive methods often needed); (2) extension to a diagonal-momentum version (NGN-MD) to address per-parameter conditioning in deep networks; and (3) experiments demonstrating that NGN-M/D maintain high performance while being significantly less sensitive to learning-rate choices than baselines (SGD with momentum, Adam, and recent adaptive methods).

**Questions:**

Questions:

Please see the weaknesses for question on analysis  — it seems we either require pre-knowledge of k or we need the c to go to 0 which means the method will align to NGN. Did I miss something?

Is there an intuitive way to understand of why momentum improves stability? The authors have cited some prior references too that have made a similar claim, but I am finding it hard to intuitively understand this — if it is just because of variance reduction, surely there would be more direct approaches that achieve that goal ?

This is not super relevant to the paper’s acceptance but it would be interesting to know the authors’ perspective on this — Would it be interesting to explicitly tie in the step size behavior to the warmup kind of schedules? For example — see Small-scale proxies for large-scale Transformer training instabilities. Iclr 2024

Given the lack of theoretical arguments, have the authors given thought to if NGN-MDv1 is always stable, or can the coupling of heavy-ball momentum and an RMSprop preconditioner ever possibly cause divergence ? I understand such analyses are notoriously hard.

**Ethical Concerns:**

["NO or VERY MINOR ethics concerns only"]

**Final Justification:**

I appreciate the authors providing more references for clarity on the "fixed horizon" sort of norms that are accepted in optimization. While I have worked in optimization for several years, I learnt something new from this paper and the discussions. I hope the authors will provide more clarity on these norms in the community so that an uninformed reader can place the results in context. i will raise my score.

**Limitations:**

yes

**Quality:**

2

**Strengths And Weaknesses:**

Strenghts:

The theoretical contribution is one of the paper’s strongest aspects. The authors build on the NGN step-size proposed by Orvieto & Xiao, which was shown to never diverge for \emph{any} learning-rate choice in convex problems and to automatically adapt to curvature. By incorporating heavy-ball momentum (Polyak’s momentum), the paper advances prior theory on combining adaptive step-sizes with momentum. Earlier attempts in this direction often required strong assumptions or achieved weaker guarantees. For example, the original {Stochastic Polyak Step-size} (SPS) method by Loizou et al guarantees convergence to the exact solution only under the interpolation assumption (informally, zero gradient noise at the optimum) or else converges to a neighborhood. Recent work by Oikonomou/Loizou introduced momentum into SPS but similarly could only ensure convergence within a non-vanishing error ball or required decaying step-sizes. In contrast, this paper’s Theorem 4.3 proves that {NGN-M converges at $\mathcal{O}(1/\sqrt{K})$ in convex stochastic settings without assuming interpolation, bounded gradients, or even bounded iterates}. This is a significant theoretical improvement over prior adaptive momentum methods. The proofs appear mathematically rigorous and leverage techniques from both momentum analysis and adaptive step-size methods. The derivation of the NGN-M update is grounded in a Gauss–Newton approximation to the squared-root loss (following Orvieto et al), but adding the momentum  term raises non-trivial analysis challenges. This paper handles this by using an “Iterative Moving Average” view of momentum and carefully coupling the momentum term with the adaptive step-size in their Lyapunov analysis. They highlight that unlike prior works, they do not require a bounded domain or bounded gradients assumption for stability. This is a notable strength: the method is provably stable for arbitrarily large chosen $c$ (learning-rate hyperparameter), just like the non-divergence property of the NGN. The proofs (mostly deferred to appendices) appear to be sound; they build on techniques from convex optimization and stochastic process bounds.

The paper compliments its theoretical contribution with extensive set of experiments covering both simpler settings to illustrate stability as well as large scale experiments to demonstrate effectiveness of the proposed technique. NGN-M converges reliably for {every} tested $c$, whereas SGDM diverges outside a narrow band of tuned values (even at moderate mis-tuning). This  supports the claim that NGN’s adaptive step-size “never diverges” and that momentum further helps by accelerating convergence for intermediate $c$ values without sacrificing stability. The evaluation is done vs several important baselines. And across many different settings, the proposed method-variants seem to outperform these baselines on validation accuracy. The experimental methodology is also sound.

The paper is also well-written with the contributions, theorem statements, contexts and existing works as well as assumptions and proofs clearly laid out.

Weaknesses:

For analysis theorem 4.3 requires pre-knowing total number of iterations (so c can be set accordingly to obtain the bound to be the same as SGDM), or as done in the appendix, follow a diminishing schedule on c, where c goes to 0 as k increases — in this case, does the method not “eventually” boil down to NGN without the momentum or is close to it?

The runtimes can be possibly concerning — the major contribution of the paper seems to be removing the need for tuning the hyper-parameter, but if the range of stable SGDM is largely known from previous experiences on similar datasets, I am a bit worried that the increased runtimes can possibly offset this advantage. For the reported generalization performance, were the models trained for the same runtime? Or same number of epochs? If it is the latter, and the fact that NGN does “more work” per iteration, is it possible that running SGDM (within the acceptable stable set of parameters ) can lead to better performance?

A brief ablation on $\beta$ would strengthen the empirical section.


Minor:
“[2, 64, 2, 79, 53]” (line 80) contain a duplicated “[2]” entry

---

> ### Author Rebuttal · Authors · 2025-07-30
>
> We thank the reviewer for valuable comments
>
> ***Answer to W1:*** The reviewer is right, we acknowledged this concern in the discussion after Theorem 4.3. Nonetheless, in Section F, we demonstrate that for simple functions like convex polynomials, ***the restriction on $\beta$ being small can be avoided***. This is substantiated by experimental evidence showing that ***NGN-M and NGN-MD perform well***. This makes us believe that the restriction on $\beta$ can be avoided but it requires a more careful analysis, which we were not able to obtain at the moment. Moreover, we highlight that other momentum-based algorithms, including ***MomSPS$_{\max}$, have similar limitations***. We also note that we can remove the restriction on $\beta$ by making the stepsize decreasing or under, additional assumptions like bounded iterates similar to [53]. While these modifications improve the theoretical analysis, we believe they would make the algorithm less favorable in practice.
>
> ***Answer to W2:*** We directly addressed the concern regarding runtime in ***section G.2*** (also mentioned in the conclusion). Table 2 reports the time needed for performing one step of training and optimizer.step() when training language models at different scale. We observe that the time needed for one training step of NGN-MDv1 ***does not increase significantly and is comparable to that of AdamW*** (see middle column of Table 2). This happens because *** backpropagation is the most time-consuming part*** of training. Therefore, the final training time of AdamW and NGN-MDv1 is comparable: for all tested scales, it ***increases by a maximum of 3%***. Therefore, running the algorithms under a fixed time budget will not affect the results significantly, and the improved robustness of NGN-M/NGN-MDv1 to the choice of lr hyperparameter will remain.
>
> ***Answer to W3:*** Following the reviewer's comment, we provide sweeps varying not only the lr hyperparameter but also the momentum parameters $\beta_1,\beta_2$ when training 70M language model on FineWeb dataset. To do that, we fix $\beta_1=0.9$ (or $\beta_2=0.95$) and make a sweep over lr and $\beta_2$ (or lr and $\beta_1$). We report the final test perplexity averaged over 3 runs for each set of hyperparameters. We thank the reviewer for this valuable suggestion, which has enabled us to strengthen our empirical evaluation. We will include these new results in the revised manuscript.
>
> |NGN-MDv1 $(\beta_2=0.95$)|$\beta_1=0.6$|$\beta_1=0.8$|$\beta_1=0.9$| $\beta_1=0.99$|Adam ($\beta_2=0.95$)|$\beta_1=0.6$|$\beta_1=0.8$|$\beta_1=0.9$| $\beta_1=0.99$|
> |-|-|-|-|-|-|-|-|-|-|
> |lr 3e-4|$49.9\pm0.2$|$47.4\pm0.2$|$47.0\pm0.2$|$49.7\pm0.3$||$49.4\pm0.2$|$46.5\pm0.1$|$46.2\pm0.3$|$57\pm1$|
> |lr 1e-3|$41.5\pm0.2$|$39.9\pm0.2$|$38.6\pm0.1$|$40.2\pm0.3$||$41.4\pm0.2$|$39.6\pm0.1$|$38.5\pm0.1$|$45.0\pm0.2$|
> |lr 3e-3|$40\pm1$|$36.9\pm0.3$|$35.9\pm0.1$|$37.2\pm0.1$||$40.7\pm0.1$|$37.0\pm0.1$|$36.0\pm0.1$|$220\pm70$|
> |lr 1e-2|$54\pm16$|$37\pm2$|$34.7\pm0.3$|$35.9\pm0.1$||$160\pm60$|$41\pm2$|$36\pm2$|$210\pm110$|
> |lr 3e-2|$278\pm6$|$129\pm2$|$34.6\pm0.1$|$35.6\pm0.1$||$420\pm20$|$340\pm50$|$320\pm60$|$330\pm130$|
>
> |NGN-MDv1 ($\beta_1=0.9$)|$\beta_2=0.6$|$\beta_2=0.8$|$\beta_2=0.9$| $\beta_2=0.95$|$\beta_2=0.999$|Adam ($\beta_1=0.9$)|$\beta_2=0.6$|$\beta_2=0.8$|$\beta_2=0.9$| $\beta_2=0.95$|$\beta_2=0.999$|
> |-|-|-|-|-|-|-|-|-|-|-|-|
> |lr 3e-4|$51.8\pm0.6$|$49.2\pm0.4$|$47.8\pm0.3$|$47.0\pm0.2$|$47.0\pm$0.2||$46.1\pm0.2$|$46.6\pm0.1$|$46.5\pm0.2$|$46.2\pm0.3$|$46.5\pm0.1$|
> |lr 1e-3|$42.6\pm0.3$|$40.5\pm0.1$|$39.3\pm0.2$|$38.6\pm0.1$|$38.9\pm0.1$||$38.8\pm0.1$|$39.0\pm0.2$|$38.9\pm0.1$|$38.5\pm0.1$|$39.5\pm0.6$|
> |lr 3e-3|$39.4\pm0.2$|$37.5\pm0.2$|$36.3\pm0.1$|$35.9\pm0.1$|$36.5\pm0.4$||$38.8\pm0.3$|$36.3\pm0.1$|$36.1\pm0.2$|$36.0\pm0.1$|$36.7\pm0.8$|
> |lr 1e-2|$37.8\pm0.2$|$35.9\pm0.1$|$35.1\pm0.3$|$34.7\pm0.3$|$35.0\pm0.3$||$35.4\pm0.2$|$35.0\pm0.1$|$34.9\pm0.3$|$36\pm2$|$41\pm3$|
> |lr 3e-2|$37.8\pm0.3$|$35.8\pm0.1$|$34.9\pm0.1$|$34.6\pm0.1$|$250\pm50$||$550\pm250$|$120\pm80$|$160\pm5$|$210\pm60$|$500\pm20$|
>
> We summarize our findings as follows:
>
> - Low lr ($\le$ 3e-3): NGN-MDv1 and Adam show similar sensitivity to changes in both $\beta_1$ and $\beta_2$.
>
> - Moderate lr (1e-2): NGN-MDv1 is noticeably more robust than Adam to extremes of $\beta_1$, while both optimizers perform similarly across $\beta_2$ (though Adam’s performance degrades slightly at $\beta_2 = 0.999$).
>
> - High lr (3e-2): Both methods suffer when $\beta_1$ is small (or $\beta_2$ is large), but NGN-MDv1 recovers lower perplexity at larger $\beta_1$ values (smaller $\beta_2$ values), whereas Adam fails to reach comparable performance.
>
> To conclude, NGN-MDv1 demonstrates greater robustness to changes in momentum parameters at high lr, and consistently attains lower perplexity than Adam, even when both methods’ performance deteriorates (we refer to the cases when both algorithms cannot achieve perplexity $\lesssim$ 50).
>
> ***Answer to Q1:*** Demonstrating the improved stability from using momentum is a ***challenging task***, especially for an adaptive algorithm like NGN-M. However, to gain intuition, we can consider minimizing a convex quadratic function $\min_{x}\frac{1}{2}x^\top Q x + b^\top x + c.$ The ***maximum allowed stepsize of GD*** that leads to convergence is $\boldsymbol{\gamma}<\mathbf{\frac{2}{\boldsymbol{\lambda}\_{\max}(Q)}}$, where $\lambda_{\max}(Q)$ is the maximum eigenvalue of $Q$. When using Heavy-ball algorithm instead, the maximum allowed stepsize for convergence is $\mathbf{\frac{2(1+\boldsymbol{\beta})}{\boldsymbol{\lambda}\_{\max}(Q)}}$ [1], which is about ***two times larger than for GD*** for a standard choice of $\beta=0.9$. We acknowledge that training neural networks is far from quadratic,s but it gives an intuition why we should expect an improved robustness to the choice of the lr hyperparameter. This is also supported by recent works that observe ***convex-like structures in the loss landscape*** of neural networks [2-4] and ***agreement between convex theory and practice*** [5].
>
> [1] Polyak, Some methods of speeding up the convergence of iteration methods. Ussr comp. math. and math. phys., 1964
>
> [2] Tran et al., Empirical tests of optimization assumptions in deep learning. arXiv preprint arXiv:2407.01825.
>
> [3] Islamov, et al. Loss landscape characterization of neural networks without over-parametrization. NeurIPS 2025
>
> [4] Guille-Escuret et al. No wrong turns: The simple geometry of neural networks optimization paths. arXiv preprint arXiv:2306.11922, 2023.
>
> [5] Schaipp et al. The surprising agreement between convex optimization theory and learning-rate scheduling for large model training. ICML 2025
>
> ***Answer to Q2:*** We believe there is a strong connection between our adaptive approach of setting the lr hyperparameter and lr scheduling, including warmup. According to Figure 5 in the main paper and Figures J.9-J.12 in the appendix, our proposed adaptive scheme ***serves as a safeguard*** during training. If the lr hyperparameter is too large, which might lead to unstable convergence/divergence, our scheme makes the effective lr smaller and stabilizes the training. This mechanism is similar in spirit to warm-up, but with two key differences. First, unlike traditional warm-up, which is typically applied only at the beginning of training, our approach can activate dynamically at any point during training. Second, while warm-up schedules require manual specification of the warm-up duration, our method automatically adapts based on the local curvature. We thank the reviewer for the reference and will incorporate this discussion into the revised version.
>
> ***Answer to Q3:*** We expect NGN-MDv1 to retain the robustness properties of NGN-M in most cases. However, its robustness also depends on the choice of preconditioning. If the chosen preconditioner is highly sensitive to variations in the loss landscape or hyperparameters, NGN-MDv1's resilience to learning rate selection may degrade. In such cases, alternative preconditioning strategies, such as those based on Shampoo [6] or the Fisher information matrix [7], might be worth exploring to improve stability.
>
> [6] Gupta et al., Shampoo: Preconditioned stochastic tensor optimization, ICML 2018
>
> [7] Gomes et al., Adafisher: Adaptive second order optimization via fisher information. arXiv preprint arXiv:2405.16397, 2024.

---

> > ### Author Response · Authors · 2025-08-05
> > **Reminder about Authors-Reviewers discussion**
> >
> > Dear Reviewer,
> >
> > Thank you for your valuable feedback. In the rebuttals, we have clarified both theoretical and empirical results, added relevant references to support our claims, and conducted new experiments to strengthen our work. Therefore, we would like to initiate the discussion. If there are any remaining concerns, we would be glad to address them.

---

> > ### Comment · Reviewer_ST2q · 2025-08-06
> >
> > I appreciate the ablation studies and the detailed answers. The point about pre-knowing number of iterations so c can be setup still bothers me, it is effectively "cheating", i can set it to whatever and obtain whatever bounds I like, no? Am I misunderstanding this? And without it the method effectively reduces the standard NGN. I think this requires a more careful analysis, or removed alltogether, the current form of the theorem seems meaningless.

---

> > > ### Author Response · Authors · 2025-08-06
> > > **Response the Reviewer**
> > >
> > > Dear Reviewer,
> > >
> > > Thank you for your insightful feedback. We would be happy to provide a more detailed response.
> > >
> > > - First, deriving iteration complexity with a known number of iterations $K$ is a standard technique in optimisation. We refer to classic results in the literature on the convergence of SGD, which also includes lecture notes [1, 2, 3, 4], and more recent works [6, 7, 8, 9]. We highlight that it is important to obtain $O(1/\sqrt{K})$ convergence to demonstrate the optimal iteration complexity of the algorithm. We trust that the 8 references we provide will be sufficient to convince the reviewer.
> > >
> > >   [1] Ghadimi and Lan, Stochastic first-and zeroth-order methods for nonconvex stochastic programming, SIAM journal on optimization, 2013 (~1900 citations)
> > >
> > >   [2] Nemirovski et al., Robust Stochastic Approximation Approach to Stochastic Programming, SIAM Journal on Optimization, 2009 (~3000 citations)
> > >
> > >   [3]  Boyd, Xiao, and Mutapcic. "Subgradient methods." lecture notes of EE392o, 2003 (~1400 citations)
> > >
> > >   [4] Duchi, Introductory lectures on stochastic optimization. The mathematics of data, 2018 (~120 citations)
> > >
> > >   [5] Nemirovsky, A. S., D. B. Yudin, and E. Dawson. Wiley-Interscience Series in Discrete Mathematics. Problem complexity and method efficiency in optimization, 1983 (~3300 citations)
> > >
> > >   [6] Hardt, Moritz, Recht, and Singer. Train faster, generalize better: Stability of stochastic gradient descent. ICML (~1600 citations)
> > >
> > >   [7] Li and Orabona. On the convergence of stochastic gradient descent with adaptive stepsizes. AISTATS, (~380 citations)
> > >
> > >   [8] Jain et al., Making the last iterate of sgd information theoretically optimal. COLT, (~90 citations)
> > >
> > >   [9] Yu, Hao, Rong Jin, and Sen Yang. On the linear speedup analysis of communication efficient momentum SGD for distributed non-convex optimization.” ICML 2019 (~450 citations)
> > >
> > >
> > > - Without knowledge of $K$, we indeed have to use a diminishing stepsize schedule $c_k=O(1/\sqrt{k})$ to guarantee convergence. Due to the restriction on $\beta$, $\beta$ should also diminish. As we discuss in our work, this is a typical issue of Polyak-stepsize-based algorithms [10]. Moreover, prior works on momentum also suggest that setting $\beta=0$ (i.e., considering a method without momentum) gives the fastest rate [9, 10, 11, 12]. Therefore, we believe that this requirement on $\beta$ is not a property of the algorithm, since both NGN-M and NGN-MDv1 work well with the default value $\beta=0.9$ across all tested experimental settings. It is rather an artefact of the existing proof techniques.
> > >
> > >    We also prove that, in the special case of convex polynomials, this requirement becomes unnecessary -- please let us know if that portion of our analysis was overlooked. Additionally, one can entirely drop the assumption by imposing a bounded‐iterate condition and a monotonically decreasing stepsizes, as in [10]. While these modifications improve the theoretical analysis, we believe they would make the algorithm less favorable in practice.
> > >
> > >   We acknowledge that relaxing the requirements on $\beta$ is an important question that should be addressed in future work, albeit a difficult one.
> > >
> > >   [10] Oikonomou and Loizou. Stochastic Polyak step-sizes and momentum: Convergence guarantees and practical performance, ICLR 2025
> > >
> > >   [11] Wang et al. Generalized Polyak step size for first order optimization with momentum. ICML 2023
> > >
> > >   [12] Loizou and Richtárik. Momentum and stochastic momentum for stochastic gradient, newton, proximal point and subspace descent methods." Computational Optimization and Applications, 2020

---

> > > > ### Comment · Reviewer_ST2q · 2025-08-09
> > > >
> > > > Thank you for the response. I will go through the papers and bring up this point in the reviewer discussion. If I am indeed wrong about this I'll happily support the paper.

---

### Note · Authors · 2025-08-11

We thank all reviewers for their thoughtful evaluations and constructive engagement during the discussion phase.

## 1. Reviewers consensus
- Reviewers 3pMo and HqrG confirmed that their concerns were well addressed. We will incorporate their suggestions to improve the quality of our work.

- Reviewer ST2q supports our work under the condition that the works that we mentioned during the rebuttals use similar techniques to study the iteration complexity of algorithms.

- Unfortunately, Reviewer R39v did not reply to the rebuttals.

## 2. Short summary of the rebuttals

- Most of the reviewers' concerns are minor. We addressed them during the rebuttals by providing clarifications on training details, algorithm design, additional experimental results and discussion on prior works.

- The main concern is the restriction on the momentum parameter $\beta$, which is standard in the literature on Polyak-stepsize algorithms. It can be removed under extra assumptions (e.g., bounded iterates, convex polynomials). Experiments show NGN-M and NGN-MD perform well, suggesting the restriction merits deeper analysis, which we plan to pursue in future work.

## 3. Comparison to prior works

- Although some limitations from coupling Polyak-type adaptive stepsize and momentum remain (restriction on $\beta$), we improve the theoretical guarantees by providing exact convergence under the smoothness assumption only.

## 4. Empirical study

- We provide rigorous experimental evaluations (10+1 datasets, 6 types of architecture with the largest model of size 1B). In all the cases, the proposed algorithms achieve competitive performance with an increased robustness to the lr hyperparameter, which is the main focus of the work.

## 5. Conclusion

Our work offers a technically rigorous, novel, and broadly relevant contribution by addressing the important question of algorithms' robustness to lr hyperparameter. We respectfully ask the AC to consider the technical strength, originality, extensive empirical evaluation, and wide applicability of our work when making the final decision.

Given the substantial effort invested in this work, we would greatly appreciate if the AC could provide a summary of the post-rebuttal discussion with the reviewers.

## References

[1] Khaled and Richtarik. Better theory for SGD in the nonconvex world, 2023

[2] Oikonomou and Loizou. Stochastic Polyak step-sizes and momentum: Convergence guarantees and practical performance, 2025

---

### Decision · Program_Chairs · 2025-09-17

**Decision:**

Accept (poster)

**Comment:**

The paper introduces a momentum enhanced version of the NGN method developed by Orvieto and Xiao. The paper also introduced a coordinate-wise diagonal step-size adaptation similar to classic Adam. Like the Orvieto, Xiao paper, the core idea here is to show optimizer stability with minimum parameter tuning required and to do it with step size that can incorporate the elements that have shown practical advantages. The authors prove this theoretically by providing non-asymptotic rates and show some experiments demonstrating their claims.

Overall 3/4 reviewers appreciated the paper's contributions with the rebuttals resolving most of the concerns raised by the reviewers. The 4th reviewer had primary one concern remaining which was experiments on LLMs. I believe the authors have performed a wide-spate of experiments across tasks and settings within reach of an academic setup. While larger scale experiments embellish the paper, from the reviews and discussion it can be concluded that even without that the paper represents a solid contribution to optimization literature and can be published.